# Model-Free Learning with Heterogeneous Dynamical Systems: A Federated LQR Approach

**Han Wang**                                                    *hw2786@columbia.edu*
*Department of Electrical Engineering*
*Columbia University*

**Leonardo F. Toso**                                    *leonardo.toso@columbia.edu*
*Department of Electrical Engineering*
*Columbia University*

**Aritra Mitra**                                                    *amitra2@ncsu.edu*
*Department of Electrical and Computer Engineering*
*North Carolina State University*

**James Anderson**                                    *james.anderson@columbia.edu*
*Department of Electrical Engineering*
*Columbia University*

**Reviewed on OpenReview:** *[https://openreview.net/forum?id=WSRQeCUc3g](https://openreview.net/forum?id=WSRQeCUc3g)*

## Abstract

We study a model-free federated linear quadratic regulator (LQR) problem where $M$ agents with *unknown, distinct yet similar* dynamics collaboratively learn an optimal policy to minimize an average quadratic cost while keeping their data private. To exploit the similarity of the agents' dynamics, we propose to use federated learning (FL) to allow the agents to periodically communicate with a central server to train policies by leveraging a larger dataset from all the agents. With this setup, we seek to understand the following questions: (i) *Is the learned common policy stabilizing for all agents?* (ii) *How close is the learned common policy to each agent's own optimal policy?* (iii) *Can each agent learn its own optimal policy faster by leveraging data from all agents*? To answer these questions, we propose the federated and model-free algorithm `FedLQR`. Our analysis overcomes numerous technical challenges, such as heterogeneity in the agents' dynamics, multiple local updates, and stability concerns. We show that `FedLQR` produces a common policy that, at each iteration, is stabilizing for all agents. Moreover, we prove that when learning each agent's optimal policy, `FedLQR` achieves a sample complexity reduction proportional to the number of agents $M$ in a low-heterogeneity regime, compared to the single-agent setting.

## 1 Introduction

There has been significant progress in the application of model-free reinforcement learning (RL) methods to fields such as video games (Mnih et al., 2015) and robotic manipulation (Rajeswaran et al., 2017; Levine et al., 2016; Tobin et al., 2017). In particular, RL has recently been used in the fine-tuning of pretrained foundation models (Ziegler et al., 2019). Although RL has shown impressive results in simulation, it often suffers from poor sample complexity, thereby limiting its effectiveness in real-world applications (Dulac-Arnold et al., 2019). To resolve the sample complexity issue and accelerate the learning process, federated learning (FL) has emerged as a popular paradigm (Konečný et al., 2016a; McMahan et al., 2017), where multiple similar agents collaboratively learn a common model without sharing their raw data. The incentive for collaboration arises from the fact that these agents are "similar" in some sense and hence end up learning a "superior"

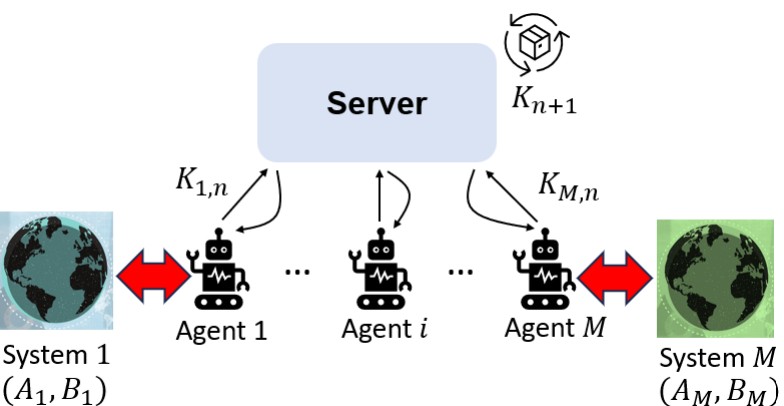

Figure 1: Illustration of the heterogeneous federated LQR setup involving $M$ agents, where the $i$-th agent interacts with its own LTI system characterized by the pair $(A_i, B_i)$ for $i \in [M]$. The agents communicate via a central server in communication rounds $n = 1, 2, \ldots, N$, and upload their locally computed control gain matrices $\{K_n^{(i)}\}_{i \in [M]}$ in each round $n$. The server averages the agents' gain matrices to compute a global control policy $K_{n+1}$ that is broadcast to all agents to initiate the $(n+1)$-th round.

model than if they were to learn alone. In the RL setting, Federated Reinforcement Learning (FRL) aims to learn a common value function (Wang et al., 2024b) or produce a better policy from multiple RL agents interacting with similar environments. In the survey paper by Qi et al. (2021), FRL has empirically shown great success in reducing the sample complexity for autonomous driving (Liang et al., 2022), IoT devices (Lim et al., 2020), and resource management in networking (Yu et al., 2020).

Lately, there has been a lot of interest in applying RL techniques to classical control problems such as the linear quadratic regulator (LQR) problem (Anderson & Moore, 2007). In the standard control setting, the dynamical model of the system is known and one seeks to obtain a controller that stabilizes the closed-loop system and provides optimal performance. RL approaches such as policy gradient (Williams, 1992; Sutton et al., 1999) (which we pursue here) differ in that they are "model-free", i.e., a control policy is obtained despite not having access to the model of the dynamics. Despite the lack of convexity in even simple problems, policy gradient (PG) methods have been shown to be globally convergent for certain structured settings such as the LQR problem (Fazel et al., 2018). While this is promising, a major challenge in applying PG methods is that in general, one does not have access to *exact deterministic* policy gradients. Instead, one relies on estimating such gradients via sampling based approaches. This typically leads to noisy gradients that can suffer from high variance. As such, reducing the variance in PG estimates to achieve "good performance" may end up requiring several samples.

**Motivation.** The main premise of this paper is to draw on ideas from the FL literature to alleviate the high sample-complexity burden of PG methods (Agarwal et al., 2019; Wang et al., 2019; Liu et al., 2020), with the focus being on model-free control. As a motivating example, consider a fleet of identical robots produced by the same manufacturer. Each robot can collect data from its own dynamics and learn its own optimal policy using, for instance, PG methods. Since the fleet of robots shares similar dynamics, and more data can potentially lead to improved policy performance (via more accurate PG estimates), it is natural to ask: *Can a robot accelerate the process of learning its own optimal policy by leveraging the data of the other robots in the fleet?* The answer is not as obvious as one might expect since in reality, it is unlikely that any two robots will have *exactly* the same underlying dynamics, i.e., *heterogeneity in system dynamics is inevitable.* The presence of such heterogeneity makes the question posed above both interesting and non-trivial. In particular, when the heterogeneity across agents' dynamics is large, leveraging data from other agents might degrade the performance of a single agent. Indeed, large heterogeneity may make it impossible to learn a common stabilizing policy[1]. Moreover, even when such a stabilizing policy exists, it may deviate from each agent's local optimal policy, rendering poor performance and discouraging participation in the FL

---

[1]See Section 5 for more details on the underlying intuition and necessity behind the low heterogeneity regime.

process. Thus, to understand whether more data[2] helps or hurts, it is crucial to characterize the effects of heterogeneity in the federated control setting.

With this aim in mind, we study a multi-agent *model-free* LQR problem based on policy gradient methods. Specifically, there are $M$ agents in our setup, each with its own *distinct yet similar* linear time-invariant (LTI) dynamics. Inspired by the typical objective in FL, our goal is to find a common policy which can minimize the average of the LQR costs across agents.[3] In particular, we address the following questions.

**Q1.** *Is this common policy stabilizing for all the systems? If so, under what conditions?*

**Q2.** *How far is the learned common policy from each agent's locally optimal policy?*

**Q3.** *Can an agent use the common policy as an initial guess to fine-tune and learn its own optimal policy faster (i.e., with fewer overall samples) than if it acted alone?*

**Challenges:** There are several challenges to answering the above questions. First, even for the single agent setting, the policy gradient-based LQR problem is non-convex, and requires a fairly intricate analysis (Fazel et al., 2018). Second, a key distinction relative to standard federated supervised learning stems from the need to guarantee *stability* at every averaging iteration – this problem is amplified in the heterogeneous multi-agent scenario we consider. It remains an open problem to design an algorithm ensuring that policies are simultaneously stabilizing for each distinct system. Third, to reduce the communication cost, FL algorithms rely on the agents performing multiple local update steps between successive communication rounds. When agents have non-identical loss functions, these local steps lead to a "client-drift" effect where each agent drifts towards its own local minimizer (Charles & Konečný, 2020a; 2021a). While several works in FL have investigated this phenomenon (Li et al., 2020; Khaled et al., 2019a; 2020; Li et al., 2019a; Karimireddy et al., 2020; Pathak & Wainwright, 2020a; Wang et al., 2020a; Acar et al., 2021; Gorbunov et al., 2021; Mitra et al., 2021; Mishchenko et al., 2022; Laguel et al., 2021), *the effect of "client-drift" on stability remains completely unexplored*. Unless accounted for, such drift effects can potentially lead to non-stabilizing controllers.

**Our Contributions:** In response to the above challenges, we propose a policy gradient method called `FedLQR` to solve the (model-based *and* model-free) federated LQR problem, and provide a rigorous finite-time analysis of its performance that accounts for the interplay between system heterogeneity, multiple local steps, client-drift effects, and stability. Our specific contributions in this regard are as follows.

• **Iterative stability guarantees.** We show via a careful inductive argument that under suitable requirements on the level of heterogeneity across systems, the learning rate schedule can be designed to ensure that `FedLQR` provides a stabilizing controller at every iteration for *all* systems. Theorem 1 provides a proof in the model-based setting, and Theorem 2 provides the model-free result.

• **Bounded policy gradient heterogeneity in the LQR problem.** We prove in Lemma 3 that, for each pair of agents $i, j \in [M]$, the policy gradient direction (in the model-based setting) of agent $i$ is close to that of agent $j$, if their dynamics are similar (i.e., Definition 1). *This is the first result to observe and characterize this bounded gradient heterogeneity phenomenon in the multi-agent LQR setting.*

• **Quantifying the gap between `FedLQR`'s output and each system's optimal policy.** Building on Lemma 3, we prove that when the agents' dynamics are similar, the common policy returned by `FedLQR` is close to each agent's optimal policy; see Theorem 1. In other words, we can leverage the federated formulation to help each agent find its own optimal policy up to some accuracy depending on the level of heterogeneity. Moreover, we prove that `FedLQR` finds a controller that is close to the average's cost optimal solution up to a heterogeneity bias (Corollary 1). Our work is the first to provide a result of this flavor.

• **Linear speedup.** As our main contribution, we prove that in the model-free setting, `FedLQR` converges to a solution that is in a neighborhood of each agent's optimal policy, *using $M$-times fewer samples* relative to when each agent just uses its own data (see Theorem 2). The radius of this neighborhood captures the level of heterogeneity across the agents' dynamics. *The key implication of this result is that in a low-*

---

[2]In accordance with both FL & FRL frameworks, the agents in our problem do not exchange their private data (e.g., rewards, states, etc.). Instead, each agent only transmits its policy gradient.

[3]While one can certainly consider other FL objective functions, we study the average LQR cost formulation here since a comprehensive analysis of this setting is missing in the current literature.

*heterogeneity regime, `FedLQR` (in the model-free setting) reduces the sample-complexity by a factor of M w.r.t. the centralized setting (Fazel et al., 2018; Malik et al., 2019), highlighting the benefit of collaboration.*[4] Simply put, `FedLQR` enables each agent to *quickly* find an *approximate* locally optimal policy; as in standard FL (Collins et al., 2022), the agent can *fine-tune* this policy based on its own data.

In summary, we provide a new theoretical framework that quantitatively *characterizes the interplay between the price of heterogeneity and the benefit of collaboration* for model-free control.

## 2 Related Work

There has been a line of work that explores various RL algorithms for solving the model-free LQR problem (Fazel et al., 2018; Malik et al., 2019; Hambly et al., 2021; Mohammadi et al., 2021; Gravell et al., 2020; Jin et al., 2020; Ju et al., 2022). However, their analysis is limited to the single-agent setting. Most recently, Ren et al. (2020) solve the model-free LQR tracking problem in a federated manner and achieve a linear convergence speedup with respect to the number of agents. However, they consider a simplified setting where all agents follow the *same* dynamics. As such, the stability analysis of Ren et al. (2020) follow from arguments for the centralized setting. In sharp contrast, to establish the linear speedup for `FedLQR`, we need to address the key technical challenges arising from the effect of heterogeneity and local steps on the stability of distinct systems. This requires new analysis tools that we develop. For related work on multi-agent RL (that do not specifically look at the control setting) we point the reader to Lin et al. (2021); Zhang et al. (2021) and the references therein. Below we highlight the relevant work related to our problem setting.

- **Federated Learning (FL):** In this work, we employ the federated learning (FL) paradigm to facilitate collaborative learning among systems without the need to share raw data with other participants or a server (Konečný et al., 2016a; McMahan et al., 2017; Konečný et al., 2016b; Bonawitz et al., 2019). Despite FL being a relatively recent approach, it has already gained significant attention and boasts a wealth of literature.

  Federated averaging (`FedAvg`) stands as the pioneering and most widely adopted algorithm in FL. Originally proposed in McMahan et al. (2017), `FedAvg` has demonstrated its effectiveness in homogeneous settings (Stich, 2018; Wang & Joshi, 2021; Spiridonoff et al., 2020; Reisizadeh et al., 2020; Haddadpour et al., 2019) where all participating clients aim to minimize the same objective function. However, ensuring convergence guarantees for `FedAvg` becomes notably more challenging in the presence of heterogeneity (Khaled et al., 2019b; 2020; Haddadpour & Mahdavi, 2019; Li et al., 2019b), thus necessitating additional assumptions on gradient and Hessian dissimilarity bounds (Li et al., 2019b; Li & Orabona, 2019; Khaled et al., 2019b; Karimireddy et al., 2020). This difficulty arises primarily due to a "client-drift" effect, which is inherent to the `FedAvg` algorithm and has a detrimental impact on its convergence performance (Charles & Konečný, 2020b; 2021b). As a result of the challenges posed by `FedAvg`, several alternative algorithms have been proposed to address its limitations. Notable examples include `FedProx` (Li et al., 2020), `Scaffold` (Karimireddy et al., 2020), `FedSplit` (Pathak & Wainwright, 2020b), `FedDR` (Tran Dinh et al., 2021), `FedADMM` (Wang et al., 2022a), `FedLin` (Mitra et al., 2021), and `S-Local-SVRG` (Gorbunov et al., 2021). Each of them introduces unique techniques and modifications to the original `FedAvg` algorithm, aiming to enhance convergence guarantees while handling communication cost concerns, statistical heterogeneity, client dropout, and sample complexity more effectively.

  Applying federated learning (FL) to control systems introduces a novel research direction that comes with its own set of challenges. Control systems exhibit unique characteristics, such as non-iid and non-isotropic data, as well as system instability, which arise due to the dynamic nature of the systems. These characteristics pose specific challenges when attempting to leverage data from multiple systems for tasks such as system identification (Wang et al., 2022b) or control synthesis (Ren et al., 2020).

  Although Ren et al. (2020) address the model-free LQR tracking problem in a multi-agent setting, it focuses on a significantly simpler scenario where all agents follow identical dynamics (i.e., homo-

---

[4]Throughout this paper, we use the terms "centralized" and "single-agent" interchangeably.

geneous). In contrast, our work introduces new analysis techniques to achieve linear speedup when dealing with heterogeneous systems and multiple local updates per communication round.[5]

- **Policy Gradient (PG):** The policy gradient (PG) approach is a fundamental component of the success of reinforcement learning (RL) and plays a crucial role in policy optimization (PO). This approach directly optimizes the policy to improve system-level performances through gradient ascent steps. The concept of policy optimization has been influential in RL (Sutton et al., 1999) with some well-known algorithms such as `REINFORCE` (Williams, 1992), trust-region policy optimization `TRPO` (Schulman et al., 2015), actor-critic methods (Konda & Tsitsiklis, 1999), and proximal policy optimization `PPO` (Schulman et al., 2017). We highlight an important difference between standard MDP models and control models in RL. In control, one requires the policy to provide closed-loop stability, i.e., all trajectories of the system must converge for a given policy. In contrast, there are no analogous stability concerns in finite state-action MDP settings studied in RL.

  The extensive body of literature on policy optimization for reinforcement learning (RL) and its adaptability to the model-free setting paves the way for leveraging policy gradient methods in the pursuit of learning optimal control policies for classical control problems (Hu et al., 2022; Perdomo et al., 2021). Despite the non-convex nature of the formulation involved in policy gradient methods, recent work (Fazel et al., 2018; Malik et al., 2019; Hambly et al., 2021; Mohammadi et al., 2021; Gravell et al., 2020; Jin et al., 2020; Ju et al., 2022; Perdomo et al., 2021; Lamperski, 2020; Toso et al., 2024b; 2025) has demonstrated global convergence in solving the model-free LQR problem via policy gradient methods in different settings. This global convergence is achieved due to certain properties of the quadratic cost function inherent in the LQR problem (e.g., smoothness and gradient dominance) as established in Fazel et al. (2018). In contrast to the aforementioned work, which exclusively focus on the centralized control setting, our paper offers convergence guarantees for the multi-agent setting. In this context, each agent follows different dynamics, thereby distinguishing it from the simpler scenario proposed in Ren et al. (2020).

- **Federated Reinforcement Learning (FRL):** The flexibility of policy gradient methods in the model-free RL setting has paved the way for a relatively recent research direction known as federated reinforcement learning (FRL), which aims to address practical implementation challenges of RL through the use of federated learning (Qi et al., 2021). FRL focuses on learning a common value function (Wang et al., 2024b; Fabbro et al., 2023) or improving the policy by leveraging multiple RL agents interacting with similar environments (Woo et al., 2023; Wang et al., 2024a). The empirical evidence presented in the survey paper (Qi et al., 2021) demonstrates the significant success of FRL in reducing sample complexity across various applications such as autonomous driving (Liang et al., 2022), IoT devices (Lim et al., 2020), resource management in networking (Yu et al., 2020), and communication efficiency (Gatsis, 2022). The FRL literature can be broadly grouped into two categories: (i) The *homogeneous* setting, where all agents interact with the same MDP (Khodadadian et al., 2022; Woo et al., 2023; Lan et al., 2023; Shen et al., 2023; Liu & Olshevsky, 2023; Fabbro et al., 2023; Tian et al., 2024; Salgia & Chi, 2024; Dal Fabbro et al., 2025), and (ii) the *heterogeneous* setting, where the agents' MDPs can potentially differ in their reward functions and/or probability transition kernels (Jin et al., 2022; Xie & Song, 2023; Wang et al., 2024b; Zhang et al., 2024; Wang et al., 2024a; Zhu et al., 2024; Mangold et al., 2024; Labbi et al., 2024; 2025). In particular, federated variants of policy gradient methods are analyzed in Xie & Song (2023); Lan et al. (2023); Wang et al. (2024a); Zhu et al. (2024); Labbi et al. (2025); Zhang et al. (2025). It is important to note that all the papers in FRL discussed above consider MDPs with finite state and action spaces, where there is no notion of stability. In sharp contrast, our work considers a federated control problem involving dynamical systems with continuous state-action spaces, where we need to tackle the challenge of finding a common, stabilizing policy for all the agents at every iteration.

---

[5]In the time since this work was first made available online it has been used to develop new results on multi-task LQR design, including studies on meta-learning linear quadratic regulators (Toso et al., 2024c; Pan et al., 2025; Aravind et al., 2024), asynchronous federated LQR design (Toso et al., 2024a; Zhao et al., 2025), learning LQR controllers from proxy systems (Ye et al., 2024), and domain randomization for the LQR problem (Fujinami et al., 2025).

## 3    Notation

Given a set of matrices $\{S^{(i)}\}_{i=1}^{M}$, we denote $||S||_{\max} := \max_i ||S^{(i)}||$, and $||S||_{\min} := \min_i ||S^{(i)}||$. All vector norms are Euclidean and matrix norms are spectral, unless otherwise stated.

## 4    Problem Setup

Classical control approaches aim to design optimal controllers from a well-defined dynamical system model. The model-based LQR problem is well understood and admits a convex solution. In this work, we consider the LQR problem but in the *model-free setting*. Moreover, we consider a *federated model-free LQR problem* in which there are $M$ agents, each with their own distinct but "similar" dynamics. Our goal is to collaboratively learn an optimal controller that minimizes an average quadratic cost. We seek to characterize the optimality of our solution as a function of the "difference" across the agent's dynamics. In what follows, we formally describe our problem of interest.

**Federated LQR:** Consider a system with $M$ agents. Associated with each agent is a linear time-invariant (LTI) dynamical system of the form

$$x_{t+1}^{(i)} = A^{(i)} x_t^{(i)} + B^{(i)} u_t^{(i)}, \quad x_0^{(i)} \sim \mathcal{D}, \quad \text{for all } i \in [M],$$

where $[M] := 1, \ldots, M$, and $A^{(i)} \in \mathbb{R}^{n_x \times n_x}$, $B^{(i)} \in \mathbb{R}^{n_x \times n_u}$. We assume each initial state $x_0^{(i)}$ is randomly generated from the same distribution $\mathcal{D}$. In the single-agent setting, the optimal LQR control policy is known to be linear and static. We denote the policy by $u_t^{(i)} = -K_i^* x_t^{(i)}$, where for each agent, $K_i^\star$ solves

$$K_i^* = \arg \min_K \left\{ C^{(i)}(K) := \mathbb{E} \left[ \sum_{t=0}^{\infty} x_t^{(i)\top} Q x_t^{(i)} + u_t^{(i)\top} R u_t^{(i)} \right] \right\}$$
$$\text{s.t.} \quad x_{t+1}^{(i)} = A^{(i)} x_t^{(i)} + B^{(i)} u_t^{(i)}, \ u_t^{(i)} = -K x_t^{(i)}, \ x_0^{(i)} \sim \mathcal{D}, \tag{1}$$

where $Q \in \mathbb{R}^{n_x \times n_x}$ and $R \in \mathbb{R}^{n_u \times n_u}$ are known positive definite matrices. In our federated setting, the objective is to find an optimal common policy $\{u_t\}_{t=0}^{\infty}$ to minimize the average cost of all the agents $C_{\text{avg}}(K) := \frac{1}{M} \sum_{i=1}^{M} C^{(i)}(K)$ *without knowledge of the system dynamics*, i.e., $(A^{(i)}, B^{(i)})$. As mentioned above, classical results (Anderson & Moore, 2007) from optimal control theory show that, given the system matrices $A^{(i)}$, $B^{(i)}$, $Q$ and $R$, the optimal policy can be written as a linear function of the current state. Thus, we consider a common policy of the form $u_t^{(i)} = -K x_t^{(i)}$. The objective of the federated LQR problem can be written as:

$$K^* = \arg \min_K \left\{ C_{\text{avg}}(K) := \frac{1}{M} \sum_{i=1}^{M} \mathbb{E} \left[ \sum_{t=0}^{\infty} x_t^{(i)\top} Q x_t^{(i)} + u_t^{(i)\top} R u_t^{(i)} \right] \right\}$$
$$\text{s.t.} \quad x_{t+1}^{(i)} = A^{(i)} x_t^{(i)} + B^{(i)} u_t^{(i)}, \quad u_t^{(i)} = -K x_t^{(i)}, x_0^{(i)} \sim \mathcal{D}. \tag{2}$$

The rationale for finding $K^*$ is as follows. Intuitively, when all agents have similar dynamics, $K^*$ will be close to each $K_i^*$. Thus, $K^*$ will serve to provide a good common initial guess from which each agent $i$ can then fine-tune/personalize (using only its own data) to converge *exactly* to its own locally optimal controller $K_i^*$. The key here is that the initial guess $K^*$ can be obtained *quickly* by using the *collective data* of all the agents. We will formalize this intuition in Theorem 2.

We make the standard assumption that for each agent, $(A^{(i)}, B^{(i)})$ is stabilizable. In addition, we make the following assumption on the distribution of the initial state:

**Assumption 1.** *Given $\mu > 0$, for any $i \in [M]$, the initial state $x_0^{(i)} \sim \mathcal{D}$ and distribution $\mathcal{D}$ satisfy*

$$\mathbb{E}_{x_0^{(i)} \sim \mathcal{D}}[x_0^{(i)}] = 0, \quad \mathbb{E}_{x_0^{(i)} \sim \mathcal{D}}[x_0^{(i)} x_0^{(i)\top}] \succ \mu \mathbb{I}_{d_x}, \text{ and } \|x_0^{(i)}\| \leq H \text{ almost surely.}$$

We quantify the heterogeneity in the agent's dynamics through the following definition:

**Definition 1.** *(Bounded system heterogeneity) There exist positive constants $\epsilon_1$ and $\epsilon_2$ such that*

$$\max_{i,j\in[M]}\|A^{(i)} - A^{(j)}\| \le \epsilon_1, \ \ and \ \max_{i,j\in[M]}\|B^{(i)} - B^{(j)}\| \le \epsilon_2.$$

We assume that $\epsilon_1$ and $\epsilon_2$ are finite. Similar bounded heterogeneity assumptions are commonly made in FL (Karimireddy et al., 2020; Khaled et al., 2020; Reddi et al., 2020). However, unlike typical FL works where one directly imposes heterogeneity assumptions on the agents' gradients, in our setting, we need to carefully characterize how heterogeneity in the system parameters $(A^{(i)}, B^{(i)})$ translates to differences in the policy gradients; see Lemma 3.

**On the meaning of system similarity:** The notion of heterogeneity introduced in Definition 1 measures similarity in terms of parameter deviations between system matrices. It is important to emphasize that similarity in the system parameters $(A^{(i)}, B^{(i)})$ does not in general imply similar closed-loop behavior (see Propositions 1 and 2). Even a small perturbation in system parameters can produce significant differences in the set of stabilizing controllers, and may not guarantee the existence of a common stabilizing controller across agents. Recent work (Stamouli et al., 2025) has proposed a refined similarity measure based on bisimulation-type metrics that quantify *closed-loop* heterogeneity rather than parameter differences, and the main results of our paper extend to their setting.

Before providing our solution to the federated LQR problem, we first recap existing results on model-free LQR in the single-agent setting.

**Single-agent setting:** When there is only one agent, i.e., $M = 1$, let us denote the system matrix as $(A, B)$ and likewise, the cost functional by $C(K)$. If $(A, B)$ is known, the optimal controller $K^*$ can be computed by solving the discrete-time algebraic Riccati equation (ARE) (Anderson & Moore, 2007).

Strikingly, Fazel et al. (2018) show that policy gradient methods can find the globally optimal LQR policy $K^*$ despite the non-convexity of the problem. The policy gradient of the LQR problem can be expressed as:

$$\nabla C(K) = 2E_K \Sigma_K = 2\left(\left(R + B^\top P_K B\right)K - B^\top P_K A\right)\Sigma_K,$$

where $P_K$ is the positive definite solution to the Lyapunov equation: $P_K = Q + K^\top R K + (A - BK)^\top P_K (A - BK)$, $E_K := \left(R + B^\top P_K B\right)K - B^\top P_K A$, and $\Sigma_K := \mathbb{E}_{x_0 \sim \mathcal{D}}[\sum_{t=0}^\infty x_t x_t^\top]$. The policy gradient method $K \leftarrow K - \eta \nabla C(K)$ will find the global optimal LQR policy, i.e., $K \to K^*$, provided that $\mathbb{E}_{x_0 \sim \mathcal{D}}[x_0 x_0^\top]$ is full rank and an initial stabilizing policy is used. When the model is unknown, the analysis technique employed by Fazel et al. (2018) is to construct near-exact gradient estimates from reward samples and show that the sample complexity of such a method is bounded polynomially in the parameters of the problem.

In contrast to the single-agent setting, the heterogeneous, multi-agent scenario we consider here is considerably more difficult to analyze. First, designing an algorithm satisfying the iterative stability guarantees becomes a complex task. Second, since each agent in the system has its own unique dynamics and gradient estimates, it can be difficult to aggregate these directions in a manner that ensures the updating direction moves toward the average optimal policy $K^*$. Nonetheless, in the sequel, we will overcome these challenges and provide a finite-time analysis of `FedLQR`.

## 5 Necessity of the Low Heterogeneity Requirement

In our main theorems, we require certain bounds on the parameters $\epsilon_1$ and $\epsilon_2$ that define the heterogeneity of the $M$ dynamical systems we work with. Here, we point out that, unlike standard federated learning settings, these bounds are *necessary* for convergence. From a control and dynamical systems viewpoint, these bounds are perhaps intuitive: if the systems are too different, then there is no reason to believe there exists a common stabilizing controller, i.e., there is no solution to the problem (2). In what follows, we will formalize this point. To do so, let us define an "instance" of our FedLQR problem via a parameter $M$ that characterizes the number of agents/systems and the set of corresponding system matrices $\{A^{(i)}, B^{(i)}\}_{i\in[M]}$.[6]

---

[6]Although technically the cost matrices $Q$ and $R$ are also part of a FedLQR problem formulation, they are not needed to establish the necessity of a low-heterogeneity requirement. As such, we do not include them here.

We now prove a couple of simple impossibility results. Our first result shows that even when the input matrices are identical across agents, heterogeneity in the state transition matrices can lead to the non-existence of simultaneously stabilizing controllers, thereby rendering the FedLQR problem infeasible.

**Proposition 1.** *There exists an instance of the FedLQR problem with $M = 2$ and $\epsilon_2 = 0$, such that if $\epsilon_1 > 2$, then it is impossible to find a common state-feedback gain $K$ that simultaneously stabilizes both systems.*

**Proof:** Consider an instance with just two scalar systems defined by:

$$x_{t+1}^{(1)} = \alpha x_t^{(1)} + u_t^{(1)} \quad \text{and} \quad x_{t+1}^{(2)} = -\alpha x_t^{(2)} + u_t^{(2)},$$

for some $\alpha > 0$. By simple inspection, note that in this case $\epsilon_1 = 2\alpha$ and $\epsilon_2 = 0$. Thus, $\epsilon_1 > 2 \Rightarrow \alpha > 1$. Now for a controller $u_t^{(i)} = -kx_t^{(i)}$ to stabilize both systems, the spectral radius conditions are $|\alpha - k| < 1$ and $|\alpha + k| < 1$. Trivially, there exists no gain $k$ that satisfies both these requirements when $\alpha > 1$.

To complement the above result, we now show that the effect of heterogeneity is not just limited to the state transition matrices. In particular, even when the state transition matrices are identical across agents, (arbitrarily small) heterogeneity in the input matrices can also lead to the non-existence of simultaneously stabilizing control gains. We formalize this below.

**Proposition 2.** *There exists an instance of the FedLQR problem with $M = 2$ and $\epsilon_1 = 0$, such that if $\epsilon_2 > 0$, then it is impossible to find a common state-feedback gain $K$ that simultaneously stabilizes both systems.*

**Proof:** Consider an instance with two scalar systems defined by:

$$x_{t+1}^{(1)} = x_t^{(1)} + \beta u_t^{(1)} \quad \text{and} \quad x_{t+1}^{(2)} = x_t^{(2)} - \beta u_t^{(2)},$$

for some $\beta$. By simple inspection, note that in this case $\epsilon_1 = 0$ and $\epsilon_2 = 2\beta$. Thus, $\epsilon_2 > 0 \Rightarrow \beta > 0$. Now for a controller $u_t^{(i)} = -kx_t^{(i)}$ to stabilize both systems, the spectral radius conditions are $|1 - \beta k| < 1$ and $|1 + \beta k| < 1$. Trivially, there exists no gain $k$ that satisfies both these requirements when $\beta > 0$. This concludes the proof.

The above example suggests that in certain settings, we can tolerate no heterogeneity whatsoever in the input matrices. More generally, the main take-home message from this section is that the requirement of a "low-heterogeneity regime" is *fundamental* to the problem and not merely an artifact of our analysis. We formally define a low heterogeneity regime in the sequel.

## 6 The FedLQR algorithm

In this section, we introduce our algorithm FedLQR, formally described by Algorithm 1, to solve for $K^*$ in (2). First, we impose the following assumption regarding the algorithm's initial condition $K_0$:

**Assumption 2.** *We can access an initial stabilizing controller, $K_0$, which stabilizes all systems $\{(A^{(i)}, B^{(i)})\}_{i=1}^{M}$, i.e., the spectral radius $\rho(A^{(i)} - B^{(i)} K_0) < 1$ holds for all $i \in [M]$.*

**Algorithm description:** At a high level, FedLQR follows the standard FL algorithmic template: a server first initializes a global policy, $K_0$, which it sends to the agents. Each agent proceeds to execute multiple PG updates using their local data. Once the local training is finished, agents transmit their model update to the server. The server aggregates the models and broadcasts an averaged model to the clients. The process repeats until a termination criterion is met. Prototypical FL algorithms that adhere to this structure include, for instance, FedAvg (Khaled et al., 2020) and FedProx (Li et al., 2020).

With this template in mind, we now dive into the details: FedLQR initializes the server and all agents with $K_{0,0}^{(i)} = K_0$ – a controller that stabilizes all agent's dynamics.[7] In each round $n$, starting from a common global policy $K_n$, each agent $i$ independently samples $n_s$ trajectories from its own system at each local iteration $l$ and performs approximate policy gradient updates using the zeroth-order optimization procedure (Fazel

---

[7]To establish notation, $K_{n,l}^{(i)}$ refers to the local controller associated to agent $i$ after $n$ rounds of averaging, and after $l^{\text{th}}$ local iteration of policy gradient updates.

et al., 2018) which we denote ZO; see line 7. For clarity, we present the explicit steps of using the zeroth-order method to estimate the true gradient in Algorithm 2, which will be discussed shortly. Between every communication round, each agent updates their local policy $L$ times. Such an $L$ is chosen to balance between the benefit of information sharing and the cost of communication. After $L$ local iterations, each agent $i$ uploads its local policy difference $\Delta_n^{(i)}$ (line 10) to the server. Once all differences are received, the server averages these differences $\{\Delta_n^{(i)}\}$ (line 12) to construct a new global policy $K_{n+1}$. The whole process is repeated $N$ times.

---

**Algorithm 1** Model-free Federated Policy Learning for the LQR (`FedLQR`)

---

1: **Input:** initial policy $K_0$, local step-size $\eta_l$ and global step-size $\eta_g$.
2: **Initialize** the server with $K_0$ and $\eta_g$
3: **for** $n = 0, \ldots, N-1$ **do**
4:     **for each system** $i \in [M]$ **do**
5:         **for** $l = 0, \cdots, L-1$ **do**
6:             Agent $i$ initializes $K_{n,0}^{(i)} = K_n$
7:             Agent $i$ estimates $\widehat{\nabla} C^{(i)}(K_{n,l}^{(i)}) = \texttt{ZO}(K_{n,l}^{(i)}, i)$ and updates local policy as
8:                 $K_{n,l+1}^{(i)} = K_{n,l}^{(i)} - \eta_l \widehat{\nabla} C^{(i)}(K_{n,l}^{(i)})$
9:         **end for**
10:         send $\Delta_n^{(i)} = K_{n,L}^{(i)} - K_n$ back to the server
11:     **end for**
12:     Server computes and broadcasts global model $K_{n+1} = K_n + \frac{\eta_g}{M} \sum_{i=1}^{M} \Delta_n^{(i)}$
13: **end for**

---

Zeroth-order optimization (Conn et al., 2009; Nesterov & Spokoiny, 2017) provides a method of optimization that only requires oracle access to the function being optimized. Here, we briefly describe the details of our zeroth-order gradient estimation step[8] in Algorithm 2. To obtain a gradient estimate at a given policy $K$, we sample trajectories from the $i$-th system $n_s$ times. At each time $s$, we use the perturbed policy $\widehat{K}_s$ (line 3) and a randomly generated initial point $x_0 \sim \mathcal{D}$ to simulate the $i$-th closed-loop system for $\tau$ steps. Thus, we can approximately calculate the cost by adding the stage cost from the first $\tau$ time steps on this trajectory (line 4), and then estimating the gradient as in line 6.

**Discussion of Assumption 2:** Assumption 2 is commonly adopted in the LQR (Fazel et al., 2018; Dean et al., 2020; Agarwal et al., 2019; Ren et al., 2020) and robust control literature (Boyd et al., 1994; Lu et al., 1996; Doyle et al., 1989). In addition, there exist efficient ways to find such a stabilizing policy $K_0$; (Boyd et al., 1994) that addresses the model-based setting, while (Lamperski, 2020; Perdomo et al., 2021; Zhao et al., 2022; Toso et al., 2025) each addresses the problem in the single-agent model-free setting. Moreover, it is well-known that the sample complexity of finding an initial stabilizing policy only adds a logarithmic factor to that for solving the single-agent LQR problem (Zhao et al., 2022). More recently, Fujinami et al. (2025) have shown that solving a sequence of discounted LQR problems using the *average* $C_{\text{avg}}(K)$ cost yields a common stabilizing controller for all systems under low heterogeneity conditions.

**Choice of the number of local updates:** The parameter $L$ controls the standard communication, drift trade-off encountered in federated optimization. Executing more local steps before aggregation reduces communication frequency, but at the cost of amplifying the deviation between the aggregated update and the true global gradient direction. Conversely, choosing a smaller $L$ keeps the local iterates closer to the global descent direction and improves agreement with the centralized update, albeit requiring more communication rounds. In our setting, this trade-off becomes particularly delicate because drift between local controllers directly affects stability. We refer the reader to Lemmata 10 and 11 for additional details on how the number of local updates enters our analysis.

**Challenges in `FedLQR` analysis:** Although `FedLQR` is similar in spirit to `FedAvg` (Li et al., 2019b; McMahan et al., 2017) (in the supervised learning setting), it is significantly more difficult to analyze the convergence of `FedLQR` for the following reasons.

---

[8]See Appendix A.5 for more details on zeroth-order optimization.

---

**Algorithm 2** Zeroth-order gradient estimation (`ZO`)

---

1: **Input:** $K$, number of trajectories $n_s$, trajectory length $\tau$, smoothing radius $r$, dimension $n_x$ and $n_u$, system index $i$.

2: **for** $s = 1, \ldots, n_s$ **do**

3:     Sample a policy $\widehat{K}_s = K + U_s$, with $U_s$ drawn uniformly at random over matrices whose (Frobenius) norm is $r$.

4:     Simulate the $i$-th system for $\tau$ steps starting from $x_0 \sim \mathcal{D}$ using policy $\widehat{K}_s$. Let $\widehat{C}_s$ be the empirical estimate: $\widehat{C}_s = \sum_{t=1}^{\tau} c_t$, where $c_t := x_t^\top \left( Q + \widehat{K}_s^\top R \widehat{K}_s \right) x_t$ on this trajectory.

5: **end for**

6: **Return** the estimate:    $\widehat{\nabla} C(K) = \frac{1}{n_s} \sum_{s=1}^{n_s} \frac{n_x n_u}{r^2} \widehat{C}_s U_s$.

---

- First, the problem we study is non-convex. Unlike most existing non-convex FL optimization results (Karimireddy et al., 2020) which only guarantee convergence to stationary points, our work investigates whether `FedLQR` can find a globally optimal policy.

- Second, standard convergence analyses in FL (McMahan et al., 2017; Karimireddy et al., 2020; Wang et al., 2020b; Li et al., 2020) rely on a "bounded gradient-heterogeneity" assumption. For the LQR problem, it is not clear a priori whether similar bounded policy gradient dissimilarity still holds. In fact, this is something we prove in Lemma 3.

- Third, the randomness in FL usually comes from only one source: the data obtained by each agent are drawn i.i.d. from some distribution; we call this sample randomness. However, in `FedLQR`, there are randomness in the initial condition, and randomness from the smoothing matrices that show up in the gradient estimation process outlined in Algorithm 2. To reason about these different forms of randomness (that are intricately coupled), we provide a careful martingale-based analysis.

- Finally, we need to determine whether the solution given by `FedLQR` is meaningful, i.e., to decide whether the policy generated at each (local and global) iteration will stabilize all the systems.

To tackle these difficulties, we first define a stability region in our setting comprising of $M$ heterogeneous systems as:

**Definition 2.** *(The stabilizing set) The stabilizing set is defined as $\mathcal{G}^0(\beta) := \cap_{i=1}^M \mathcal{G}^{(i)}(\beta)$ where*

$$\mathcal{G}^{(i)}(\beta) := \left\{ K : C^{(i)}(K) - C^{(i)}(K_i^*) \leq \beta \left( C^{(i)}(K_0) - C^{(i)}(K_i^*) \right) \right\}.$$

As in Malik et al. (2019), $\mathcal{G}^0(\beta)$ is defined as the intersection of sub-level sets containing points $K$ whose cost gap is at most $\beta$ times the initial cost gap for all systems. We drop the dependence on $\beta$ when it is clear from the analysis and context. It was shown in Hu et al. (2022) that this is a compact set. Each sub-level set corresponds to a cost gap to agent $i$'s optimal policy $K_i^*$, which is at most $\beta$ times the initial cost gap $C^{(i)}(K_0) - C^{(i)}(K_i^*)$. Note that $\beta$ can be any positive finite constant. Since any finite cost function indicates that $K$ is a stabilizing controller, we conclude that any $K \in \mathcal{G}^0(\beta)$ stabilizes all the systems. Following from Assumption 2, there exists a constant $\beta$ such that $\mathcal{G}^0(\beta)$ is nonempty. Moreover, it is worth remarking that the LQR cost function in the single-agent setting is *coercive*. That is, the cost acts as a barrier function, ensuring that the policy gradient update remains within the feasible stabilizing set $\mathcal{G}^{(i)}(\beta)$. By defining the stabilizing set $\mathcal{G}^0(\beta)$ as above, the cost function $C^{(i)}(K)$ retains its coerciveness on $\mathcal{G}^0(\beta)$ for the federated setting considered in this paper. We also note that for our per-iteration stability analysis in Lemma 10, $\beta$ is properly set and kept fixed over iterations.

In order to solve the federated LQR problem and provide convergence guarantees for `FedLQR`, we first need to recap some favorable properties of the LQR problem in the single-agent setting that enables PG to find the globally optimal policy.

# 7 Background on the centralized LQR using PG

In the single-agent setting, it was shown that policy gradient methods (i.e., model-free) can produce the global optimal policy despite the LQR problem being non-convex (Fazel et al., 2018). We summarize the properties that make this possible and which we also exploit in our analysis.

**Lemma 1.** *(Local Cost and Gradient Smoothness) Suppose $K'$ is such that $\|K' - K\| \leq h_\Delta(K) < \infty$. Then, the cost and gradient function satisfy:*

$$|C(K') - C(K)| \leq h_{cost}(K)\|K' - K\|,$$
$$\|\nabla C(K') - \nabla C(K)\| \leq h_{grad}(K)\|\Delta\| \ and \ \|\nabla C(K') - \nabla C(K)\|_F \leq h_{grad}(K)\|\Delta\|_F,$$

*respectively, where $h_\Delta(K)$, $h_{cost}(K)$ and $h_{grad}(K)$ are positive scalars depending on $C(K)$.*

**Lemma 2.** *(Gradient Domination) Let $K^*$ be an optimal policy. Then,*

$$C(K) - C(K^*) \leq \frac{\|\Sigma_{K^*}\|}{4\mu^2\sigma_{\min}(R)}\|\nabla C(K)\|_F^2$$

*holds for any stabilizing controller $K$, i.e., any $K$ satisfying the spectral radius $\rho(A - BK) < 1$.*

For simplicity, we skip the explicit expressions in these lemmas for $h_\Delta(K)$, $h_{cost}(K)$, and $h_{grad}(K)$ as functions of the parameters of the LQR problem. Interested readers are referred to the appendix for full details. With Definition 2 of the stabilizing set in hand, we can define the following quantities:

$$\bar{h}_{grad} := \sup_{K \in \mathcal{G}^0} h_{grad}(K), \ \bar{h}_{cost} := \sup_{K \in \mathcal{G}^0} h_{cost}(K), \ \text{and} \ \underline{h}_\Delta := \inf_{K \in \mathcal{G}^0} h_\Delta(K).$$

With these quantities, we can transform the *local* properties of the LQR problem discussed in Lemmas 1–2 into properties that hold over the *global* stabilizing set $\mathcal{G}^0$. For convenience, we use letters with(under and over) bars such as $\bar{h}_{grad}$ to denote the global parameters. We are now ready to present our main results of `FedLQR` in the next section.

# 8 Main results

To analyze the performance of `FedLQR` in the model-free case, we first need to examine its behavior in the model-based case. Although this is not our end goal, these results are of independent interest.

## 8.1 Model-based setting

When $(A^{(i)}, B^{(i)})$ are available, exact gradients can be computed, and so the `ZO` estimation scheme is no longer needed. In this case, the updating rule of `FedLQR` (Algorithm 1, line 12) reduces to

$$K_{n+1} = K_n - \frac{\eta}{ML}\sum_{i=1}^{M}\sum_{l=0}^{L-1}\nabla C(K_{n,l}^{(i)}),$$

where $\eta := L\eta_g\eta_l$. Intuitively, if two systems are similar, i.e., satisfy Assumption 1, their exact policy gradient directions should not differ too much. We formalize this intuition as follows.

**Lemma 3.** *(**Policy gradient heterogeneity**) For any $i, j \in [M]$ and $K \in \mathcal{G}^0$, we have:*

$$||\nabla C^{(i)}(K) - \nabla C^{(j)}(K)|| \leq \epsilon_1 h_{het}^1(K) + \epsilon_2 h_{het}^2(K), \tag{3}$$

*where $h_{het}^1(K)$ and $h_{het}^2(K)$ are positive bounded functions depending on the parameters of the LQR problem.[9]*

---

[9]For simplicity, we write $h_{het}^1, h_{het}^2$ as a function of only $K$ since only $K$ changes during the iterations while other parameters remain fixed.

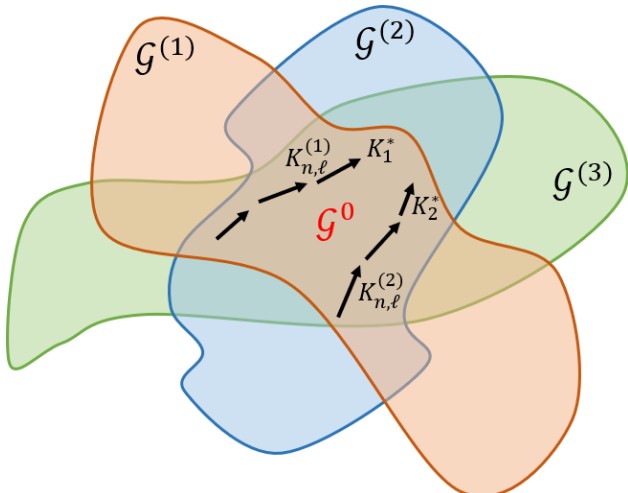

Figure 2: Illustration of the dynamics of the `FedLQR` algorithm for a setting with $M = 3$ agents. Recall from Definition 2 that $\mathcal{G}^{(i)}$ represents the set of stabilizing policies for the $i$-th agent, and $\mathcal{G}^0 = \bigcap_{i=1}^3 \mathcal{G}^{(i)}$ is the common stabilizing set. In a low-heterogeneity regime, the agents' policy gradients are "close" to one another (as formalized by Lemma 3), causing the sequence/trajectory of policies (controllers) generated by `FedLQR` to also evolve close to each other during the course of the algorithm. These trajectories $\{K_{n,l}^{(i)}\}_{i\in[3]}$ are illustrated by the black arrows. Crucially, initializing these trajectories within $\mathcal{G}^0$ causes them to always remain within $\mathcal{G}^0$, i.e., the set $\mathcal{G}^0$ is *forward-invariant* for the trajectories generated by `FedLQR` - a fact established rigorously in Lemma 10. This forward invariance property is the key to showing that the policies generated by `FedLQR` remain stabilizing for all systems.

By Lemma 3 (the proof of which is deferred to Appendix A.4), if $K$ belongs to a bounded set, the right-hand side of Eq. (3) is of the order $\mathcal{O}(\epsilon_1 + \epsilon_2)$. In other words, the exact gradient direction of agent $i$ can be well-approximated by the gradient direction of agent $j$ when the heterogeneity constants $\epsilon_1$ and $\epsilon_2$ are small. This justifies why it is beneficial to use other agents' data under the low-heterogeneity setting. Moreover, we can immediately conclude that the exact update direction of our `FedLQR` algorithm is also close to each agent's policy gradient direction based on Lemma 3. This fact is crucial for analyzing the convergence of `FedLQR` since we can map the convergence of `FedLQR` to that of the centralized LQR problem (with only one agent). However, Lemma 3 alone is not sufficient to provide the final guarantees since we still need to consider the impact of multiple local updates i.e., $L > 1$ (Algorithm 1, line 5) and stability concerns with heterogeneous systems – i.e., is $\rho(A^{(i)} - B^{(i)} K_N) < 1$ for all agents $i \in [M]$ *and* all $n \in [N]$? Nevertheless, by overcoming these difficulties, we establish the convergence of `FedLQR` in the model-based setting as follows:

**Theorem 1.** *(Optimality in each agent's cost function)* *When the heterogeneity level satisfies[10] $(\epsilon_1 \bar{h}_{het}^1 + \epsilon_2 \bar{h}_{het}^2)^2 \leq \bar{h}_{het}^3$ , there exist constant step-sizes $\eta_g$ and $\eta_l$ such that `FedLQR` enjoys the following performance guarantees over $N$ rounds:*

$$C^{(i)}(K_N) - C^{(i)}(K_i^*) \leq \left(1 - \frac{\eta\mu^2\sigma_{\min}(R)}{\left\|\Sigma_{K_i^*}\right\|}\right)^N \left(C^{(i)}(K_0) - C^{(i)}(K_i^*)\right) + c_{uni,1} \times \mathcal{B}(\epsilon_1, \epsilon_2),$$

*with $\mathcal{B}(\epsilon_1, \epsilon_2) := \frac{\upsilon\left\|\Sigma_{K_i^*}\right\|}{4\mu^2\sigma_{\min}(R)}(\epsilon_1 h_{het}^1 + \epsilon_2 h_{het}^2)^2$, where $\bar{h}_{het}^{1,2} := \sup_{K\in\mathcal{G}^0} h_{het}^{1,2}(K)$, $\upsilon := \min\{n_x, n_u\}$, and $c_{uni,1}$ is a universal constant. Moreover, we have $K_n \in \mathcal{G}^0$ for all $n = 0, \cdots, N$.*

**Main Takeaways:** Theorem 1 reveals that the output $K_n$ of `FedLQR` can stabilize all $M$ systems at each round $n$. However, `FedLQR` can only converge to a ball of radius $\mathcal{B}(\epsilon_1, \epsilon_2)$ around each system's optimal controller $K_i^*$, regardless of the choice of the step-sizes. The term $\mathcal{B}(\epsilon_1, \epsilon_2)$ captures the effect of heterogeneity

---

[10]The notation $\bar{h}_{het}^3$ is a positive scalar depending on the parameters of the LQR problem; see Appendix A.4.2 for full details.

and becomes zero when each agent follows the same system dynamics, i.e., $\epsilon_1 = \epsilon_2 = 0$. When there is no heterogeneity, the convergence rate matches the rate of the centralized setting (Fazel et al., 2018) up to a constant factor. But, since there is no noise introduced by the zeroth-order gradient estimate, there is no expectation of obtaining a benefit from collaboration. Nonetheless, understanding the model-based setting provides valuable insights for exploring the model-free setting. The proof of Theorem 1 is given in Appendix A.4.2. We are now ready to provide the convergence guarantees for `FedLQR` with respect to solving Eq. (2) – the average cost.

**Corollary 1.** *(Optimality in average cost function)* *When the heterogeneity level satisfies*[11] $(\epsilon_1 \bar{h}_{het}^1 + \epsilon_2 \bar{h}_{het}^2)^2 \leq \bar{h}_{het}^3$, *after $N$ rounds, `FedLQR` enjoys the following optimality gap in average cost function across all $M$ agents:*

$$C_{avg}(K_N) - C_{avg}(K^*) \leq \left(1 - \frac{\eta\mu^2\sigma_{\min}(R)}{\max_i \|\Sigma_{K_i^*}\|}\right)^N \sup_{i\in[M]} (C^{(i)}(K_0) - C^{(i)}(K_i^*)) + c_{uni,1} \times \mathcal{B}(\epsilon_1, \epsilon_2).$$

The main message conveyed by Corollary 1 is that `FedLQR` can converge to a ball around the average optimal controller $K^*$ with a linear convergence rate. The size of the ball depends on the system heterogeneity level, i.e., $\epsilon_1$ and $\epsilon_2$. Combining Theorem 1 and Corollary 1, we infer that `FedLQR` not only approximates each system's optimal controller $K_i^*$ but also approximately converges toward the average optimal controller $K^*$ when the underlying $M$ systems are close (in the sense of Definition 1). The primary distinction between converging to $K_i^*$ and $K^*$ lies in the linear convergence rate. Compared to converging to $K_i^*$, where the linear converge rate depends only on system $i$'s parameter $\|\Sigma_{K_i^*}\|$, the rate in converging to $K^*$ depends on all systems' parameters, i.e., $\{\|\Sigma_{K_i^*}\|\}_{i=1}^N$. See Appendix A.4.3 for a comprehensive proof.

**How to ensure `FedLQR`'s stability?** We briefly discuss our proof technique for ensuring the iterative stability guarantees. The main idea is to leverage an inductive argument. We start from a stabilizing global policy $K_n \in \mathcal{G}^0$. We aim to show that the next global policy $K_{n+1}$ is stabilizing; see Fig. 2 for an illustration. This is achieved by demonstrating that $K_{n+1}$ can reduce each system's cost function compared to $K_n$. To achieve this goal, we take the following steps: (1) at each iteration, initiate from the globally stabilizing controller computed at the previous iterate, (2) determine a small global step-size such that inequalities in Section 7 can be applied; (3) use Lemma 3 to provide a descent direction to reduce each system's cost function; (4) bound the drift term $\frac{1}{ML}\sum_{i=1}^M \sum_{l=0}^{L-1} \|K_{n,l}^{(i)} - K_n\|^2$. Step (4) can be accomplished using a small local step-size $\eta_l$ such that each local policy is a small perturbation of the global policy $K_n$.

**On the role of control variates for local drift mitigation:** Control-variate techniques such as proposed in SCAFFOLD (Karimireddy et al., 2020) are designed to correct the optimization drift that arises in federated optimization under non-IID data by making each local update an approximately unbiased estimator of the global gradient. However, we emphasize that such methods do not eliminate the intrinsic gap between the minimizer of the averaged LQR cost $C_{avg}(K)$ and the individual optima $K_i^\star$ when the underlying dynamical systems differ. This limitation reflects a fundamental distinction between control-variate methods in classical FL and their applicability to our setting of dynamical systems. In standard FL, bias correction can ensure that the global iterates closely follow an "idea" gradient-descent step on a strongly convex objective, enabling the use of gradient-domination inequalities to guarantee fast convergence. In contrast, for heterogeneous LQR systems, the averaged cost $C_{avg}(K)$ does not, in general, correspond to the LQR cost of any single "averaged" dynamical system, and hence, one cannot directly appeal to gradient-domination properties like in the optimization setting to achieve fast linear rates. An important exception arises when all agents share identical dynamics but differ only through their cost matrices: in that case, $C_{avg}$ reduces to the LQR cost associated with $(A, B, \bar{Q}, \bar{R})$, where $\bar{Q} = (1/M)\sum_{i\in[M]} Q_i$ and $\bar{R} = (1/M)\sum_{i\in[M]} R_i$, making gradient-domination applicable and mitigating the local drift as discussed in Zhu et al. (2024).

Equipped with these model-based results, we are ready to present our main results of the model-free setting.

---

[11]The notation $\bar{h}_{het}^3$ is a positive scalar depending on the parameters of the LQR problem; see Appendix A.4.2 for full details.

## 8.2 Model-free setting

We now analyze `FedLQR`'s convergence in the model-free setting, where the policy gradient steps are approximately computed using zeroth-order optimization (Algorithm 2), without knowing the true dynamics, i.e., $A^{(i)}, B^{(i)}$ are not available and so $\nabla C^{(i)}(K^{(i)})$ can't be directly computed. The key point in this setting is to bound the gap between the estimated gradient and the true gradient. In the centralized setting (Fazel et al., 2018), the gap can be made arbitrarily accurate with enough trajectory samples $n_s$, sufficiently long trajectory length $\tau$, and small smoothing radius $r$.

We aim to achieve a sample complexity reduction for each agent by utilizing data from other similar but non-identical systems with the help of the server. This presents a significant challenge, as averaging gradient estimates from multiple agents may not necessarily reduce the variance even for homogeneous systems due to the high correlation between local gradient estimates. This challenge is compounded in our case as the gradient estimates are not only *correlated* but also come from *non-identical systems*. As a result, the variance reduction and sample complexity reduction for the `FedLQR` algorithm is not obvious a priori. After addressing these challenges using a martingale-based analysis, we show that one can establish variance reduction for our setting as well. This is formalized in the next result:

**Lemma 4.** *(Variance Reduction)* *Suppose the smoothing radius $r$ and trajectory length $\tau$ from Algorithm 2 satisfy $r \leq h_r\left(\frac{\epsilon}{4}\right)$ and $\tau \geq h_\tau\left(\frac{r\epsilon}{4n_x n_u}\right)$, respectively.[12] Moreover, suppose the sample size satisfies:[13]*

$$n_s \geq \frac{h_{sample,trunc}\left(\frac{\epsilon}{4}, \frac{\delta}{ML}, \frac{H^2}{\mu}\right)}{ML}. \tag{4}$$

*Then, when $K_n \in \mathcal{G}^0$, with probability $1 - \delta$, the estimated gradients satisfy:*

$$\left\| \frac{1}{ML} \sum_{i=1}^{M} \sum_{l=0}^{L-1} \left[ \widehat{\nabla} C^{(i)}(K_{n,l}^{(i)}) - \nabla C^{(i)}(K_{n,l}^{(i)}) \right] \right\|_F \leq \epsilon.$$

We prove this result and provide the definition of the parameters of $h_{sample,trunc}$ in Appendix A.6. The most important information conveyed by our variance reduction lemma is that each agent at each local step only needs to sample $\frac{1}{ML}$ fraction of samples required in the centralized setting. Notably, this lemma plays an important role in showing that `FedLQR` can help improve the sample efficiency. Equipped with Lemma 4, we now present the main convergence guarantees for `FedLQR`:

**Theorem 2.** *(Model-free)* *Suppose the trajectory length satisfies $\tau \geq h_\tau\left(\frac{r\epsilon'}{4n_x n_u}\right)$, the smoothing radius satisfies $r \leq h'_r\left(\frac{\epsilon'}{4}\right)$, and the sample size of each agent $n_s$ satisfies Eq. (4) with $\epsilon' = \frac{4\left\|\Sigma_{K_i^\star}\right\|}{\mu^2 \sigma_{\min}(R)}\epsilon$. When the heterogeneity level satisfies $(\epsilon_1 \bar{h}_{het}^1 + \epsilon_2 \bar{h}_{het}^2)^2 \leq \bar{h}_{het}^3$, then, given any $\delta \in (0,1)$, with probability $1 - \delta$, there exist constant step-sizes $\eta_g$ and $\eta_l$, which are independent of $\epsilon'$, such that `FedLQR` enjoys the following performance guarantees:*

1. *(Stability of the global policy)* *The global policy at each round $n$ is stabilizing, i.e., $K_n \in \mathcal{G}^0$;*

2. *(Stability of the local policies)* *The local policies satisfy $K_{n,l}^{(i)} \in \mathcal{G}^0$ for all $i$ and $l$;*

3. *(Convergence rate)* *After $N \geq \frac{\left\|\Sigma_{K_i^\star}\right\|}{\eta\mu^2\sigma_{\min}(R)} \log\left(\frac{2(C^{(i)}(K_0) - C^{(i)}(K_i^\star))}{\epsilon'}\right)$ rounds, we have*

$$C^{(i)}(K_N) - C^{(i)}(K_i^\star) \leq \epsilon' + 2\mathcal{B}(\epsilon_1, \epsilon_2), \forall i \in [M], \tag{5}$$

*where $\mathcal{B}(\epsilon_1, \epsilon_2)$ is defined in Theorem 1.*

---

[12]The notation $h_r$, $h_\tau$, $h_{\text{sample,trunc}}$ and $h'_r$ in Lemma 4 and Theorem 2 are polynomial functions of the LQR problem, depending on $\epsilon$. For simplicity, we defer their definition to the appendix.

[13]For the convenience of comparison with existing literature, we use the same notation as in Fazel et al. (2018).

This theorem establishes the finite-time convergence guarantees for `FedLQR`. The first two points in Theorem 2 provide the iterative stability guarantees of `FedLQR`, i.e., the trajectories of `FedLQR` will always stay inside the stabilizing set $\mathcal{G}^0$. The third point implies that when heterogeneity is small, i.e., $\mathcal{B}(\epsilon_1, \epsilon_2)$ is negligible, `FedLQR` converges to each system's optimal policy with a linear speedup w.r.t. the number of agents $M$, which we discuss further next.

**Discussion:** For a fixed desired precision $\epsilon$, we denote $N$ to be the number of rounds such that the first term $\epsilon'$ in Eq (5) is smaller than $\epsilon$. In what follows, we focus on analyzing the total sample complexity of `FedLQR` for each agent, which can be calculated by $N \times L \times n_s$. Note that $N$, in our case, is in the same order as the centralized setting. However, in terms of the sample size $n_s$ requirement at each local step, it is only a $\frac{1}{ML}$-fraction of that needed in the centralized setting, as presented in the variance reduction Lemma 4. Therefore, in a low-heterogeneity regime, where $\mathcal{B}(\epsilon_1, \epsilon_2)$ is negligible, our *FedLQR algorithm* reduces the sample complexity of learning the optimal LQR policy by $\tilde{\mathcal{O}}(\frac{1}{M})$ of the centralized setting (Fazel et al., 2018; Malik et al., 2019).[14] Specifically, `FedLQR` improves the sample cost required by each agent from $\tilde{\mathcal{O}}(\frac{1}{\epsilon^2})$ to $\tilde{\mathcal{O}}(\frac{1}{M\epsilon^2})$ up to a small heterogeneity bias term. This result is highly desirable since the number of agents in FL is usually large; leading to a significant speedup due to collaboration.

It is important to mention that our results also capture the cost of federation embedded in the term $\mathcal{B}(\epsilon_1, \epsilon_2)$. That is when two systems exhibit significant differences from each other, leveraging data across them may not be beneficial in finding a common stabilizing policy that applies to both. *In summary, Eq. (4)–(5) provide an explicit interplay between the price of heterogeneity and the benefit of collaboration.* The trade-off in Theorem 2 is explored in the simulation study presented in the next section.

## 9 Numerical Results

The following section describes the experimental setup and results for `FedLQR` in the model-free setting[15].

### 9.1 System Generation

Numerical experiments are conducted to illustrate and evaluate the effectiveness of `FedLQR` (Algorithm 1). The simulations involve different and unstable dynamical systems described by discrete-time linear time-invariant (LTI) models, as in (4), where each system has $n_x = 3$ states and $n_u = 3$ inputs. To generate different systems while respecting the bounded heterogeneity assumption (Assumption 1), the following steps are followed:

1. Given nominal system matrices $(A_0, B_0)$, generate random variables $\gamma_1^{(i)} \sim \mathcal{U}(0, \epsilon_1)$ and $\gamma_2^{(i)} \sim \mathcal{U}(0, \epsilon_2)$, $\forall i \in [M]$, with $\epsilon_1$ and $\epsilon_2$ being predefined dissimilarity parameters.

2. The random variables generated above are combined with modification masks $Z_1 \in \mathbb{R}^{3 \times 3}$ and $Z_2 \in \mathbb{R}^{3 \times 3}$ to generate the different systems matrices $(A^{(i)}, B^{(i)})$ for all $i \in [M]$.

3. The systems $(A^{(i)}, B^{(i)})$ for $0 < i \leq M$ are then constructed by perturbing the nominal systems according to: $A^{(i)} = A_0 + \gamma_1^{(i)} Z_1$ and $B^{(i)} = B_0 + \gamma_2^{(i)} Z_2$, where $Z_1$ and $Z_2$ are defined in step 2.

4. The nominal matrices are included in the set of generated systems as $(A^{(1)}, B^{(1)}) = (A_0, B_0)$.

In particular, we consider

$$A_0 = \begin{bmatrix} 1.20 & 0.50 & 0.40 \\ 0.01 & 0.75 & 0.30 \\ 0.10 & 0.02 & 1.50 \end{bmatrix}, \ \ B_0 = I_3, \ \ Q = 2I_3, \text{ and } R = \frac{1}{2}I_3,$$

---

[14]In Fazel et al. (2018), the sample complexity of policy gradient with one-point zeroth-order estimation is $\tilde{\mathcal{O}}(\frac{1}{\epsilon^4})$, this was later improved to $\tilde{\mathcal{O}}(\frac{1}{\epsilon^2})$ by Malik et al. (2019). We compare our results to the refined analysis in Malik et al. (2019).

[15]Code can be downloaded from https://github.com/jd-anderson/FedLQR

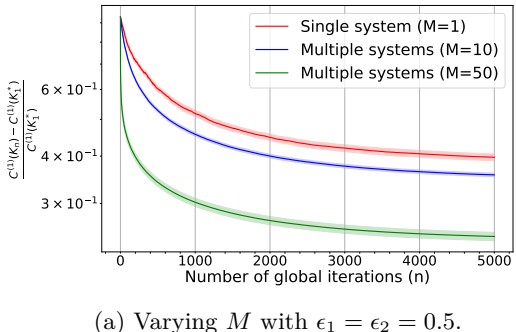 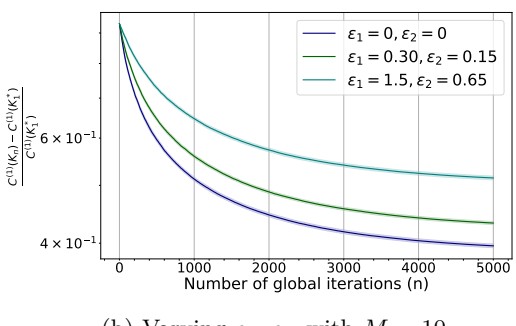

(a) Varying $M$ with $\epsilon_1 = \epsilon_2 = 0.5$.  (b) Varying $\epsilon_1$, $\epsilon_2$, with $M = 10$.

Figure 3: Gap between the current and optimal cost with respect to the number of global iterations.

for the nominal system matrices and cost matrices respectively.

The optimal controller for the nominal system $(A^{(1)}, B^{(1)})$ is

$$K_1^* = \begin{bmatrix} 1.0056 & 0.4293 & 0.3570 \\ 0.0262 & 0.6239 & 0.2657 \\ 0.1003 & 0.0298 & 1.2960 \end{bmatrix},$$

and was obtained by solving the discrete algebraic Riccati equation (DARE).

## 9.2 Algorithm Parameters

For the gradient estimation step in the zeroth-order algorithm (Algorithm 2), we set the initial state for cost computation as a random sample from a standard normal distribution, denoted as $\mathcal{D} \stackrel{d}{=} \mathcal{N}(0, I_3)$, for all systems $i \in [M]$. Additionally, we consider $n_s = 5$ trajectories, where each trajectory has a rollout length of $\tau = 15$, and we set the smoothing radius $r = 0.1$ for the zeroth-order gradient estimation.

Throughout our simulations, we consider the following initial stabilizing controller $K_0 = 1.62 I_3$ (Line 1 in Algorithm 1). Note that although the control action $u_t^{(i)} = -K_0 x_t^{(i)}$ may not be optimal for any of the $M$ systems. For example, the suboptimality of $K_0$ applied to the nominal system is evidenced by its cost of $C^{(1)}(K_0) = 18.4049$, compared to the optimal cost of $C^{(1)}(K_1^*) = 9.5220$, when computed from an initial state $x_0^{(1)} = [1 \ 1 \ 1]^\top$ and time horizon $T = 500$. However, it is important to note that $K_0$ is still stabilizing all $M$ systems. Note that we will use $K_0$ as the initial controller for all of the experiments in this paper.

## 9.3 Experiments

To assess the performance of FedLQR, we evaluate the normalized gap between the current cost $C^{(1)}(K_n)$ of the nominal system when using the common stabilizing controller $K_n$ and its corresponding optimal cost $C^{(1)}(K_1^*)$. This metric is represented as $\frac{C^{(1)}(K_n) - C^{(1)}(K_1^*)}{C^{(1)}(K_1^*)}$ for each global iteration $n \in [N]$. In our experiments, we set the step sizes as $\eta_g = 1 \times 10^{-2}$, with an adaptive decrease of $0.05\%$ per global iteration, and $\eta = 1 \times 10^{-4}$. Further details regarding other parameters, such as the number of systems $M$, heterogeneity levels $(\epsilon_1, \epsilon_2)$ are provided in the figures and the subsequent discussion.

Figures 3-(a,b) present the normalized distance between the current cost associated with the common stabilizing controller and the optimal cost for the nominal system, plotted with respect to the number of global iterations. These figures demonstrate the impact of varying the number of systems $M$ and the heterogeneity parameters $(\epsilon_1, \epsilon_2)$ on the convergence and performance of Algorithm 1.

In Figure 3-(a), we specifically investigate the effect of the number of systems $M$ participating in the collaboration to compute a common controller $K^*$ on the convergence of our algorithm. In this analysis, we set the heterogeneity parameters as $\epsilon_1 = 0.5$ and $\epsilon_2 = 0.5$ and consider modification masks $Z_1 = Z_2 = I_3$.

The figure reveals a noticeable reduction in the gap between the current and optimal cost as the number of participating systems $M$ increases. This numerical result aligns with our theoretical findings, which indicate that the number of samples required to achieve reliable estimation for the cost function's gradient can be scaled down with the number of systems participating in the collaboration. Consequently, as the number of systems involved increases, there is a considerable reduction in the gap between the common computed controller and the optimal one.

Figure 3-(b) illustrates the influence of the heterogeneity parameters $(\epsilon_1, \epsilon_2)$ on the convergence rate of Algorithm 1. In this analysis, we set the number of systems as $M = 10$, and the modification masks $Z_1 = \texttt{diag}([3.5 \ 1 \ 0.1])$ and $Z_2 = \texttt{diag}([1.5 \ 0.1 \ 1])$. Consistent with our theoretical findings, we observe that an increase in the dissimilarity among the systems results in a significant gap between the common and optimal controller. This discrepancy arises due to the additive effect of system heterogeneity on the convergence rate of our algorithm, as elaborated in Theorem 2.

## 10 Conclusions and Future Work

We investigated the problem of learning a common and optimal LQR policy with the objective of minimizing an average quadratic cost. The primary focus of this paper was to thoroughly examine and provide comprehensive answers to the following questions: (i) Is the learned common policy stabilizing for all agents? (ii) How close is the learned common policy to each agent's own optimal policy? (iii) Can each agent learn its own optimal policy faster by leveraging data from all agents? To address these questions, we proposed a federated and model-free approach, `FedLQR`, where $M$ heterogeneous systems collaborate to learn a common and optimal policy while keeping the system's data *private*. Our analysis tackles numerous technical challenges, including system heterogeneity, multiple local gradient descent updates, and stability. We have demonstrated that `FedLQR` produces a common policy that stabilizes all systems and converges to the optimal policy (Theorem 2) of each agent up to a heterogeneity-induced bias term. Furthermore, `FedLQR` achieves a reduction in sample complexity proportional to the number of participating agents $M$ (Lemma 4). We have also provided numerical results to effectively showcase and evaluate the performance of our `FedLQR` approach in a model-free setting. Future work will address the assumption of requiring full-state information to extend our results to the Linear Quadratic Gaussian (LQG) problem in a federated setting. We are currently investigating data-driven and system-theoretic metrics for heterogeneity, as well as personalization-based methods to mitigate the impact of heterogeneity on the performance of the proposed approach.

## Acknowledgments

The authors thank the anonymous reviewers for their thorough and constructive feedback, which significantly improved this work. Leonardo F. Toso is funded by the Center for AI and Responsible Financial Innovation (CAIRFI) Fellowship and by the Columbia Presidential Fellowship. James Anderson is partially funded by NSF grants ECCS 2144634 and 2231350 and the Columbia Center of AI Technology in collaboration with Amazon.

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

# A   Appendix

## A.1   Appendix Roadmap

This appendix is organized as follows. Sections A.2 and A.3 present auxiliary norm inequalities and lemmas that play a key role in proving the main results of this paper. The proof of our main results in the model-based setting is provided in Section A.4, while Section A.6 is dedicated to the corresponding results in the model-free setting. Additional details on the zeroth-order optimization method are provided in Section A.5.

### A.1.1   Notation Recap

For convenience we briefly recap and summarize our notation. We use $\|S\|_{max}$ to denote the maximum spectral norm taken over the family of matrices $S^{(1)}, \ldots, S^{(M)}$. All norms for matrices and vectors are spectral and Euclidean respectively, unless otherwise stated. The integer sequence $\{1, 2, \ldots, N\}$ is denoted as $[N]$. The spectral radius of a square matrix is denoted by $\rho(\cdot)$.

Table 1: Notations

| Symbol | Meaning |
|---|---|
| $M$ | number of systems |
| $L$ | number of local updates (counter: $l$) |
| $N$ | number of rounds of averaging (counter: $n$) |
| $K_n$ | averaged controller at round $n$ |
| $K_i^*$ | optimal controller for system $(A^{(i)}, B^{(i)})$ |
| $K_{n,l}^{(i)}$ | controller for system $i$ after $l$ local iterations and $n$ averaging rounds |

## A.2   Useful Norm Inequalities

- Given any two matrices $A, B$ of the same dimensions, for any $\xi > 0$, we have

$$\|A + B\|_F^2 \leq (1 + \xi)\|A\|_F^2 + \left(1 + \frac{1}{\xi}\right)\|B\|_F^2. \tag{6}$$

- Given any two matrices $A, B$ of the same dimensions, for any $\xi > 0$, we have

$$\langle A, B \rangle \leq \frac{\xi}{2}\|A\|_F^2 + \frac{1}{2\xi}\|B\|_F^2. \tag{7}$$

This inequality goes by the name of Young's inequality.

- Given $m$ matrices $A_1, \ldots, A_m$ of the same dimensions, the following is a simple application of Jensen's inequality:

$$\left\|\sum_{i=1}^m A_i\right\|^2 \leq m \sum_{i=1}^m \|A_i\|^2,$$

$$\left\|\sum_{i=1}^m A_i\right\|_F^2 \leq m \sum_{i=1}^m \|A_i\|_F^2. \tag{8}$$

- Given any two vectors $x, y \in \mathbb{R}^d$, for any constant $\zeta > 0$, we have

$$\|x + y\|^2 \leq (1 + \zeta)\|x\|^2 + \left(1 + \frac{1}{\zeta}\right)\|y\|^2. \tag{9}$$

- Given any two vectors $x, y \in \mathbb{R}^d$, for any constant $\zeta > 0$, we have

$$\langle x, y \rangle \leq \frac{\zeta}{2}\|x\|^2 + \frac{1}{2\zeta}\|y\|^2. \tag{10}$$

### A.3 Useful Lemmas and Constants

**Lemma 5.** *For each $i \in [M]$, we have that:*

$$||\Sigma_K^{(i)}|| \leq \frac{C^{(i)}(K)}{\sigma_{min}(Q)}, \quad ||P_K^{(i)}|| \leq \frac{C^{(i)}(K)}{\mu}. \tag{11}$$

*Proof.* The proof of this lemma is detailed in (Fazel et al., 2018, Lemma 13). $\qquad\square$

**Lemma 6.** *(Uniform bounds for $\nabla C^{(i)}(K)$ and $||K||$) For each agent $i \in [M]$, the gradient $\nabla C^{(i)}(K)$ and $\|K\|$ can be bounded as follows:*

$$\|\nabla C^{(i)}(K)\| \leq \|\nabla C^{(i)}(K)\|_F \leq h_1(K) \quad and \quad \|K\| \leq h_2(K),$$

*where $h_1(K)$, and $h_2(K)$ are some positive scalars depending on the function $C(K)$.*

*Proof.* In this lemma, $h_1(K)$, and $h_2(K)$ are the functions defined as:

$$h_0(K) := \sqrt{\frac{\|R_K\|_{\max}(C_{\max}(K) - C_{\min}(K))}{\mu}},$$

$$h_1(K) := \frac{C_{\max}(K)h_0(K)}{\sigma_{\min}(Q)}, \quad h_2(K) := \frac{h_0(K) + \|B^\top P_K A\|_{\max}}{\sigma_{\min}(R)},$$

where $\|R_K\|_{\max} := \max_i \left\|R + B^{(i)\top} P_K^{(i)} B^{(i)}\right\|$. By using (Fazel et al., 2018, Lemma 13), we have

$$\|\nabla C^{(i)}(K)\|^2 \leq \text{Tr}\left(\Sigma_K^{(i)} E_K^{(i)\top} E_K^{(i)} \Sigma_K^{(i)}\right) \leq \left\|\Sigma_K^{(i)}\right\|^2 \text{Tr}\left(E_K^{(i)\top} E_K^{(i)}\right)$$

$$\leq \left(\frac{C^{(i)}(K)}{\sigma_{\min}(Q)}\right)^2 \text{Tr}\left(E_K^{(i)\top} E_K^{(i)}\right).$$

By (Fazel et al., 2018, Lemma 11), we obtain

$$\text{Tr}\left(E_K^{(i)\top} E_K^{(i)}\right) \leq \frac{\left\|R + B^{(i)\top} P_K^{(i)} B^{(i)}\right\|\left(C^{(i)}(K) - C^{(i)}(K_i^*)\right)}{\mu},$$

which proves the first claim:

$$\|\nabla C^{(i)}(K)\| \leq \frac{C^{(i)}(K)}{\sigma_{\min}(Q)} \sqrt{\frac{\left\|R + B^{(i)\top} P_K^{(i)} B^{(i)}\right\|\left(C^{(i)}(K) - C^{(i)}(K_i^*)\right)}{\mu}}$$

$$\leq \frac{C_{\max}(K)}{\sigma_{\min}(Q)} \sqrt{\frac{\left\|R + B^{(i)\top} P_K^{(i)} B^{(i)}\right\|_{\max}(C_{\max}(K) - C_{\min}(K))}{\mu}}.$$

On the other hand, by exploiting (Fazel et al., 2018, Lemma 11) we can also write

$$
\begin{aligned}
\|K\| &\leq \left\| \left( R + B^{(i)\top} P_K^{(i)} B^{(i)} \right)^{-1} \right\| \left\| \left( R + B^{(i)\top} P_K^{(i)} B^{(i)} \right) K \right\| \\
&\leq \frac{1}{\sigma_{\min}(R)} \left\| \left( R + B^{(i)\top} P_K^{(i)} B^{(i)} \right) K \right\| \\
&\leq \frac{1}{\sigma_{\min}(R)} \left( \left\| \left( R + B^{(i)\top} P_K^{(i)} B^{(i)} \right) K - B^{(i)\top} P_K^{(i)} A^{(i)} \right\| + \left\| B^{(i)\top} P_K^{(i)} A^{(i)} \right\| \right) \\
&= \frac{\left\| E_K^{(i)} \right\|}{\sigma_{\min}(R)} + \frac{\left\| B^{(i)\top} P_K^{(i)} A^{(i)} \right\|}{\sigma_{\min}(R)} \\
&\leq \frac{\sqrt{\operatorname{Tr}\left( E_K^{(i)\top} E_K^{(i)} \right)}}{\sigma_{\min}(R)} + \frac{\left\| B^{(i)\top} P_K^{(i)} A^{(i)} \right\|}{\sigma_{\min}(R)} \\
&= \frac{\sqrt{\left( C^{(i)}(K) - C^{(i)}\left( K_i^* \right) \right) \left\| R + B^{(i)\top} P_K^{(i)} B^{(i)} \right\|}}{\sqrt{\mu}\,\sigma_{\min}(R)} + \frac{\left\| B^{(i)\top} P_K^{(i)} A^{(i)} \right\|}{\sigma_{\min}(R)},
\end{aligned}
$$

which completes the proof for the second claim. $\qquad\square$

It is worth noting that the local cost and gradient smoothness in Lemma 1 not only hold for the single-agent setting but also hold for the multi-agent setting. Moreover, we will make use of the following matrix concentration inequality.

**Definition 3** (Matrix Martingale). *Consider a random process $\{Y_k : k = 0, 1, 2, \ldots\}$ with matrices $Y_k$ of finite dimension, we say that the process is martingale if $Y_0 = 0$, $\mathbb{E}_{k-1} Y_k = Y_{k-1}$, and $\mathbb{E}\|Y_k\| < \infty$, where $\mathbb{E}_{k-1}$ is the expectation conditioned on a filtration $\mathcal{F}_{k-1}$.*

**Lemma 7.** *(Rectangular Matrix Freedman (Tropp, 2011)). Consider a matrix martingale $\{Y_k : k = 0, 1, 2, \ldots\}$ whose values are matrices with dimension $d_1 \times d_2$, and denote the difference sequence as $\{X_k = Y_k - Y_{k-1} : k = 1, 2, 3, \ldots\}$. Assume that the difference sequence is uniformly bounded:*

$$
\|X_k\| \leq R \quad \text{almost surely} \quad \text{for } k = 1, 2, 3, \ldots.
$$

*Define two predictable quadratic variation processes for this martingale:*

$$
W_{\mathrm{col},k} := \sum_{j=1}^{k} \mathbb{E}_{j-1}\left( X_j X_j^* \right) \quad \text{and}
$$

$$
W_{row,k} := \sum_{j=1}^{k} \mathbb{E}_{j-1}\left( X_j^* X_j \right) \quad \text{for } k = 1, 2, 3, \ldots
$$

*Then, for all $t \geq 0$ and $\sigma^2 > 0$,*

$$
\mathbb{P}\left\{ \exists k \geq 0 : \|Y_k\| \geq t \text{ and } \max\left\{ \|W_{col,k}\|, \|W_{row,k}\| \right\} \leq \sigma^2 \right\} \leq (d_1 + d_2) \cdot \exp\left\{ \frac{-t^2/2}{\sigma^2 + Rt/3} \right\}.
$$

### A.3.1 Proof of Lemma 1

*Proof.* In this proof, we aim to show

$$
\left| C^{(i)}\left( K' \right) - C^{(i)}(K) \right| \leq h_{\mathrm{cost}}(K)\|K' - K\|,
$$

$$
\left\| \nabla C^{(i)}\left( K' \right) - \nabla C^{(i)}(K) \right\| \leq h_{\mathrm{grad}}(K)\|\Delta\|, \quad \text{and}
$$

$$
\left\| \nabla C^{(i)}\left( K' \right) - \nabla C^{(i)}(K) \right\|_F \leq h_{\mathrm{grad}}(K)\|\Delta\|_F,
$$

hold for all agents $i \in [M]$, and $K'$ satisfying $\|K' - K\| \leq h_\Delta(K) < \infty$. The term $h_\Delta(K)$ is the polynomial defined as

$$h_\Delta(K) := \frac{\sigma_{\min}(Q)\mu}{4\|B\|_{\max}C_{\max}(K)\left(\|A - BK\|_{\max} + 1\right)},$$

the term $h_{\mathrm{cost}}(K)$ and $h_{\mathrm{grad}}(K)$ are defined as

$$h_{\mathrm{cost}}(K) := \frac{4\operatorname{Tr}(\Sigma_0)C_{\max}(K)\|R\|}{\mu\sigma_{\min}(Q)}\left(\|K\| + \frac{h_\Delta(K)}{2} + \|B\|_{\max}\|K\|^2\left(\|A - BK\|_{\max} + 1\right)\frac{C_{\max}(K)}{\mu\sigma_{\min}(Q)}\right)$$

$$\begin{aligned}
h_{\mathrm{grad}}(K) :=& 4\left(\frac{C_{\max}(K)}{\sigma_{\min}(Q)}\right)\left[\|R\| + \|B\|_{\max}\left(\|A\|_{\max} + \|B\|_{\max}\left(\|K\| + h_\Delta(K)\right)\right)\right. \\
& \times \left.\left(\frac{h_{\mathrm{cost}}(K)C_{\max}(K)}{\operatorname{Tr}(\Sigma_0)}\right) + \|B\|_{\max}^2\frac{C_{\max}(K)}{\mu}\right] \\
& + 8\left(\frac{C_{\max}(K)}{\sigma_{\min}(Q)}\right)^2\left(\frac{\|B\|_{\max}\left(\|A - BK\|_{\max} + 1\right)}{\mu}\right)h_0(K).
\end{aligned}$$

For the single-agent (i.e., $M = 1$) setting, the proof is explained in detail in (Fazel et al., 2018, Lemma 24 and 25). For the multi-agent setting (i.e., $M > 1$), we can complete the proof by taking the maximum over the clients $i \in [M]$ of all the system-dependent parameters, such as $\|B\|_{\max}$. $\qquad\square$

### A.3.2 Proof of Lemma 2

*Proof.* For the single-agent (i.e., $M = 1$) setting, the proof is explained in (Fazel et al., 2018, Lemma 11). For the multi-agent setting (i.e., $M > 1$), it is easy to see that

$$C^{(i)}(K) - C^{(i)}(K_i^*) \leq \frac{\left\|\Sigma_{K_i^*}\right\|}{4\mu^2\sigma_{\min}(R)}\|\nabla C^{(i)}(K)\|_F^2 \leq \frac{\max_i\left\|\Sigma_{K_i^*}\right\|}{4\mu^2\sigma_{\min}(R)}\|\nabla C^{(i)}(K)\|_F^2$$

holds for any stabilizing controller $K$ and any agent $i \in [M]$. $\qquad\square$

### A.4 The model-based setting

We first introduce the following operators on a symmetric matrix $X$,

$$\begin{aligned}
\mathcal{T}_K^{(i)}(X) &:= \sum_{t=0}^\infty (A^{(i)} - B^{(i)}K)^t X \left[(A^{(i)} - B^{(i)}K)^\top\right]^t, \\
\mathcal{F}_K^{(i)}(X) &:= (A^{(i)} - B^{(i)}K)X(A^{(i)} - B^{(i)}K)^\top.
\end{aligned} \tag{12}$$

We also define the induced norms of $\mathcal{T}$ and $\mathcal{F}$ as

$$\|\mathcal{T}_K\| = \sup_X \frac{\|\mathcal{T}_K(X)\|}{\|X\|}, \quad \|\mathcal{F}_K\| = \sup_X \frac{\|\mathcal{F}_K(X)\|}{\|X\|}.$$

**Lemma 8.** *When $(A^{(i)} - B^{(i)}K)$ has spectral radius smaller than 1, we have*

$$\mathcal{T}_K^{(i)} = \left(\mathrm{I} - \mathcal{F}_K^{(i)}\right)^{-1}$$

*holds for each $i \in [M]$.*

*Proof.* The proof is detailed in (Fazel et al., 2018, Lemma 18). $\qquad\square$

**Lemma 9.** *If* [16]

$$\left\|\mathcal{T}_K^{(i)}\right\| \left\|\mathcal{F}_K^{(i)} - \mathcal{F}_K^{(j)}\right\| \leq \frac{1}{2} \tag{13}$$

*holds for any system $i, j \in [M]$, then we have*

$$\left\|\left(\mathcal{T}_K^{(i)} - \mathcal{T}_K^{(j)}\right)(X)\right\| \leq 2 \left\|\mathcal{T}_K^{(i)}\right\| \left\|\mathcal{F}_K^{(i)} - \mathcal{F}_K^{(j)}\right\| \left\|\mathcal{T}_K^{(i)}(X)\right\|$$

$$\leq 2 \left\|\mathcal{T}_K^{(i)}\right\|^2 \left\|\mathcal{F}_K^{(i)} - \mathcal{F}_K^{(j)}\right\| \| X\|.$$

*Proof.* Define $\mathcal{A} = \mathrm{I} - \mathcal{F}_K^{(i)}$, and $\mathcal{B} = \mathcal{F}_K^{(i)} - \mathcal{F}_K^{(j)}$. In this case $\mathcal{A}^{-1} = \mathcal{T}_K^{(i)}$ and $(\mathcal{A} - \mathcal{B})^{-1} = \mathcal{T}_K^{(j)}$. Hence, the condition $\left\|\mathcal{T}_K^{(i)}\right\| \left\|\mathcal{F}_K^{(i)} - \mathcal{F}_K^{(j)}\right\| \leq \frac{1}{2}$ translates to the condition $\|\mathcal{A}^{-1}\| \|\mathcal{B}\| \leq \frac{1}{2}$.

First, we observe that

$$\left(\mathcal{A}^{-1} - (\mathcal{A} - \mathcal{B})^{-1}\right)(X) = \left(\mathrm{I} - \left(\mathrm{I} - \mathcal{A}^{-1} \circ \mathcal{B}\right)^{-1}\right)\left(\mathcal{A}^{-1}(X)\right)$$

$$= \left(\mathrm{I} - \left(\mathrm{I} - \mathcal{A}^{-1} \circ \mathcal{B}\right)^{-1}\right)\left(\mathcal{T}_K^{(i)}(X)\right), \tag{14}$$

where $f \circ g$ denotes the composition $f(g(x))$. Since

$$\left(\mathrm{I} - \mathcal{A}^{-1} \circ \mathcal{B}\right)^{-1} = \mathrm{I} + \mathcal{A}^{-1} \circ \mathcal{B} \circ \left(\mathrm{I} - \mathcal{A}^{-1} \circ \mathcal{B}\right)^{-1},$$

we have:

$$\left\|\left(\mathrm{I} - \mathcal{A}^{-1} \circ \mathcal{B}\right)^{-1}\right\| \leq 1 + \left\|\mathcal{A}^{-1} \circ \mathcal{B}\right\| \left\|\left(\mathrm{I} - \mathcal{A}^{-1} \circ \mathcal{B}\right)^{-1}\right\| \leq 1 + \frac{1}{2}\left\|\left(\mathrm{I} - \mathcal{A}^{-1} \circ \mathcal{B}\right)^{-1}\right\| \tag{15}$$

Now rearranging terms in Eq.(15), we obtain $\left\|\left(\mathrm{I} - \mathcal{A}^{-1} \circ \mathcal{B}\right)^{-1}\right\| \leq 2$. Therefore, we have

$$\left\|\mathrm{I} - \left(\mathrm{I} - \mathcal{A}^{-1} \circ \mathcal{B}\right)^{-1}\right\| = \left\|\mathcal{A}^{-1} \circ \mathcal{B} \circ \left(\mathrm{I} - \mathcal{A}^{-1} \circ \mathcal{B}\right)^{-1}\right\| \leq \left\|\mathcal{A}^{-1}\right\| \|\mathcal{B}\| \left\|\left(\mathrm{I} - \mathcal{A}^{-1} \circ \mathcal{B}\right)^{-1}\right\|$$

$$\leq 2 \left\|\mathcal{A}^{-1}\right\| \|\mathcal{B}\|,$$

and so

$$\left\|\mathrm{I} - \left(\mathrm{I} - \mathcal{A}^{-1} \circ \mathcal{B}\right)^{-1}\right\| \leq 2 \left\|\mathcal{A}^{-1}\right\| \|\mathcal{B}\| = 2 \left\|\mathcal{T}_K^{(i)}\right\| \left\|\mathcal{F}_K^{(i)} - \mathcal{F}_K^{(j)}\right\|. \tag{16}$$

Then, we have

$$\left\|\left(\mathcal{T}_K^{(i)} - \mathcal{T}_K^{(j)}\right)(X)\right\| = \left\|\left(\mathcal{A}^{-1} - (\mathcal{A} - \mathcal{B})^{-1}\right)(X)\right\|$$

$$\overset{(a)}{\leq} \left\|\left(\mathrm{I} - \left(\mathrm{I} - \mathcal{A}^{-1} \circ \mathcal{B}\right)^{-1}\right)\right\| \left\|\mathcal{T}_K^{(i)}(X)\right\|$$

$$\overset{(b)}{\leq} 2 \left\|\mathcal{T}_K^{(i)}\right\| \left\|\mathcal{F}_K^{(i)} - \mathcal{F}_K^{(j)}\right\| \left\|\mathcal{T}_K^{(i)}(X)\right\|$$

$$\leq 2 \left\|\mathcal{T}_K^{(i)}\right\| \left\|\mathcal{F}_K^{(i)} - \mathcal{F}_K^{(j)}\right\| \left\|\mathcal{T}^{(i)}\right\| \|X\|,$$

where $(a)$ is due to Eq.(14) and $(b)$ is due to Eq.(16). This completes the proof of Lemma 9. $\qquad\square$

---

[16]This lemma has a similar flavor to that of Lemma 20 in (Fazel et al., 2018). It is worthwhile to mention that the inequality (13) imposes certain conditions on heterogeneity. Note that the constant $\frac{1}{2}$ can be changed into any finite constant. Thus, this heterogeneity requirement can be subsumed by that in Eq.(21).

### A.4.1 Proof of Lemma 3

*Proof.* First, we know that $\nabla C^{(i)}(K)$ and $\nabla C^{(j)}(K)$ are given by,

$$\nabla C^{(i)}(K) = 2E_K^{(i)}\Sigma_K^{(i)}, \quad \text{and} \quad \nabla C^{(j)}(K) = 2E_K^{(j)}\Sigma_K^{(j)}$$

where,

$$E_K^{(i)} = (R + B^{(i)\top}P_K^{(i)}B^{(i)})K - B^{(i)\top}P_K^{(i)}A^{(i)},$$

and

$$\Sigma_K^{(i)} = _{x_0^{(i)}\sim D}\sum_{t=0}^{\infty}x_t^{(i)}x_t^{(i)\top}.$$

Thus, we can write,

$$||\nabla C^{(i)}(K) - \nabla C^{(j)}(K)|| = ||2E_K^{(i)}\Sigma_K^{(i)} - 2E_K^{(j)}\Sigma_K^{(j)}||$$
$$\leq 2(||E_K^{(i)} - E_K^{(j)}||\underbrace{||\Sigma_K^{(i)}||}_{\beta_1} + \underbrace{||E_K^{(j)}||}_{\beta_2}||\Sigma_K^{(i)} - \Sigma_K^{(j)}||).$$

From Eq. (11) we can upper bound $||\Sigma_K^{(i)}||$ as follows:

$$||\Sigma_K^{(i)}|| \leq \frac{C^{(i)}(K)}{\sigma_{\min}(Q)}.$$

With the definition of $E_K^{(j)} = RK + B^{(j)\top}P_K^{(j)}B^{(j)}K - B^{(j)\top}P_K^{(j)}A^{(j)}$, we can use triangle inequality to write,

$$||E_K^{(j)}|| \leq ||RK|| + ||B^{(j)}||||P_K^{(j)}||||B^{(j)}K|| + ||B^{(j)}||||P_K^{(j)}||||A^{(j)}||$$
$$\leq ||RK|| + \frac{||B^{(j)}||C^{(j)}(K)}{\mu}(||B^{(j)}K|| + ||A^{(j)}||),$$

where $||P_K^{(j)}|| \leq \frac{C^{(j)}(K)}{\mu}$ from Eq. (11), with $\mu = \sigma_{\min}(\Sigma_0^{(j)})$.

With the notation that we introduced previously, we can write

$$\beta_1 = ||\Sigma_K^{(i)}|| \leq ||\Sigma_K||_{\max} \leq \frac{C_{\max}(K)}{\sigma_{\min}(Q)},$$

and,

$$\beta_2 = ||E_K^{(j)}|| \leq ||E_K||_{\max} \leq ||R||||K|| + \frac{||B||_{\max}C_{\max}(K)}{\mu}(||B||_{\max} + ||A||_{\max}),$$

where $C_{\max}(K) := \max_i C^{(i)}(K)$.

Next we will derive an upper bound for $||E_K^{(i)} - E_K^{(j)}||$.

**Upper bound for $||E_K^{(i)} - E_K^{(j)}||$:** We can first use the definition of $E_K^{(i)}$ and $E_K^{(j)}$ to write,

$$E_K^{(i)} - E_K^{(j)} = B^{(j)\top}P_K^{(j)}(A^{(j)} - B^{(j)}K) - B^{(i)\top}P_K^{(i)}(A^{(i)} - B^{(i)}K)$$
$$= -B^{(i)\top}P_K^{(i)}(A^{(i)} - B^{(i)}K) + B^{(i)\top}P_K^{(i)}(A^{(j)} - B^{(j)}K) - B^{(i)\top}P_K^{(i)}(A^{(j)} - B^{(j)}K)$$
$$+ B^{(i)\top}P_K^{(j)}(A^{(j)} - B^{(j)}K) - B^{(i)\top}P_K^{(j)}(A^{(j)} - B^{(j)}K) + B^{(j)\top}P_K^{(j)}(A^{(j)} - B^{(j)}K).$$

Then, by using triangle inequality, we obtain the following expression:

$$||E_K^{(i)} - E_K^{(j)}|| \leq || \underbrace{B^{(i)\top} P_K^{(i)} (A^{(i)} - B^{(i)} K) - B^{(i)\top} P_K^{(i)} (A^{(j)} - B^{(j)} K)}_{H_1} ||$$

$$+ || \underbrace{B^{(i)\top} P_K^{(i)} (A^{(j)} - B^{(j)} K) - B^{(i)\top} P_K^{(j)} (A^{(j)} - B^{(j)} K)}_{H_2} ||$$

$$+ || \underbrace{B^{(i)\top} P_K^{(j)} (A^{(j)} - B^{(j)} K) - B^{(j)\top} P_K^{(j)} (A^{(j)} - B^{(j)} K)}_{H_3} ||.$$

Incorporating the heterogeneity bounds from assumption 1 gives

$$||H_1|| \leq ||B^{(i)}|| ||P_K^{(i)}|| (\epsilon_1 + \epsilon_2 ||K||),$$

to which we apply the max-norm definition to arrive at

$$||H_1|| \leq ||B||_{\max} (\epsilon_1 + \epsilon_2 ||K||) ||P_K||_{\max}. \tag{17}$$

Similarly, we can also derive upper bounds for $||H_2||$ and $||H_3||$, as follows,

$$||H_2|| \leq ||B^{(i)}|| ||P_K^{(i)} - P_K^{(j)}|| ||A^{(j)} - B^{(j)} K|| \leq ||B||_{\max} ||P_K^{(i)} - P_K^{(j)}|| ||A - BK||_{\max} \tag{18}$$

and

$$||H_3|| \leq \epsilon_2 ||A^{(i)} - B^{(i)} K|| ||P_K^{(j)}|| \leq \epsilon_2 ||A - BK||_{\max} ||P_K||_{\max}. \tag{19}$$

To bound $H_2$, we need to derive an upper bound for $||P_K^{(i)} - P_K^{(j)}||$. For this purpose, we have that for any fixed system $i \in [M]$

$$||P_K^{(i)} - P_K^{(j)}|| = \left\| \mathcal{T}_K^{(i)} \left( Q + K^\top R K \right) - \mathcal{T}_K^{(j)} \left( Q + K^\top R K \right) \right\|.$$

Thus, by using Lemma 9, we can write,

$$||P_K^{(i)} - P_K^{(j)}|| \leq 2 \left\| \mathcal{T}_K^{(i)} \right\|^2 \left\| \mathcal{F}_K^{(i)} - \mathcal{F}_K^{(j)} \right\| \left\| Q + K^\top R K \right\|,$$

where $||\mathcal{T}_K^{(i)}|| \leq \frac{C^{(i)}(K)}{\sigma_{\min}(Q)\mu} \leq \frac{C_{\max}(K)}{\sigma_{\min}(Q)\mu}$ (detailed in Lemma 17 of (Fazel et al., 2018)). With the following upper bound for $\left\| \mathcal{F}_K^{(i)} - \mathcal{F}_K^{(j)} \right\|$:

$$||(\mathcal{F}_K^{(i)} - \mathcal{F}_K^{(j)})(X)|| = ||(A^{(i)} - B^{(i)} K) X (A^{(i)} - B^{(i)} K)^\top$$
$$- (A^{(j)} - B^{(j)} K) X (A^{(j)} - B^{(j)} K)^\top||$$
$$\leq 2(\epsilon_1 + \epsilon_2 ||K||) ||X|| ||A - BK||_{\max},$$

we have

$$||P_K^{(i)} - P_K^{(j)}|| \leq 4 \left( \frac{C_{\max}(K)}{\sigma_{\min}(Q)\mu} \right)^2 (\epsilon_1 + \epsilon_2 ||K||) ||A - BK||_{\max} (||Q|| + ||R|| ||K||^2), \tag{20}$$

Plugging in Eq. (20) into $H_2$ and adding the upper bounds of $H_1$ (Eq. 17), $H_2$ (Eq. 18) and $H_3$ (Eq. 19) together, we have

$$||E_K^{(i)} - E_K^{(j)}|| \leq g_1(\epsilon_1, \epsilon_2, K),$$

where $g_1$ is a linear in $\epsilon_1, \epsilon_2$ and polynomial in the remaining problem data. Specifically,

$$g_1(\epsilon_1, \epsilon_2, K) := \epsilon_1 \left( \frac{||B||_{\max} C_{\max}(K)}{\mu} \left[ 1 + 4 \left( \frac{C_{\max}(K)}{\sigma_{\min}(Q)\mu} \right) (||A - BK||_{\max})^2 \left( ||Q|| + ||R||||K||^2 \right) \right] \right)$$
$$+ \epsilon_2 \left( \frac{||B||_{\max}||K|| C_{\max}(K)}{\mu} \left[ 1 + 4 \left( \frac{C_{\max}(K)}{\sigma_{\min}(Q)\mu} \right) (||A - BK||_{\max})^2 \left( ||Q|| + ||R||||K||^2 \right) \right]$$
$$+ ||A - BK||_{\max} \right).$$

In what follows, we will derive an upper bound for $||\Sigma_K^{(i)} - \Sigma_K^{(j)}||$:

**Upper bound for** $||\Sigma_K^{(i)} - \Sigma_K^{(j)}||$**:** From the previous definitions in Eq.(12) and Lemma 9, we have,

$$||\Sigma_K^{(i)} - \Sigma_K^{(j)}|| = ||\mathcal{T}_K^{(i)}(\Sigma_0) - \mathcal{T}_K^{(j)}(\Sigma_0)|| \leq 2 \left\| \mathcal{T}_K^{(i)} \right\|^2 \left\| \mathcal{F}_K^{(i)} - \mathcal{F}_K^{(j)} \right\| \| \Sigma_0 \|$$

$$\leq 4 \left( \frac{C_{\max}}{\sigma_{\min}(Q)\mu} \right)^2 (\epsilon_1 + \epsilon_2 ||K||) ||A - BK||_{\max} ||\Sigma_0||$$

where $\Sigma_0 = \mathbb{E}_{x_0^{(i)} \sim \mathcal{D}} \left[ x_0^{(i)} x_0^{(i)\top} \right].$

Thus, we have the following upper bound for $||\Sigma_K^{(i)} - \Sigma_K^{(j)}||$,

$$||\Sigma_K^{(i)} - \Sigma_K^{(j)}|| \leq g_2(\epsilon_1, \epsilon_2, K)$$

with,

$$g_2(\epsilon_1, \epsilon_2, K) := \epsilon_1 \left( \frac{C_{\max}(K)}{\sigma_{\min}(Q)\mu} \right)^2 (4||A - BK||_{\max} ||\Sigma_0||) + \epsilon_2 ||K|| \left( \frac{C_{\max}(K)}{\sigma_{\min}(Q)\mu} \right)^2 (4||A - BK||_{\max} ||\Sigma_0||).$$

Therefore, we can finally write an upper bound for $||\nabla C^{(i)}(K) - \nabla C^{(j)}(K)||$, which is:
$$||\nabla C^{(i)}(K) - \nabla C^{(j)}(K)|| \leq f(\epsilon_1, \epsilon_2, K)$$
where,

$$f(\epsilon_1, \epsilon_2, K) = 2(\beta_1 g_1(\epsilon_1, \epsilon_2, K) + \beta_2 g_2(\epsilon_1, \epsilon_2, K)).$$

After some rearrangement, we have that

$$f(\epsilon_1, \epsilon_2, K) = \epsilon_1 h_{\text{het}}^1(K) + \epsilon_1 h_{\text{het}}^2(K),$$

where $h_{\text{het}}^1 = h_{1f} + h_{2f}$ and $h_{\text{het}}^2 = h_{3f} + h_{4f}$, and

$$h_{1f} = \frac{2||B||_{\max}(C_{\max}(K))^2}{\sigma_{\min}(Q)\mu} \left[ 1 + 4 \left( \frac{C_{\max}(K)}{\sigma_{\min}(Q)\mu} \right) (||A - BK||_{\max})^2 \left( ||Q|| + ||R||||K||^2 \right) \right],$$

$$h_{2f} = \frac{2}{\mu} \left( \frac{C_{\max}(K)}{\sigma_{\min}(Q)} \right)^3 (4||A - BK||_{\max} ||\Sigma_0||),$$

$$h_{3f} = 2 \left( ||R||||K|| + \frac{||B||_{\max} C_{\max}(K)}{\mu} (||B||_{\max} + ||A||_{\max}) \right)$$
$$\times \left( \frac{||B||_{\max}||K|| C_{\max}(K)}{\mu} \left[ 1 + 4 \left( \frac{C_{\max}(K)}{\sigma_{\min}(Q)\mu} \right) (||A - BK||_{\max})^2 \left( ||Q|| + ||R||||K||^2 \right) \right] \right.$$
$$+ ||A - BK||_{\max} \bigg),$$

$$h_{4f} = 8 \left( ||R||||K|| + \frac{||B||_{\max} C_{\max}(K)}{\mu} (||B||_{\max} + ||A||_{\max}) \right) ||K|| \left( \frac{C_{\max}(K)}{\sigma_{\min}(Q)\mu} \right)^2 ||A - BK||_{\max} ||\Sigma_0||.$$

$\square$

### A.4.2   Proof of Theorem 1

In this theorem, we consider the setting where $(\epsilon_1 \bar{h}_{\text{het}}^1 + \epsilon_2 \bar{h}_{\text{het}}^2)^2 \leq \bar{h}_{\text{het}}^3$ with

$$\bar{h}_{\text{het}}^3 := \min_{j \in [M]} \left\{ \frac{\mu^2 \sigma_{\min}(R) \left( C^{(j)}(K_0) - C^{(j)}(K_j^*) \right)}{4 \|\Sigma_{K_j^*}\| \min\{n_x, n_u\}} \right\}. \tag{21}$$

**Outline:**   To prove Theorem 1, we first introduce some lemmas: Lemma 10 establishes stability of the local policies; Lemma 11 provides the drift analysis; Lemma 12 quantifies the per-round progress of our `FedLQR` algorithm. As a result, we are able to present the iterative stability guarantees and convergence analysis of `FedLQR` in the model-based setting.

**Lemma 10.** *(Stability of the local policies) Suppose $K_n \in \mathcal{G}^0$. If the local step-size satisfies $\eta_l \leq \min\{\frac{\underline{h}_\Delta}{\bar{h}_1}, \frac{1}{4\bar{h}_{grad}}\}$ and the heterogeneity level satisfies $(\epsilon_1 \bar{h}_{het}^1 + \epsilon_2 \bar{h}_{het}^2)^2 \leq \bar{h}_{het}^3$, then $K_{n,l}^{(i)} \in \mathcal{G}^0$ holds for all $i \in [M]$ and $l \in [L]$.*

*Proof.* Since $K_n \in \mathcal{G}^0$, based on the local Lipschitz property in Lemma 1, we have:

$$\begin{aligned}
C^{(j)}(K_{n,1}^{(i)}) - C^{(j)}(K_n) &\leq \left\langle \nabla C^{(j)}(K_n), K_{n,1}^{(i)} - K_n \right\rangle + \frac{h_{\text{grad}}(K_n)}{2} \left\| K_{n,1}^{(i)} - K_n \right\|_F^2 \\
&\leq - \left\langle \nabla C^{(j)}(K_n), \eta_l \nabla C^{(i)}(K_n) \right\rangle + \frac{h_{\text{grad}}(K_n)}{2} \left\| \eta_l \nabla C^{(i)}(K_n) \right\|_F^2
\end{aligned} \tag{22}$$

holds for any $i, j \in [M]$, if $\left\| \eta_l \nabla C^{(i)}(K_n) \right\|_F \leq \underline{h}_\Delta \leq h_\Delta(K_n)$, which holds when

$$\left\| \eta_l \nabla C^{(i)}(K_n) \right\|_F \overset{(a)}{\leq} \eta_l h_1(K_n) \leq \eta_l \bar{h}_1 \overset{(b)}{\leq} \underline{h}_\Delta,$$

where $(a)$ comes from Lemma 6 and $(b)$ holds because of the requirement on $\eta_l$ in the statement of the lemma. We note that although $h_{\text{grad}}(K_n)$ depends on $K_n$, our analysis is always restricted to the common stabilizing sublevel set $\mathcal{G}^0(\beta)$, on which, $h_{\text{grad}}(K_n)$ is bounded from above by some $\bar{h}_{\text{grad}} < \infty$ (see $(\phi, \rho)$-local smoothness property in Malik et al. (2019)).

Following the analysis in Eq (22), we have

$$\begin{aligned}
C^{(j)}(K_{n,1}^{(i)}) - C^{(j)}(K_n) \leq &-\eta_l \left\langle \nabla C^{(j)}(K_n), \nabla C^{(j)}(K_n) \right\rangle \\
&- \eta_l \underbrace{\left\langle \nabla C^{(j)}(K_n), \nabla C^{(i)}(K_n) - \nabla C^{(j)}(K_n) \right\rangle}_{T_1} \\
&+ \frac{h_{\text{grad}}(K_n)}{2} \left\| \eta_l \nabla C^{(i)}(K_n) \right\|_F^2.
\end{aligned} \tag{23}$$

Now $T_1$ can be bounded as

$$\begin{aligned}
T_1 &\leq \eta_l \left\| \nabla C^{(j)}(K_n) \right\|_F \left\| \nabla C^{(i)}(K_n) - \nabla C^{(j)}(K_n) \right\|_F \\
&\leq \eta_l \sqrt{\min\{n_x, n_u\}} \left\| \nabla C^{(j)}(K_n) \right\|_F \left\| \nabla C^{(i)}(K_n) - \nabla C^{(j)}(K_n) \right\| \\
&\overset{(c)}{\leq} \eta_l \sqrt{\min\{n_x, n_u\}} \left\| \nabla C^{(j)}(K_n) \right\|_F (\epsilon_1 \bar{h}_{\text{het}}^1 + \epsilon_2 \bar{h}_{\text{het}}^2),
\end{aligned} \tag{24}$$

where $(c)$ is due to Lemma 3. Plugging in the upper bound of $T_1$ into (22), we have:

$$\begin{aligned}
C^{(j)}(K_{n,1}^{(i)}) - C^{(j)}(K_n) \leq &-\eta_l \left\langle \nabla C^{(j)}(K_n), \nabla C^{(j)}(K_n) \right\rangle \\
&+ \eta_l \sqrt{\min\{n_x, n_u\}} \left\| \nabla C^{(j)}(K_n) \right\|_F (\epsilon_1 \bar{h}_{\text{het}}^1 + \epsilon_2 \bar{h}_{\text{het}}^2) + \frac{h_{\text{grad}}(K_n)}{2} \left\| \eta_l \nabla C^{(i)}(K_n) \right\|_F^2
\end{aligned}$$

$$\overset{(d)}{\leq} -\eta_l \left\langle \nabla C^{(j)}(K_n), \nabla C^{(j)}(K_n) \right\rangle + \eta_l \sqrt{\min\{n_x, n_u\}} \left\| \nabla C^{(j)}(K_n) \right\|_F (\epsilon_1 \bar{h}_{\text{het}}^1 + \epsilon_2 \bar{h}_{\text{het}}^2)$$

$$+ h_{\text{grad}}(K_n) \left\| \eta_l \nabla C^{(j)}(K_n) \right\|_F^2 + h_{\text{grad}}(K_n) \left\| \eta_l \nabla C^{(i)}(K_n) - \eta_l \nabla C^{(j)}(K_n) \right\|_F^2$$

$$\overset{(e)}{\leq} -\eta_l \left\langle \nabla C^{(j)}(K_n), \nabla C^{(j)}(K_n) \right\rangle + \eta_l \sqrt{\min\{n_x, n_u\}} \left\| \nabla C^{(j)}(K_n) \right\|_F (\epsilon_1 \bar{h}_{\text{het}}^1 + \epsilon_2 \bar{h}_{\text{het}}^2)$$

$$+ \eta_l^2 h_{\text{grad}}(K_n) \left\| \nabla C^{(j)}(K_n) \right\|_F^2 + \eta_l^2 h_{\text{grad}}(K_n) \min\{n_x, n_u\} (\epsilon_1 \bar{h}_{\text{het}}^1 + \epsilon_2 \bar{h}_{\text{het}}^2)^2$$

$$\overset{(f)}{\leq} -\eta_l \left\langle \nabla C^{(j)}(K_n), \nabla C^{(j)}(K_n) \right\rangle + \frac{\eta_l}{4} \left\| \nabla C^{(j)}(K_n) \right\|_F^2 + \eta_l \min\{n_x, n_u\} (\epsilon_1 \bar{h}_{\text{het}}^1 + \epsilon_2 \bar{h}_{\text{het}}^2)^2$$

$$+ \eta_l^2 \bar{h}_{\text{grad}} \left\| \nabla C^{(j)}(K_n) \right\|_F^2 + \eta_l^2 \bar{h}_{\text{grad}} \min\{n_x, n_u\} (\epsilon_1 \bar{h}_{\text{het}}^1 + \epsilon_2 \bar{h}_{\text{het}}^2)^2$$

$$= -\eta_l \left\langle \nabla C^{(j)}(K_n), \nabla C^{(j)}(K_n) \right\rangle + (\frac{\eta_l}{4} + \eta_l^2 \bar{h}_{\text{grad}}) \left\| \nabla C^{(j)}(K_n) \right\|_F^2$$

$$+ (\eta_l + \eta_l^2 \bar{h}_{\text{grad}}) \min\{n_x, n_u\} (\epsilon_1 \bar{h}_{\text{het}}^1 + \epsilon_2 \bar{h}_{\text{het}}^2)^2$$

$$\overset{(g)}{\leq} -\frac{\eta_l}{2} \left\| \nabla C^{(j)}(K_n) \right\|_F^2 + 2\eta_l \min\{n_x, n_u\} (\epsilon_1 \bar{h}_{\text{het}}^1 + \epsilon_2 \bar{h}_{\text{het}}^2)^2,$$

which implies

$$C^{(j)}(K_{n,1}^{(i)}) - C^{(j)}(K^*) \overset{(h)}{\leq} \left( 1 - \frac{2\eta_l \mu^2 \sigma_{\min}(R)}{\left\| \Sigma_{K_j^*} \right\|} \right) (C^{(j)}(K_0) - C^{(j)}(K_j^*))$$

$$+ 2\eta_l \min\{n_x, n_u\} (\epsilon_1 \bar{h}_{\text{het}}^1 + \epsilon_2 \bar{h}_{\text{het}}^2)^2, \tag{25}$$

where $(d)$ is due to Eq. (8); $(e)$ is due to Lemma 3; $(f)$ is due to Eq.(10) with $\zeta = \frac{1}{2}$; $(g)$ is due to the choice of step-size such that $\eta_l^2 \bar{h}_{\text{grad}} \leq \frac{\eta_l}{4}$, which holds when $\eta_l \leq \frac{1}{4\bar{h}_{\text{grad}}}$; and $(h)$ is due to Lemma 2 and the fact that $K_n \in \mathcal{G}^0$. If $\epsilon_1$ and $\epsilon_2$ are small enough that

$$(\epsilon_1 \bar{h}_{\text{het}}^1 + \epsilon_2 \bar{h}_{\text{het}}^2)^2 \leq \min_{j \in [M]} \left\{ \frac{\mu^2 \sigma_{\min}(R) \left( C^{(j)}(K_0) - C^{(j)}(K_j^*) \right)}{4 \|\Sigma_{K_j^*}\| \min\{n_x, n_u\}} \right\},$$

we have that

$$C^{(j)}(K_{n,1}^{(i)}) - C^{(j)}(K_n) \leq C^{(j)}(K_0) - C^{(j)}(K_j^*),$$

holds for any $j \in [M]$.

The above inequality implies $K_{n,1}^{(i)} \in \mathcal{G}^0$ as long as $K_n \in \mathcal{G}^0$. Then we can use the induction method to obtain that $K_{n,2}^{(i)} \in \mathcal{G}^0$ since $K_{n,1}^{(i)} \in \mathcal{G}^0$. As a result, an identical argument can be used from $K_{n,1}^{(i)}$ to $K_{n,2}^{(i)}$. Therefore, by repeating this step for $L$ times, we have that all the local polices $K_{n,l}^{(i)} \in \mathcal{G}^0$ holds for all $i \in [M]$ and $l = 1, \cdots, L$, when the global policy $K_n \in \mathcal{G}^0$. $\qquad \square$

**Lemma 11.** *(Drift term analysis) If $\eta_l \leq \min \left\{ \frac{1}{4\bar{h}_{grad}}, \frac{1}{2}, \frac{h_\Delta}{h_1}, \frac{\log 2}{L(3\bar{h}_{grad}+1)} \right\}$ and $K_n \in \mathcal{G}^0$, the difference between the local policy and global policy can be bounded as follows $\forall i \in [M]$ and $l \in [L]$:*

$$\left\| K_{n,l}^{(i)} - K_n \right\|_F^2 \leq 2\eta_l L \left\| \nabla C^{(i)}(K_n) \right\|_F^2 = \frac{2\eta}{\eta_g} \left\| \nabla C^{(i)}(K_n) \right\|_F^2.$$

*Proof.* We have

$$\left\| K_{n,l}^{(i)} - K_n \right\|_F^2 = \left\| K_{n,l-1}^{(i)} - K_n - \eta_l \nabla C^{(i)}(K_{n,l-1}^{(i)}) \right\|_F^2$$

$$= \left\|K_{n,l-1}^{(i)} - K_n\right\|_F^2 - 2\eta_l \left[\left\langle \nabla C^{(i)}(K_{n,l-1}^{(i)}), K_{n,l-1}^{(i)} - K_n \right\rangle\right]$$
$$+ \left\|\eta_l \nabla C^{(i)}(K_{n,l-1}^{(i)})\right\|_F^2$$
$$= \left\|K_{n,l-1}^{(i)} - K_n\right\|_F^2 - 2\eta_l \left[\left\langle \nabla C^{(i)}(K_{n,l-1}^{(i)}) - \nabla C^{(i)}(K_n), K_{n,l-1}^{(i)} - K_n \right\rangle\right]$$
$$- 2\eta_l \left[\left\langle \nabla C^{(i)}(K_n), K_{n,l-1}^{(i)} - K_n \right\rangle\right] + \left\|\eta_l \nabla C^{(i)}(K_{n,l-1}^{(i)})\right\|_F^2$$
$$\leq \left\|K_{n,l-1}^{(i)} - K_n\right\|_F^2 + 2\eta_l \left\|\nabla C^{(i)}(K_{n,l-1}^{(i)}) - \nabla C^{(i)}(K_n)\right\|_F \left\|K_{n,l-1}^{(i)} - K_n\right\|_F$$
$$+ 2\eta_l \left\|\nabla C^{(i)}(K_n)\right\|_F \left\|K_{n,l-1}^{(i)} - K_n\right\|_F + \left\|\eta_l \nabla C^{(i)}(K_{n,l-1}^{(i)})\right\|_F^2$$
$$\overset{(a)}{\leq} \left\|K_{n,l-1}^{(i)} - K_n\right\|_F^2 + 2\eta_l h_{\text{grad}}(K_n) \left\|K_{n,l-1}^{(i)} - K_n\right\|_F \left\|K_{n,l-1}^{(i)} - K_n\right\|_F$$
$$+ \eta_l \left\|\nabla C^{(i)}(K_n)\right\|_F^2 + \eta_l \left\|K_{n,l-1}^{(i)} - K_n\right\|_F^2 + \left\|\eta_l \nabla C^{(i)}(K_{n,l-1}^{(i)})\right\|_F^2$$
$$\leq \left(1 + 2\eta_l h_{\text{grad}}(K_n) + \eta_l\right) \left\|K_{n,l-1}^{(i)} - K_n\right\|_F^2 + \left(\eta_l + 2\eta_l^2\right) \left\|\nabla C^{(i)}(K_n)\right\|_F^2$$
$$+ 2\eta_l^2 \left\|\nabla C^{(i)}(K_{n,l-1}^{(i)}) - \nabla C^{(i)}(K_n)\right\|_F^2$$
$$\overset{(b)}{\leq} \left(1 + 2\eta_l h_{\text{grad}}(K_n) + \eta_l\right) \left\|K_{n,l-1}^{(i)} - K_n\right\|_F^2 + \left(\eta_l + 2\eta_l^2\right) \left\|\nabla C^{(i)}(K_n)\right\|_F^2$$
$$+ 2\eta_l^2 h_{\text{grad}}^2(K_n) \left\|K_{n,l-1}^{(i)} - K_n\right\|_F^2$$
$$\overset{(c)}{\leq} \left(1 + 2\eta_l \bar{h}_{\text{grad}} + \eta_l + 2\eta_l^2 \bar{h}_{\text{grad}}^2\right) \left\|K_{n,l-1}^{(i)} - K_n\right\|_F^2 + \left(\eta_l + 2\eta_l^2\right) \left\|\nabla C^{(i)}(K_n)\right\|_F^2$$
$$\overset{(d)}{\leq} \left(1 + 3\eta_l \bar{h}_{\text{grad}} + \eta_l\right) \left\|K_{n,l-1}^{(i)} - K_n\right\|_F^2 + 2\eta_l \left\|\nabla C^{(i)}(K_n)\right\|_F^2, \tag{26}$$

where $(a)$ and $(b)$ are due to Lemma 1; $(c)$ is due to the fact that $K_n \in \mathcal{G}^0$; $(d)$ is due to the choice of step-size such that $2\eta_l^2 \bar{h}_{\text{grad}}^2 \leq \eta_l \bar{h}_{\text{grad}}$ and $2\eta_l^2 \leq \eta_l$, which hold when $\eta_l \leq \min\{\frac{1}{2\bar{h}_{\text{grad}}}, \frac{1}{2}\}$. Therefore, we have

$$\left\|K_{n,l}^{(i)} - K_n\right\|_F^2 \leq \left(1 + 3\eta_l \bar{h}_{\text{grad}} + \eta_l\right) \left\|K_{n,l-1}^{(i)} - K_n\right\|_F^2 + 2\eta_l \left\|\nabla C^{(i)}(K_n)\right\|_F^2$$
$$\leq \left(1 + 3\eta_l \bar{h}_{\text{grad}} + \eta_l\right)^l \underbrace{\left\|K_{n,0}^{(i)} - K_n\right\|_F^2}_{=0} + 2\sum_{j=0}^{l-1} \left(1 + 3\eta_l \bar{h}_{\text{grad}} + \eta_l\right)^j \eta_l \left\|\nabla C^{(i)}(K_n)\right\|_F^2$$
$$\leq \frac{\left(1 + 3\eta_l \bar{h}_{\text{grad}} + \eta_l\right)^l - 1}{\left(1 + 3\eta_l \bar{h}_{\text{grad}} + \eta_l\right) - 1} 2\eta_l \left\|\nabla C^{(i)}(K_n)\right\|_F^2$$
$$\overset{(a)}{\leq} 2 \times \frac{1 + l(3\eta_l \bar{h}_{\text{grad}} + \eta_l) - 1}{3\bar{h}_{\text{grad}} + 1} \left\|\nabla C^{(i)}(K_n)\right\|_F^2$$
$$\leq 2\eta_l L \left\|\nabla C^{(i)}(K_n)\right\|_F^2,$$

where, for $(a)$, we used the fact that $(1 + x)^{\tau+1} \leq 1 + 2x(\tau + 1)$ holds for $x \leq \frac{\log 2}{\tau}$. In other words, $\left(1 + 3\eta_l \bar{h}_{\text{grad}} + \eta_l\right)^l \leq 1 + l(3\eta_l \bar{h}_{\text{grad}} + \eta_l)$ holds when $3\eta_l \bar{h}_{\text{grad}} + \eta_l \leq \frac{\log 2}{l}$, i.e., when $\eta_l \leq \frac{\log 2}{L(3\bar{h}_{\text{grad}}+1)}$. $\qquad \square$

**Lemma 12.** *(Per round progress) Suppose $K_n \in \mathcal{G}^0$. If we choose the local step-size as*

$$\eta_l = \frac{1}{2} \min \left\{ \frac{1}{4\bar{h}_{grad}}, \frac{1}{2}, \frac{\underline{h}_\Delta}{\bar{\bar{h}}_1}, \frac{\log 2}{L(3\bar{h}_{grad} + 1)}, \frac{1}{80L\bar{h}_{grad}^2} \right\},$$

*choose* $\eta = \frac{1}{2}\min\{\frac{h_\Delta}{\bar{h}_1}, 1, \frac{2}{3\bar{h}_{grad}}\}$, *and the global step-size as* $\eta_g = \frac{\eta}{L\eta_l}$, *then, for all* $i \in [M]$, *it holds that*

$$C^{(i)}(K_{n+1}) - C^{(i)}(K_n) \leq -\frac{\eta\mu^2\sigma_{\min}(R)}{\left\|\Sigma_{K_i^*}\right\|_{\max}}(C^{(i)}(K_n) - C^{(i)}(K_i^*)) + 3\eta\min\{n_x, n_u\}(\epsilon_1\bar{h}_{het}^1 + \epsilon_2\bar{h}_{het}^2)^2. \quad (27)$$

*Proof.*

$$C^{(i)}(K_{n+1}) - C^{(i)}(K_n) \overset{(a)}{\leq} \langle\nabla C^{(i)}(K_n), K_{n+1} - K_n\rangle + \frac{h_{\mathrm{grad}}(K_n)}{2}\|K_{n+1} - K_n\|_F^2$$

$$= -\left\langle\nabla C^{(i)}(K_n), \frac{\eta}{ML}\sum_{j=1}^{M}\sum_{l=0}^{L-1}\nabla C^{(j)}(K_{n,l}^{(j)})\right\rangle + \frac{h_{\mathrm{grad}}(K_n)}{2}\left\|\frac{\eta}{ML}\sum_{j=1}^{M}\sum_{l=0}^{L-1}\nabla C^{(j)}(K_{n,l}^{(j)})\right\|_F^2$$

$$= -\left\langle\nabla C^{(i)}(K_n), \frac{\eta}{ML}\sum_{j=1}^{M}\sum_{l=0}^{L-1}\left[\nabla C^{(j)}(K_{n,l}^{(j)}) - \nabla C^{(i)}(K_n)\right]\right\rangle - \eta\left\|\nabla C^{(i)}(K_n)\right\|_F^2$$

$$+ \frac{h_{\mathrm{grad}}(K_n)}{2}\left\|\frac{\eta}{ML}\sum_{j=1}^{M}\sum_{l=0}^{L-1}\nabla C^{(j)}(K_{n,l}^{(j)})\right\|_F^2$$

$$= -\left\langle\nabla C^{(i)}(K_n), \frac{\eta}{ML}\sum_{j=1}^{M}\sum_{l=0}^{L-1}\left[\nabla C^{(j)}(K_{n,l}^{(j)}) - \nabla C^{(j)}(K_n) + \nabla C^{(j)}(K_n) - \nabla C^{(i)}(K_n)\right]\right\rangle$$

$$- \eta\left\|\nabla C^{(i)}(K_n)\right\|_F^2 + \frac{h_{\mathrm{grad}}(K_n)}{2}\left\|\frac{\eta}{ML}\sum_{j=1}^{M}\sum_{l=0}^{L-1}\nabla C^{(j)}(K_{n,l}^{(j)})\right\|_F^2$$

$$\leq \eta\left\|\nabla C^{(i)}(K_n)\right\|_F\left\|\frac{1}{ML}\sum_{j=1}^{M}\sum_{l=0}^{L-1}\left[\nabla C^{(j)}(K_{n,l}^{(j)}) - \nabla C^{(j)}(K_n)\right]\right\|_F$$

$$+ \frac{\eta}{M}\sum_{j=1}^{M}\left\|\nabla C^{(i)}(K_n)\right\|_F\left\|\nabla C^{(j)}(K_n) - \nabla C^{(i)}(K_n)\right\|_F$$

$$- \eta\left\|\nabla C^{(i)}(K_n)\right\|_F^2 + \frac{h_{\mathrm{grad}}(K_n)}{2}\left\|\frac{\eta}{ML}\sum_{j=1}^{M}\sum_{l=0}^{L-1}\nabla C^{(j)}(K_{n,l}^{(j)})\right\|_F^2$$

$$\overset{(b)}{\leq} \eta\left\|\nabla C^{(i)}(K_n)\right\|_F\left[\frac{h_{\mathrm{grad}}(K_n)}{ML}\sum_{j=1}^{M}\sum_{l=0}^{L-1}\left\|K_{n,l}^{(j)} - K_n\right\|_F\right]$$

$$+ \frac{\eta}{4}\left\|\nabla C^{(i)}(K_n)\right\|_F^2 + \frac{\eta}{M}\sum_{j=1}^{M}\left\|\nabla C^{(j)}(K_n) - \nabla C^{(i)}(K_n)\right\|_F^2 - \eta\left\|\nabla C^{(i)}(K_n)\right\|_F^2$$

$$+ h_{\mathrm{grad}}(K_n)\frac{3\eta^2}{2ML}\sum_{j=1}^{M}\sum_{l=0}^{L-1}\left\|\nabla C^{(j)}(K_{n,l}^{(j)}) - \nabla C^{(j)}(K_n)\right\|_F^2$$

$$+ \frac{3\eta^2 h_{\mathrm{grad}}(K_n)}{2M}\sum_{j=1}^{M}\left\|\nabla C^{(j)}(K_n) - \nabla C^{(i)}(K_n)\right\|_F^2 + \frac{3\eta^2 h_{\mathrm{grad}}(K_n)}{2}\left\|\nabla C^{(i)}(K_n)\right\|_F^2$$

$$\overset{(c)}{\leq} \frac{\eta}{4}\left\|\nabla C^{(i)}(K_n)\right\|_F^2 + \frac{\eta\bar{h}_{\mathrm{grad}}^2}{ML}\sum_{i=1}^{M}\sum_{l=0}^{L-1}\left\|K_{n,l}^{(i)} - K_n\right\|_F^2$$

$$+ \frac{\eta}{4}\left\|\nabla C^{(i)}(K_n)\right\|_F^2 + \left(\eta + \frac{3\eta^2\bar{h}_{\mathrm{grad}}}{2}\right)\frac{1}{M}\sum_{j=1}^{M}\left\|\nabla C^{(j)}(K_n) - \nabla C^{(i)}(K_n)\right\|_F^2$$

$$- \eta\left\|\nabla C^{(i)}(K_n)\right\|_F^2 + \frac{3\eta^2\bar{h}_{\mathrm{grad}}^2}{2ML}\sum_{j=1}^{M}\sum_{l=0}^{L-1}\left\|K_{n,l}^{(j)} - K_n\right\|_F^2 + \frac{\eta}{8}\left\|\nabla C^{(i)}(K_n)\right\|_F^2$$

$$\overset{(d)}{\leq} -\frac{3\eta}{8}\left\|\nabla C^{(i)}(K_n)\right\|_F^2 + \frac{5\eta^2 \bar{h}_{\text{grad}}^2}{\eta_g M}\sum_{j=1}^{M}\left\|\nabla C^{(j)}(K_n)\right\|_F^2$$

$$+ 2\eta\min\{n_x, n_u\}(\epsilon_1\bar{h}_{\text{het}}^1 + \epsilon_2\bar{h}_{\text{het}}^2)^2$$

$$\overset{(e)}{\leq} -\frac{3\eta}{8}\left\|\nabla C^{(i)}(K_n)\right\|_F^2 + \frac{10\eta^2 \bar{h}_{\text{grad}}^2}{\eta_g M}\sum_{j=1}^{M}\left\|\nabla C^{(j)}(K_n) - \nabla C^{(i)}(K_n)\right\|_F^2$$

$$+ \frac{10\eta^2 \bar{h}_{\text{grad}}^2}{\eta_g}\left\|\nabla C^{(i)}(K_n)\right\|_F^2 + 2\eta\min\{n_x, n_u\}(\epsilon_1\bar{h}_{\text{het}}^1 + \epsilon_2\bar{h}_{\text{het}}^2)^2$$

$$\overset{(f)}{\leq} -\frac{\eta}{4}\left\|\nabla C^{(i)}(K_n)\right\|_F^2 + 3\eta\min\{n_x, n_u\}(\epsilon_1\bar{h}_{\text{het}}^1 + \epsilon_2\bar{h}_{\text{het}}^2)^2$$

$$\leq -\frac{\eta\mu^2\sigma_{\min}(R)}{\left\|\Sigma_{K_i^*}\right\|}(C^{(i)}(K_n) - C^{(i)}(K_i^*)) + 3\eta\min\{n_x, n_u\}(\epsilon_1\bar{h}_{\text{het}}^1 + \epsilon_2\bar{h}_{\text{het}}^2)^2.$$

In the above steps, (a) is due to the choice of step-size $\eta$ such that

$$\|K_{n+1} - K_n\| = \|\frac{\eta}{ML}\sum_{i=1}^{M}\sum_{l=0}^{L-1}\nabla C^{(i)}(K_{n,l}^{(i)})\| \leq \eta\bar{h}_1 \leq \underline{h}_\Delta,$$

holds when $\eta \leq \frac{h_\Delta}{\bar{h}_1}$. For $(b)$, we use the Lipschitz property of the gradient (Lemma 1) in the first line, and use Eq.(10) with $\zeta = \frac{1}{2}$ in the second line, and for the third and forth lines we use Eq. (8); $(c)$ is due to Lemma 1 and $\frac{3\eta^2\bar{h}_{\text{grad}}}{2} \leq \frac{\eta}{8}$; $(d)$ is due to Lemma 3, Lemma 11 and the choice of step-size such that $\frac{3\eta^2\bar{h}_{\text{grad}}}{2} \leq \frac{\eta}{8} \leq \eta$; $(e)$ is due to Eq.(8); and for $(f)$, we use the fact that $\frac{10\eta^2\bar{h}_{\text{grad}}^2}{\eta_g} \leq \frac{\eta}{8} \leq \eta$, which holds when $\eta_l \leq \frac{1}{80L\bar{h}_{\text{grad}}^2}$. We use the gradient domination property (Lemma 2) in the last equality. □

With this lemma, we are now ready to provide the convergence guarantees for `FedLQR` in the model-based setting.

**Proof of the iterative stability guarantees of `FedLQR`:** Here we leverage the method of induction to prove `FedLQR`'s iterative stability guarantees. First, we start from an initial policy $K_0 \in \mathcal{G}^0$. At round $n$, we assume $K_n \in \mathcal{G}^0$. According to Lemma 10, we can show that all the local policies $K_{n,l}^{(i)} \in \mathcal{G}^0$. Furthermore, by choosing the step-sizes properly in Lemma 12, we have that

$$C^{(i)}(K_{n+1}) - C^{(i)}(K_n) \leq -\frac{\eta\mu^2\sigma_{\min}(R)}{\left\|\Sigma_{K_i^*}\right\|_{\max}}(C^{(i)}(K_n) - C^{(i)}(K_i^*))$$
$$+ 3\eta\min\{n_x, n_u\}(\epsilon_1\bar{h}_{\text{het}}^1 + \epsilon_2\bar{h}_{\text{het}}^2)^2.$$

for any $i \in [M]$. Then, for any fixed system $i \in [M]$, with $(\epsilon_1\bar{h}_{\text{het}}^1 + \epsilon_2\bar{h}_{\text{het}}^2)^2 \leq \bar{h}_{\text{het}}^3$, we have that

$$C^{(i)}(K_{n+1}) - C^{(i)}(K_i^*) \leq \left(1 - \frac{\eta\mu^2\sigma_{\min}(R)}{\left\|\Sigma_{K_i^*}\right\|}\right)(C^{(i)}(K_n) - C^{(i)}(K_i^*))$$
$$+ 3\eta\min\{n_x, n_u\}(\epsilon_1\bar{h}_{\text{het}}^1 + \epsilon_2\bar{h}_{\text{het}}^2)^2$$
$$\leq \left(1 - \frac{\eta\mu^2\sigma_{\min}(R)}{\left\|\Sigma_{K_i^*}\right\|}\right)(C^{(i)}(K_0) - C^{(i)}(K_i^*))$$
$$+ \frac{\eta\mu^2\sigma_{\min}(R)}{\left\|\Sigma_{K_i^*}\right\|}(C^{(i)}(K_0) - C^{(i)}(K_i^*))$$
$$\leq C^{(i)}(K_0) - C^{(i)}(K_i^*).$$

With this, we can easily see that the global policy $K_{n+1}$ at the next round $n + 1$ is also stabilizing, i.e., $K_{n+1} \in \mathcal{G}^0$, by using the definition of $\mathcal{G}^0$ (Definition 2). Therefore, we can complete proving FedLQR's iterative stability property by inductive reasoning.

**Proof of FedLQR's convergence:** From Eq.(27), we have

$$
C^{(i)}(K_{n+1}) - C^{(i)}(K_i^*) \leq \left( 1 - \frac{\eta \mu^2 \sigma_{\min}(R)}{\left\| \Sigma_{K_i^*} \right\|} \right) (C^{(i)}(K_n) - C^{(i)}(K_i^*))
$$
$$
+ 3\eta \min\{n_x, n_u\} (\epsilon_1 \bar{h}_{\mathrm{het}}^1 + \epsilon_2 \bar{h}_{\mathrm{het}}^2)^2.
$$

Under the assumptions in Lemma 12, FedLQR thus enjoys the following convergence guarantee after $N$ rounds:

$$
C^{(i)}(K_N) - C^{(i)}(K_i^*) \leq \left( 1 - \frac{\eta \mu^2 \sigma_{\min}(R)}{\left\| \Sigma_{K_i^*} \right\|} \right)^N (C^{(i)}(K_0) - C^{(i)}(K_i^*))
$$
$$
+ \frac{3 \min\{n_x, n_u\} \left\| \Sigma_{K_i^*} \right\|}{\mu^2 \sigma_{\min}(R)} (\epsilon_1 h_{\mathrm{het}}^1 + \epsilon_1 h_{\mathrm{het}}^2)^2.
$$

Thus, we finish the proof of Theorem 1 with $c_{\mathrm{uni},1} = 12$ and $\mathcal{B}(\epsilon_1, \epsilon_2) := \frac{v \left\| \Sigma_{K_i^*} \right\|}{4\mu^2 \sigma_{\min}(R)} (\epsilon_1 h_{\mathrm{het}}^1 + \epsilon_1 h_{\mathrm{het}}^2)^2.$ □

### A.4.3 Proof of Corollary 8.3

*Proof.* From Theorem 1, we have that

$$
C^{(i)}(K_N) - C^{(i)}(K_i^*) \leq \left( 1 - \frac{\eta \mu^2 \sigma_{\min}(R)}{\left\| \Sigma_{K_i^*} \right\|} \right)^N (C^{(i)}(K_0) - C^{(i)}(K_i^*))
$$
$$
+ \frac{3 \min\{n_x, n_u\} \left\| \Sigma_{K_i^*} \right\|}{\mu^2 \sigma_{\min}(R)} (\epsilon_1 h_{\mathrm{het}}^1 + \epsilon_1 h_{\mathrm{het}}^2)^2
$$

holds for all $i \in [M]$. Based on the fact that $K_i^*$ is system $i$'s optimal LQR controller, i.e., $C^{(i)}(K_i^*) \leq C^{(i)}(K^*)$, we have that

$$
C^{(i)}(K_N) - C^{(i)}(K^*) \leq \left( 1 - \frac{\eta \mu^2 \sigma_{\min}(R)}{\left\| \Sigma_{K_i^*} \right\|} \right)^N (C^{(i)}(K_0) - C^{(i)}(K_i^*))
$$
$$
+ \frac{3 \min\{n_x, n_u\} \left\| \Sigma_{K_i^*} \right\|}{\mu^2 \sigma_{\min}(R)} (\epsilon_1 h_{\mathrm{het}}^1 + \epsilon_1 h_{\mathrm{het}}^2)^2 \tag{28}
$$

We can finish the proof of Corollary 1 by averaging Eq. (28) across all $M$ systems. □

### A.5 Zeroth-order optimization

To prepare for the model-free setting where the controllers only have access to the system's trajectories, we first quickly recap the basic idea behind zeroth-order optimization. Say our goal is to minimize a loss function $f(x)$, where $x \in \mathbb{R}^d$. When one has access to exact deterministic gradients of this loss function via an oracle, the standard approach for minimization would be to query the gradient oracle at each iteration, and run gradient descent. Concretely, one would run the following iterative scheme: $x_{t+1} = x_t - \eta \nabla f(x_t)$, where $\eta$ is a suitably chosen learning-rate/step-size. While such first-order optimization schemes have a rich history, there has also been a growing interest in understanding the behavior of derivative-free (zeroth-order) methods that can *only query function values*, as opposed to the gradients. Two immediate reasons (among many) for studying zeroth-order optimization are as follows: (i) in practice, one may only have access to a black-box procedure that cannot evaluate gradients; and (ii) computing gradients might prove to be too computationally-expensive.

Given two or more function evaluations, the basic idea behind zeroth-order algorithms is to construct an estimate of the true gradient for evaluating and updating model parameters. For instance, a typical zeroth-order scheme with single-point function evaluation would take the following form (Polyak, 1987):

$$x_{t+1} = x_t - \eta_t \left( \frac{f(x_t + \mu_t u) - f(x_t)}{\mu_t} \right) u.$$

In the expression above, $\{\eta_t\}$ is the learning-rate sequence, $\{\mu_t\}$ is a sequence typically chosen in a way such that $\mu_t \to 0$, and $u$ is a random vector distributed uniformly over the unit sphere. For details about the convergence of zeroth-order optimization algorithms such as the one above, we refer the interested reader to (Nesterov & Spokoiny, 2017; Duchi et al., 2015; Bach & Perchet, 2016).

We now turn to briefly describing the model-free setup for our LQR problem. (Fazel et al., 2018) propose a zeroth-order-based algorithm (Algorithm 1 in (Fazel et al., 2018)) to compute an estimation $\widehat{\nabla C(K)}$ and $\widehat{\Sigma_K}$ for both $\nabla C(K)$ and $\Sigma_K$, for a given $K$. Algorithm 1 in (Fazel et al., 2018) exploits a multiple-trajectory-based technique that uses a Gaussian perturbed cost function (i.e., producing a Gaussian smoothing function) to estimate $\nabla C(K)$ from cost function perturbed values. That is, given the cost function $C(K)$, we can define its perturbed function as,

$$C_r(K) = \mathbb{E}_{U \sim \mathbb{B}_r}[C(K + U)]$$

where $\mathbb{B}_r$ is the uniform distribution over all matrices with Frobenius norm at most $r$ and $U$ is a random matrix with proper dimension and generated from $\mathbb{B}_r$. For small $r$, the smooth cost $C_r(K)$ is a good approximation to the original cost $C(K)$. Due to the Gaussian smoothing, the gradient has a particularly simple functional form (Gravell et al., 2020):

$$\nabla C_r(K) = \frac{n_x n_u}{r^2} \mathbb{E}_{U \sim \mathbb{B}_r}[C(K + U)U].$$

Therefore, this expression implies a straightforward method to obtain an unbiased estimate of $\nabla C_r(K)$, through obtaining the infinite-horizon rollouts. However, in practice, we can only obtain the finite-horizon rollouts to approximate the gradient. Thanks to (Fazel et al., 2018), they showed that the approximation error of the exact gradient can be reduced to arbitrary accuracy if the number of sample trajectories $n_s$ and the length of each rollout $\tau$ are sufficiently large, and the smoothing radius $r$ is small enough.

We also emphasize that the zeroth-order estimator obtained from $C_r(K)$ is a biased approximation of the true gradient $\nabla C(K)$, with the bias arising from the smoothing step. The magnitude of this bias is controlled by the choice of the smoothing radius $r$: smaller values of $r$ yield a closer approximation to the original cost landscape at the cost of higher variance. In our analysis, we make the estimation error sufficiently small by selecting $r$, the number of samples $n_s$, and the rollout horizon $\tau$ appropriately.

It is important to emphasize that our results can be readily extended to the two-point zeroth-order estimation scheme, where the estimation variance can be reduced, as discussed in Malik et al. (2019). However, addressing the two-point zeroth-order estimation is beyond the scope of this work, and we refer the reader to (Toso et al., 2024a) for an instantiation of the results in the asynchronous aggregation setting under the two-point gradient estimation scheme.

## A.6   The model-free setting

For notational brevity we rewrite $\widehat{\nabla C^{(i)}(K)}$ as $\tilde{\nabla} C^{(i)}(K)$ where

$$\widehat{\nabla C^{(i)}(K)} = \tilde{\nabla} C^{(i)}(K) := \frac{1}{n_s} \sum_{s=1}^{n_s} \frac{n_x n_u}{r^2} \tilde{C}^{(i),(\tau)} \left( K + U_s^{(i)} \right) U_s^{(i)},$$

and introduce two new gradient-based terms:

$$\nabla' C^{(i)}(K) := \frac{1}{n_s} \sum_{s=1}^{n_s} \frac{n_x n_u}{r^2} C^{(i),(\tau)} \left( K + U_s^{(i)} \right) U_s^{(i)},$$

$$\hat{\nabla} C^{(i)}(K) := \frac{1}{n_s} \sum_{s=1}^{n_s} \frac{n_x n_u}{r^2} C^{(i)} \left( K + U_s^{(i)} \right) U_s^{(i)},$$

where $\tilde{C}^{(i),(\tau)} \left( K + U_s^{(i)} \right) := \sum_{t=0}^{\tau-1} \left( x_t^{(i)\top} Q x_t^{(i)} + u_t^{(i)\top} R u_t^{(i)} \right)$ with $x_t^{(i)} = (K + U_s^{(i)}) u_t^{(i)}$, $C^{(i),(\tau)} \left( K + U_s^{(i)} \right) := \mathbb{E}_{x_0^{(i)} \sim \mathcal{D}} \sum_{t=0}^{\tau-1} \left( x_t^{(i)\top} Q x_t^{(i)} + u_t^{(i)\top} R u_t^{(i)} \right)$ and

$$C^{(i)} \left( K + U_s^{(i)} \right) := \mathbb{E}_{x_0^{(i)} \sim \mathcal{D}} \sum_{t=0}^{\infty} \left( x_t^{(i)\top} Q x_t^{(i)} + u_t^{(i)\top} R u_t^{(i)} \right).$$

### A.6.1   Auxiliary Lemmas

**Lemma 13.** *(Approximating $C^{(i)}(K)$ and $\Sigma_K^{(i)}$ with finite horizon) Suppose $K$ is such that $C^{(i)}(K)$ is finite. Define the finite horizon estimates,*

$$\Sigma_K^{(i),(\tau)} := \mathbb{E} \left[ \sum_{t=0}^{\tau-1} x_t^{(i)} x_t^{(i)\top} \right] \quad and \quad C^{(i),(\tau)}(K) := \mathbb{E} \left[ \sum_{t=0}^{\tau-1} x_t^{(i)\top} Q x_t^{(i)} + u_t^{(i)\top} R u_t^{(i)} \right],$$

*for all systems $i \in [M]$. Now, let $\epsilon$ be an arbitrarily small constant such that*

$$\tau \geq h_\tau^1(\epsilon) := \max_{i \in [M]} \left\{ \frac{n_x \cdot (C^{(i)}(K))^2}{\epsilon \mu (\sigma_{min}(Q))^2} \right\} = \frac{n_x \cdot (C_{\max}(K))^2}{\epsilon \mu (\sigma_{min}(Q))^2},$$

*such that*

$$\left\| \Sigma_K^{(i),(\tau)} - \Sigma_K^{(i)} \right\| \leq \epsilon.$$

*If*

$$\tau \geq h_\tau^2(\epsilon) := \max_{i \in [M]} \left\{ \frac{n_x \cdot (C^{(i)}(K))^2 (\|Q\| + \|R\| \|K\|^2)}{\epsilon \mu (\sigma_{min}(Q))^2} \right\}$$

$$= \frac{n_x \cdot (C_{\max}(K))^2 (\|Q\| + \|R\| \|K\|^2)}{\epsilon \mu (\sigma_{min}(Q))^2},$$

*we have*

$$\left| C^{(i),(\tau)}(K) - C^{(i)}(K) \right| \leq \epsilon,$$

*where $C_{\max}(K) := \max_{i \in [M]} C^{(i)}(K)$.*

*Proof.* The proof for this lemma is detailed in (Fazel et al., 2018, Lemma 23). □

**Lemma 14.** *(Estimating $\nabla C^{(i)}(K)$ with finitely many infinite-horizon rollouts) Given an arbitrary tolerance $\epsilon$ and probability $\delta$, suppose the radius $r$ satisfies*

$$r \leq h_r\left(\frac{\epsilon}{2}\right) := \min\left\{\underline{h}_\Delta, \frac{\bar{C}_{\max}}{\bar{h}_{cost}}, \frac{\epsilon}{2\bar{h}_{grad}}\right\},$$

*and the number of samples $n_s$ satisfies,*

$$n_s \geq h_{sample}\left(\frac{\epsilon}{2}, \delta\right) := \frac{8\sigma_{\hat{\nabla}}^2 \min(n_x, n_u)}{\epsilon^2} \log\left[\frac{n_x + n_u}{\delta}\right]$$

$$\sigma_{\hat{\nabla}}^2 := \left(\frac{2n_x n_u \bar{C}_{\max}}{r}\right)^2 + \left(\frac{\epsilon}{2} + \bar{h}_1\right)^2$$

*Then with a high probability of at least $1 - \delta$, the estimate*

$$\hat{\nabla}C^{(i)}(K) = \frac{1}{n_s} \sum_{s=1}^{n_s} \frac{n_x n_u}{r^2} C^{(i)}\left(K + U_s^{(i)}\right) U_s^{(i)}$$

*satisfies*

$$\|\hat{\nabla}C^{(i)}(K) - \nabla C^{(i)}(K)\|_F \leq \epsilon$$

*for any system $i \in [M]$ and $K \in \mathcal{G}^0$.*

*Proof.* The proof for this lemma is detailed in Lemma B.6 of (Gravell et al., 2020). It is worthwhile to mention that, in (Gravell et al., 2020), the number of samples $n_s$ satisfies

$$n_s \geq \left[\underbrace{\frac{8\sigma_{\hat{\nabla}}^2 \min(n_x, n_u)}{\epsilon^2}}_{T_1} + \underbrace{\frac{8\min(n_x, n_u)}{\epsilon^2} \frac{R_{\hat{\nabla}}\epsilon}{6\sqrt{\min(n_x, n_u)}}}_{T_2}\right] \log\left[\frac{n_x + n_u}{\delta}\right]$$

with $R_{\hat{\nabla}} = \frac{2n_x n_u \bar{C}_{\max}}{r} + \frac{\epsilon}{2} + \bar{h}_1$. In the analysis throughout the paper, we only keep the dominant term $T_1$ in $n_s$, since $T_1$ is in the order $\mathcal{O}(\epsilon^{-2})$ while $T_2$ is in the order $\mathcal{O}(\epsilon^{-1})$.

By taking the maximum over $K$ inside $\mathcal{G}^0$, we make the local parameters become the global parameters, e.g., $\bar{C}_{\max} := \sup_{K \in \mathcal{G}^0, i \in [M]} C^{(i)}(K)$. $\qquad\square$

**Lemma 15.** *(Estimating $\nabla C^{(i)}(K)$ with finitely many finite-horizon rollouts): Given an arbitrary tolerance $\epsilon$ and probability $\delta$, suppose that the smoothing radius $r$ satisfies,*

$$r \leq h_r\left(\frac{\epsilon}{4}\right) = \min\left\{\bar{h}_\Delta, \frac{\bar{C}_{\max}}{\bar{h}_{cost}}, \frac{\epsilon}{4\bar{h}_{grad}}\right\},$$

*and the trajectory length $\tau$ satisfies*

$$\tau \geq h_\tau\left(\frac{r\epsilon}{4n_x n_u}\right) = \frac{4n_u n_x^2 (C_{\max}(K))^2 \left(\|Q\| + \|R\|\|K\|^2\right)}{r\epsilon\mu\sigma_{\min}(Q)^2}.$$

*According to Assumption 1, the distribution of the initial states satisfies $x_0^{(i)} \sim \mathcal{D}$ and $\left\|x_0^{(i)}\right\| \leq H$ almost surely. Thus, for any given realization $x_{0,s}^{(i)}$[17] of $x_0^{(i)}$, and for any system $i \in [M]$, we have*

$$\left\|x_{0,s}^{(i)}\right\| \leq H, \quad \left(x_{0,s}^{(i)}\right)\left(x_{0,s}^{(i)}\right)^\top \preceq \frac{H^2}{\mu}\mathbb{E}\left[x_0^{(i)} x_0^{(i)\top}\right].$$

---

[17]The notation $x_{0,s}^{(i)}$ denotes $s$-th sample of the initial state from $i$-th system.

*As a result, the summation over the finite-time horizon*

$$\sum_{t=0}^{\tau-1} \left( x_{t,j}^{(i)\top} Q x_{t,j}^{(i)} + u_{t,j}^{(i)\top} R u_{t,j}^{(i)} \right) \leq \frac{H^2}{\mu} C^{(i)} \left( K + U_j^{(i)} \right).$$

*Furthermore, suppose the number of samples $n_s$ satisfies*

$$n_s \geq h_{sample,trunc} \left( \frac{\epsilon}{4}, \delta, \frac{H^2}{\mu} \right) := \frac{32 \sigma_{\tilde{\nabla}}^2 \min(n_x, n_u)}{\epsilon^2} \log \left[ \frac{n_x + n_u}{\delta} \right],$$

*where*

$$\sigma_{\tilde{\nabla}}^2 := \left( \frac{2 n_x n_u H^2 \bar{C}_{\max}}{r\mu} \right)^2 + \left( \frac{\epsilon}{2} + \bar{h}_1 \right)^2,$$

*then, with a high probability of at least $1 - \delta$, the estimated gradient*

$$\tilde{\nabla} C^{(i)}(K) := \frac{1}{n_s} \sum_{s=1}^{n_s} \frac{n_u n_x}{r^2} \tilde{C}^{(i),(\tau)} \left( K + U_s^{(i)} \right) U_s^{(i)}$$

*satisfies*

$$\|\tilde{\nabla} C^{(i)}(K) - \nabla C^{(i)}(K)\|_F \leq \epsilon$$

*for any system $i \in [M]$ and $K \in \mathcal{G}^0$.*

*Proof.* The proof for this lemma is detailed in Lemma B.7 of (Gravell et al., 2020). As in Lemma 14, we only keep the dominant term in the requirement of sample size $n_s$. By taking the maximum over $K$ inside $\mathcal{G}^0$, all the local parameters inside the polynomials such as $h_r(\frac{\epsilon}{4})$ become global parameters. $\square$

### A.6.2   Proof of Lemma 4

*Proof.* For our subsequent analysis, we will use $\mathcal{F}_l^n$ to denote the filtration that captures all the randomness up to the $l$-th local step in round $n$. We have

$$\left\| \frac{1}{ML} \sum_{i=1}^{M} \sum_{l=0}^{L-1} \left[ \widehat{\nabla C^{(i)}(K_{n,l}^{(i)})} - \nabla C^{(i)}(K_{n,l}^{(i)}) \right] \right\|_F$$

$$= \left\| \frac{1}{ML} \sum_{i=1}^{M} \sum_{l=0}^{L-1} \left[ \tilde{\nabla} C^{(i)}(K_{n,l}^{(i)}) - \nabla C^{(i)}(K_{n,l}^{(i)}) \right] \right\|_F$$

$$\leq \underbrace{\left\| \frac{1}{ML} \sum_{i=1}^{M} \sum_{l=0}^{L} \left[ \tilde{\nabla} C^{(i)}(K_{n,l}^{(i)}) - \nabla' C^{(i)}(K_{n,l}^{(i)}) \right] \right\|_F}_{T_1}$$

$$+ \underbrace{\left\| \frac{1}{ML} \sum_{i=1}^{M} \sum_{l=0}^{L-1} \left[ \nabla' C^{(i)}(K_{n,l}^{(i)}) - \hat{\nabla} C^{(i)}(K_{n,l}^{(i)}) \right] \right\|_F}_{T_2}$$

$$+ \underbrace{\left\| \frac{1}{ML} \sum_{i=1}^{M} \sum_{l=0}^{L-1} \left[ \hat{\nabla} C^{(i)}(K_{n,l}^{(i)}) - \nabla C^{(i)}(K_{n,l}^{(i)}) \right] \right\|_F}_{T_3}.$$

Next, we will bound $T_1$, $T_2$, and $T_3$, respectively.

**Bounding $T_2$:** From the proof of Lemma B.7 in (Gravell et al., 2020), we have

$$T_2 \leq \frac{1}{ML} \sum_{i=1}^{M} \sum_{l=0}^{L-1} \left\| \nabla' C^{(i)}(K_{n,l}^{(i)}) - \hat{\nabla} C^{(i)}(K_{n,l}^{(i)}) \right\|_F \leq \frac{\epsilon}{4} \tag{29}$$

holds as long as $\tau \geq h_\tau \left( \frac{r\epsilon}{4 n_x n_u} \right)$.

**Bounding $T_3$:** To precede, we bound $T_3$ as

$$T_3 = \left\| \frac{1}{ML} \sum_{i=1}^{M} \sum_{l=0}^{L-1} \left[ \hat{\nabla} C^{(i)}(K_{n,l}^{(i)}) - \nabla C^{(i)}(K_{n,l}^{(i)}) \right] \right\|_F$$

$$\leq \frac{1}{ML} \sum_{i=1}^{M} \sum_{l=0}^{L-1} \underbrace{\left\| \nabla C_r^{(i)}(K_{n,l}^{(i)}) - \nabla C^{(i)}(K_{n,l}^{(i)}) \right\|_F}_{\text{Bias term } \textcircled{1}}$$

$$+ \underbrace{\left\| \frac{1}{ML} \sum_{i=1}^{M} \sum_{l=0}^{L-1} \left[ \hat{\nabla} C^{(i)}(K_{n,l}^{(i)}) - \nabla C_r^{(i)}(K_{n,l}^{(i)}) \right] \right\|_F}_{\text{Variance term } \textcircled{2}} \tag{30}$$

where $\nabla C_r^{(i)}(K_{n,l}^{(i)}) := \mathbb{E}_{U_{n,l}^{(i)} \sim \mathbb{B}_r} \left[ \nabla C^{(i)}(K_{n,l}^{(i)} + U_{n,l}^{(i)}) \right]$.

For the bias term $\textcircled{1}$, since the smoothing radius $r \leq h_r \left( \frac{\epsilon}{4} \right)$, we have that

$$\textcircled{1} = \left\| \nabla C_r^{(i)}(K_{n,l}^{(i)}) - \nabla C^{(i)}(K_{n,l}^{(i)}) \right\|_F \leq h_{\text{grad}}(K_{n,l}^{(i)}) r \leq \bar{h}_{\text{grad}} r \leq \frac{\epsilon}{4}. \tag{31}$$

For the variance term, $\textcircled{2}$, we will exploit the matrix Freedman inequality (Lemma 7) to bound it. For simplicity, we denote

$$e_l^{(i)} := \frac{1}{ML} \left[ \hat{\nabla} C^{(i)}(K_{n,l}^{(i)}) - \nabla C_r^{(i)}(K_{n,l}^{(i)}) \right], \quad e_l := \sum_{i=1}^{M} e_l^{(i)},$$

Then, we have

$$\frac{1}{ML} \sum_{i=1}^{M} \sum_{l=0}^{L-1} \left[ \hat{\nabla} C^{(i)}(K_{n,l}^{(i)}) - \nabla C_r^{(i)}(K_{n,l}^{(i)}) \right] = \sum_{l=0}^{L-1} e_l.$$

Next, we aim to prove the following claims:

**Claim I:** $Y_t := \sum_{l=0}^{t} e_l$ is a martingale w.r.t $\mathcal{F}_{t-1}^n$ for $t = 1, \cdots, L-1$ and $e_l := \sum_{i=1}^{M} e_l^{(i)}$ is a martingale difference sequence.

**Proof:** Note that $\mathbb{E}[\hat{\nabla} C^{(i)}(K_{n,l}^{(i)})] = \nabla C_r^{(i)}(K_{n,l}^{(i)})$. Then we can easily have $\mathbb{E}[e_l] = 0$ for $l = 0, \cdots, L-1$. As a result, we have $\mathbb{E}[Y_t \mid \mathcal{F}_{t-1}^n] = Y_{t-1}$ since $Y_t = Y_{t-1} + e_t$. In other words, $Y_t := \sum_{l=0}^{t} e_l$ is a martingale w.r.t $\mathcal{F}_{t-1}^n$ for $t = 1, \cdots, L-1$.

**Claim II:** $\left\| \mathbb{E} \left[ e_l e_l^\top \mid \mathcal{F}_{l-1}^n \right] \right\| \leq \frac{\sigma_{\hat{\nabla}}^2}{n_s M L^2}$ where $\sigma_{\hat{\nabla}}^2$ is as defined in Lemma 14.

**Proof:** From Lemma B.7 in (Gravell et al., 2020), we can write

$$\left\| \mathbb{E} \left[ e_l^{(i)} e_l^{(i)\top} \mid \mathcal{F}_{l-1}^n \right] \right\| \leq \frac{\sigma_{\hat{\nabla}}^2}{n_s M^2 L^2}, \left\| \mathbb{E} \left[ e_l^{(i)\top} e_l^{(i)} \mid \mathcal{F}_{l-1}^n \right] \right\| \leq \frac{\sigma_{\hat{\nabla}}^2}{n_s M^2 L^2},$$

and based on this fact, we have

$$\left\| \mathbb{E}\left[ e_l e_l^\top \mid \mathcal{F}_{l-1}^n \right] \right\| = \left\| \mathbb{E}\left[ \left( \sum_{i=1}^M e_l^{(i)} \right) \left( \sum_{i=1}^M e_l^{(i)\top} \right) \mid \mathcal{F}_{l-1}^n \right] \right\|$$

$$\leq \sum_{i=1}^M \left\| \mathbb{E}\left[ e_l^{(i)} e_l^{(i)\top} \mid \mathcal{F}_{l-1}^n \right] \right\| + \underbrace{\sum_{i \neq j}^M \left\| \mathbb{E}\left[ e_l^{(i)} e_l^{(j)\top} \mid \mathcal{F}_{l-1}^n \right] \right\|}_{T_4 = 0} \leq \frac{\sigma_{\hat{\nabla}}^2}{n_s M L^2},$$

where we use the fact that $T_4 = 0$ because $e_l^{(i)}$ and $e_l^{(j)}$ are independent, if we conditioned on $\mathcal{F}_l^n$. An identical argument holds for $\left\| \mathbb{E}\left[ e_l^\top e_l \mid \mathcal{F}_{l-1}^n \right] \right\|$.

Define $W_{\mathrm{col},t} := \sum_{l=0}^t \mathbb{E}\left[ e_l e_l^\top \mid \mathcal{F}_{l-1}^n \right]$ and $W_{\mathrm{row},t} := \sum_{l=0}^t \mathbb{E}\left[ e_l^\top e_l \mid \mathcal{F}_{l-1}^n \right]$, then we have

$$\|W_{\mathrm{col},t}\| \leq \frac{\sigma_{\hat{\nabla}}^2}{n_s M L}, \quad \|W_{\mathrm{row},t}\| \leq \frac{\sigma_{\hat{\nabla}}^2}{n_s M L}.$$

**Claim III:** $\|e_l\| \leq \frac{R_{\hat{\nabla}}}{n_s L}$ where $R_{\hat{\nabla}} = \frac{2 n_x n_u \bar{C}_{\max}}{r} + \frac{\epsilon}{2} + \bar{h}_1$.

**Proof:** From Lemma B.7 in (Gravell et al., 2020), we have $\|e_l^{(i)}\| \leq \frac{n_s R_{\hat{\nabla}}}{M L}$. With this fact, we have

$$\|e_l\| \leq \sum_{i=1}^M \left\| e_l^{(i)} \right\| \leq \frac{R_{\hat{\nabla}}}{n_s L}.$$

With **Claim I, II** and **Claim III** and the matrix Freedman inequality (7), we have, for all $\epsilon \geq 0$,

$$\mathbb{P}\left\{ \exists t \geq 0 : \lambda_{\max}(Y_t) \geq \epsilon \text{ and } \max\left\{ \|W_{\mathrm{col},t}\|, \|W_{\mathrm{row},t}\| \right\} \leq \frac{\sigma_{\hat{\nabla}}^2}{n_s M L} \right\}$$

$$\leq (n_x + n_u) \exp\left\{ -\frac{\epsilon^2/2}{\frac{\sigma_{\hat{\nabla}}^2}{n_s M L} + \frac{R_{\hat{\nabla}} \epsilon}{3 n_s L}} \right\}. \tag{32}$$

Therefore, rephrasing Eq.(32), if

$$n_s \geq \left( \underbrace{\frac{32 \sigma_{\hat{\nabla}}^2 \min(n_x, n_u)}{M L \epsilon^2}}_{T_5} + \underbrace{\frac{32 L R_{\hat{\nabla}} \sqrt{\min(n_x, n_u)}}{12 L \epsilon}}_{T_6} \right) \log\left[ \frac{M L (n_x + n_u)}{\delta} \right], \tag{33}$$

we have that

$$\|Y_L\|_F = \|\frac{1}{M L} \sum_{i=1}^M \sum_{l=0}^{L-1} \left[ \hat{\nabla} C^{(i)}(K_{n,l}^{(i)}) - \nabla C_r^{(i)}(K_{n,l}^{(i)}) \right] \|_F \leq \frac{\epsilon}{4}, \tag{34}$$

holds with probability $1 - \delta$. As we discussed in Lemma 14, we only keep the dominant term $T_5$ in the requirement of the sample size $n_s$ (as in Eq.(33)). Because $T_5$ is in the order $\mathcal{O}(\epsilon^{-2})$ while $T_6$ is in the order $\mathcal{O}(\epsilon^{-1})$. Moreover, we note that by making $\epsilon = \mathcal{O}(1/M)$, $T_6$ when compared to $T_5$ is negligible. We also note that, as an alternative to imposing the restriction $\epsilon = \mathcal{O}(1/M)$, one can apply the Freedman lemma (Lemma 7) to $ML$ terms rather than to $L$ terms.

In summary, if $n_s \geq \frac{32 \sigma_{\hat{\nabla}}^2 \min(n_x, n_u)}{M L \epsilon^2} \left[ \frac{M L (n_x + n_u)}{\delta} \right] = \frac{h_{\mathrm{sample}}\left( \frac{\epsilon}{4}, \frac{\delta}{M L} \right)}{M L}$,

$$②= \left\| \frac{1}{M L} \sum_{i=1}^M \sum_{l=0}^{L-1} \left[ \hat{\nabla} C^{(i)}(K_{n,l}^{(i)}) - \nabla C_r^{(i)}(K_{n,l}^{(i)}) \right] \right\|_F \leq \frac{\epsilon}{4} \tag{35}$$

holds with probability $1 - \delta$.

As a result, we have $T_3 \leq \frac{\epsilon}{2}$ holds with probability $1 - \delta$, when $r \leq h_r\left(\frac{\epsilon}{4}\right)$ and $n_s \geq \frac{h_{\text{sample}}\left(\frac{\epsilon}{4}, \frac{\delta}{ML}\right)}{ML}$. In what follows, we will provide an upper bound on the term $T_1$.

**Bounding $T_1$:** We can follow the same analysis of bounding ②in $T_3$ to bound $T_1$. Different from the filtration we define in analyzing ②, we need to define a new filtration $\tilde{\mathcal{F}}_{l-1}^n$, where $\tilde{\mathcal{F}}_{l-1}^n := \mathcal{F}_{l-1}^n \cup U_l^n$ and $U_l^n := \left\{U_{n,l,s}^{(i)}\right\}_{s=1,\cdots,n_s}^{i=1,\cdots,N}$. Note that $U_l^n$ is the sigma-field generated by the randomness of all random smoothing matrices $U_{n,l,s}^{(i)}$ [18] from all the systems at the $n$-th global iteration and $l$-th local iteration. Replacing $\sigma_{\hat{\nabla}}^2$ Eq.(33) with into $\sigma_{\tilde{\nabla}}^2$ and $R_{\hat{\nabla}}$ with $R_{\tilde{\nabla}}$, we have that

$$T_1 = \left\| \frac{1}{ML} \sum_{i=1}^{M} \sum_{l=0}^{L} \left[ \tilde{\nabla} C^{(i)}(K_{n,l}^{(i)}) - \nabla' C^{(i)}(K_{n,l}^{(i)}) \right] \right\|_F \leq \frac{\epsilon}{4} \tag{36}$$

holds with probability $1 - \delta$ when

$$n_s \geq \frac{32 \sigma_{\tilde{\nabla}}^2 \min(n_x, n_u)}{ML\epsilon^2} \log\left[ \frac{ML(n_x + n_u)}{\delta} \right] = \frac{h_{\text{sample,trunc}}\left(\frac{\epsilon}{4}, \frac{\delta}{ML}, \frac{H^2}{\mu}\right)}{ML}.$$

Combing the upper bound of $T_1$ (Eq.(36)), $T_2$ (Eq.(29)) and $T_3$ (Eq.(31) and (35)), we have

$$\left\| \frac{1}{ML} \sum_{i=1}^{M} \sum_{l=0}^{L-1} \left[ \widehat{\nabla C^{(i)}(K_{n,l}^{(i)})} - \nabla C^{(i)}(K_{n,l}^{(i)}) \right] \right\|_F \leq T_1 + T_2 + T_3 \leq \epsilon$$

when the trajectory length $\tau$ satisfies $\tau \geq h_\tau\left(\frac{r\epsilon}{4n_x n_u}\right)$, the smoothing radius satisfies $r \leq h_r\left(\frac{\epsilon}{4}\right)$ and the size of samples satisfies $n_s \geq \max\left\{ \frac{h_{\text{sample,trunc}}\left(\frac{\epsilon}{4}, \frac{\delta}{ML}, \frac{H^2}{\mu}\right)}{ML}, \frac{h_{\text{sample}}\left(\frac{\epsilon}{4}, \frac{\delta}{ML}\right)}{ML} \right\} = \frac{h_{\text{sample,trunc}}\left(\frac{\epsilon}{4}, \frac{\delta}{ML}, \frac{H^2}{\mu}\right)}{ML}$. Thus, we complete the proof of Lemma 4. □

### A.6.3 Proof of Theorem 2

**Outline:** To prove Theorem 2, we first introduce some lemmas: Lemma 16 establishes stability of the local policies; Lemma 17 provides the drift analysis; Lemma 18 quantifies the per-round progress of our `FedLQR` algorithm. As a result, we are able to present the iterative stability guarantees and convergence analysis of `FedLQR` in the model-free setting.

**Lemma 16.** *(Stability of the local policies) Suppose $K_n \in \mathcal{G}^0$ and the heterogeneity level satisfies $(\epsilon_1 \bar{h}_{het}^1 + \epsilon_2 \bar{h}_{het}^2)^2 \leq \bar{h}_{het}^3$, where $\bar{h}_{het}^3$ is as defined in Eq.(21). If the local step-size $\eta_l$ satisfies*

$$\eta_l \leq \min\left\{ \frac{\underline{h}_\Delta \mu}{H^2\left(h_1 + \sqrt{\bar{\epsilon}}\right)}, \frac{1}{9 \bar{h}_{grad}} \right\},$$

*the smoothing radius satisfies*

$$r \leq \min\left\{ \frac{\min_{i \in [M]} C^{(i)}(K_0)}{\bar{h}_{cost}}, \underline{h}_\Delta, h_r\left(\frac{\sqrt{\bar{\epsilon}}}{4}\right) \right\},$$

*the trajectory length satisfies $\tau \geq h_\tau\left(\frac{r\sqrt{\bar{\epsilon}}}{4n_x n_u}\right)$, and the number of the sample size satisfies*

$$n_s \geq \max\left\{ h_{sample,trunc}\left(\frac{\sqrt{\bar{\epsilon}}}{4}, \frac{\delta}{L}, \frac{H^2}{\mu}\right), h_{sample}\left(\frac{\sqrt{\bar{\epsilon}}}{2}, \frac{\delta}{L}\right) \right\}$$

---

[18]Here we use the index $s$ to denote $s$-th sample. Note that in each local iteration $l$, we need to generate the random smoothing matrices $n_s$ times.

*where we choose a fixed error tolerance $\bar{\epsilon}$ to be*

$$\bar{\epsilon} := \min_{j \in [M]} \left\{ \frac{3\mu^2 \sigma_{\min}(R) \left( C^{(j)}(K_0) - C^{(j)}(K_j^*) \right)}{5 \|\Sigma_{K_j^*}\|} \right\},$$

*then with probability $1 - \delta$,, where $\delta \in (0, 1)$, $K_{n,l}^{(i)} \in \mathcal{G}^0$ holds for all $i \in [M]$ and $l = 0, 1, \cdots, L - 1$.*

*Proof.* For any $i, j \in [M]$, according to the local Lipschitz property in Lemma 1, we have that

$$\begin{aligned}
C^{(j)}(K_{n,1}^{(i)}) - C^{(j)}(K_n) &\leq \left\langle \nabla C^{(j)}(K_n), K_{n,1}^{(i)} - K_n \right\rangle \\
&\quad + \frac{h_{\text{grad}(K_n)}}{2} \left\| K_{n,1}^{(i)} - K_n \right\|_F^2 \quad \text{(Local lipschitz)} \\
&= -\left\langle \nabla C^{(j)}(K_n), \eta_l \tilde{\nabla} C^{(i)}(K_n) \right\rangle + \frac{h_{\text{grad}(K_n)}}{2} \left\| \eta_l \tilde{\nabla} C^{(i)}(K_n) \right\|_F^2,
\end{aligned}$$

holds if $\left\| \eta_l \tilde{\nabla} C^{(i)}(K_n) \right\|_F \leq \underline{h}_\Delta \leq h_\Delta(K_n)$. Note that this inequality holds when $\eta_l$ satisfies

$$\begin{aligned}
\left\| \eta_l \tilde{\nabla} C^{(i)}(K_n) \right\|_F &= \eta_l \left\| \frac{1}{n_s} \sum_{s=1}^{n_s} \frac{n_u n_x}{r^2} \tilde{C}^{(i),(\tau)} \left( K_n + U_s^{(i)} \right) U_s^{(i)} \right\|_F \\
&\overset{(a)}{\leq} \eta_l \frac{H^2}{\mu} \left\| \frac{1}{n_s} \sum_{i=1}^{n_s} \frac{n_x n_u}{r^2} C^{(i)} \left( K_n + U_s^{(i)} \right) U_s^{(i)} \right\|_F \\
&= \frac{\eta_l H^2}{\mu} \left\| \hat{\nabla} C^{(i)}(K) \right\|_F \\
&\leq \frac{\eta_l H^2}{\mu} \left[ \left\| \nabla C^{(i)}(K) \right\|_F + \left\| \hat{\nabla} C^{(i)}(K) - \nabla C^{(i)}(K) \right\|_F \right] \\
&\overset{(b)}{\leq} \frac{\eta_l H^2}{\mu} \left[ \left\| \nabla C^{(i)}(K_n) \right\|_F + \sqrt{\bar{\epsilon}} \right] \\
&\leq \frac{\eta_l H^2}{\mu} \left( h_1 + \sqrt{\bar{\epsilon}} \right)
\end{aligned} \tag{37}$$

where[19] (a) is due to Lemma 15; according to Lemma 14, (b) holds with high probability, when the number of the sample size satisfies $n_s \geq h_{\text{sample}} \left( \frac{\sqrt{\bar{\epsilon}}}{2}, \frac{\delta}{L} \right)$. The last inequality follows from the uniform upper gradient bound in Lemma 6. Then we can easily conclude that $\left\| \eta_l \tilde{\nabla} C^{(i)}(K_n) \right\|_F \leq \underline{h}_\Delta$ holds when $\eta_l \leq \frac{\underline{h}_\Delta \mu}{H^2 \left( h_1 + \sqrt{\bar{\epsilon}} \right)}$.

Following the analysis in Eq (22), we have

$$\begin{aligned}
C^{(j)}(K_{n,1}^{(i)}) - C^{(j)}(K_n) &\leq -\eta_l \left\langle \nabla C^{(j)}(K_n), \nabla C^{(j)}(K_n) \right\rangle \\
&\quad - \eta_l \underbrace{\left\langle \nabla C^{(j)}(K_n), \nabla C^{(i)}(K_n) - \nabla C^{(j)}(K_n) \right\rangle}_{T_1} \\
&\quad \underbrace{-\eta_l \left\langle \nabla C^{(j)}(K_n), \tilde{\nabla} C^{(i)}(K_n) - \nabla C^{(i)}(K_n) \right\rangle}_{T_2} + \frac{h_{\text{grad}(K_n)}}{2} \left\| \eta_l \tilde{\nabla} C^{(i)}(K_n) \right\|_F^2,
\end{aligned}$$

where $T_1$ can be upper bounded as

$$T_1 \leq \eta_l \left\| \nabla C^{(j)}(K_n) \right\|_F \left\| \nabla C^{(i)}(K_n) - \nabla C^{(j)}(K_n) \right\|_F$$

---

[19] For sake of the notation, we ignore the dependence on the local iteration $l$ and global iteration $n$ when we index $U_s^{(i)}$ in this part.

$$\leq \eta_l \sqrt{\min\{n_x, n_u\}} \left\| \nabla C^{(j)}(K_n) \right\|_F (\epsilon_1 \bar{h}_{\text{het}}^1 + \epsilon_2 \bar{h}_{\text{het}}^2),$$

where we use the policy gradient heterogeneity bound in Lemma 3 and the fact that $K_n \in \mathcal{G}^0$.

We can bound $T_2$ as follows

$$T_2 \leq \eta_l \left\| \nabla C^{(j)}(K_n) \right\|_F \left\| \tilde{\nabla} C^{(i)}(K_n) - \nabla C^{(i)}(K_n) \right\|_F$$

$$\leq \eta_l \left\| \nabla C^{(j)}(K_n) \right\|_F \sqrt{\epsilon},$$

where it holds with probability $1 - \delta$. Here we use the Cauchy-Schwarz inequality in the first inequality, and the second inequality is due to Lemma 15 since $n_s \geq h_{\text{sample,trunc}} \left( \frac{\sqrt{\epsilon}}{4}, \delta, \frac{H^2}{\mu} \right)$, the smoothing radius satisfies $r \leq h_r \left( \frac{\sqrt{\epsilon}}{4} \right)$ and the length of trajectories satisfies $\tau \geq h_\tau \left( \frac{r\sqrt{\epsilon}}{4n_x n_u} \right)$.

Plugging the upper bounds of $T_1$ and $T_2$ in Eq (23), we have:

$$C^{(j)}(K_{n,1}^{(i)}) - C^{(j)}(K_n) \overset{(a)}{\leq}$$

$$- \eta_l \left\| \nabla C^{(j)}(K_n) \right\|_F^2 + \eta_l \sqrt{\min\{n_x, n_u\}} \left\| \nabla C^{(j)}(K_n) \right\|_F (\epsilon_1 \bar{h}_{\text{het}}^1 + \epsilon_2 \bar{h}_{\text{het}}^2)$$

$$+ \eta_l \left\| \nabla C^{(j)}(K_n) \right\|_F \sqrt{\epsilon} + \frac{3 h_{\text{grad}(K_n)} \eta_l^2}{2} \left\| \tilde{\nabla} C^{(i)}(K_n) - \nabla C^{(i)}(K_n) \right\|_F^2$$

$$+ \frac{3 h_{\text{grad}(K_n)} \eta_l^2}{2} \left\| \nabla C^{(i)}(K_n) - \nabla C^{(j)}(K_n) \right\|_F^2$$

$$+ \frac{3 h_{\text{grad}(K_n)} \eta_l^2}{2} \left\| \nabla C^{(j)}(K_n) \right\|_F^2$$

$$\overset{(b)}{\leq} - \eta_l \left\| \nabla C^{(j)}(K_n) \right\|_F^2 + \eta_l \sqrt{\min\{n_x, n_u\}} \left\| \nabla C^{(j)}(K_n) \right\|_F (\epsilon_1 \bar{h}_{\text{het}}^1 + \epsilon_2 \bar{h}_{\text{het}}^2)$$

$$+ \eta_l \left\| \nabla C^{(j)}(K_n) \right\|_F \sqrt{\epsilon} + \frac{3 \bar{h}_{\text{grad}} \eta_l^2}{2} \bar{\epsilon}$$

$$+ \frac{3 \bar{h}_{\text{grad}} \eta_l^2 \min\{n_x, n_u\}}{2} (\epsilon_1 \bar{h}_{\text{het}}^1 + \epsilon_2 \bar{h}_{\text{het}}^2)^2 + \frac{3 \bar{h}_{\text{grad}} \eta_l^2}{2} \left\| \nabla C^{(j)}(K_n) \right\|_F^2,$$

where $(a)$ follows from Eq.(8); $(b)$ follows from the same reasoning as we bound $T_1$ and $T_2$ and the fact that $K_n \in \mathcal{G}^0$. If we choose the local step-size $\eta_l$ satisfies $\eta_l \leq \frac{1}{9\bar{h}_{\text{grad}}}$, i.e., $\frac{3\bar{h}_{\text{grad}} \eta_l^2}{2} \leq \frac{\eta_l}{6}$, we have

$$C^{(j)}(K_{n,1}^{(i)}) - C^{(j)}(K_n) \overset{(a)}{\leq} -\eta_l \left\| \nabla C^{(j)}(K_n) \right\|_F^2 + \frac{\eta_l}{6} \left\| \nabla C^{(j)}(K_n) \right\|_F^2$$

$$+ \frac{3\eta_l \min\{n_x, n_u\}}{2} (\epsilon_1 \bar{h}_{\text{het}}^1 + \epsilon_2 \bar{h}_{\text{het}}^2)^2 + \frac{\eta_l \left\| \nabla C^{(j)}(K_n) \right\|_F^2}{6} + \frac{3\eta_l \bar{\epsilon}}{2} + \frac{\eta_l}{6} \bar{\epsilon}$$

$$+ \frac{\eta_l \min\{n_x, n_u\}}{6} (\epsilon_1 \bar{h}_{\text{het}}^1 + \epsilon_2 \bar{h}_{\text{het}}^2)^2 + \frac{\eta_l}{6} \left\| \nabla C^{(j)}(K_n) \right\|_F^2$$

$$\leq -\frac{\eta_l}{2} \left\| \nabla C^{(j)}(K_n) \right\|_F^2 + \frac{5\eta_l \min\{n_x, n_u\}}{3} (\epsilon_1 \bar{h}_{\text{het}}^1 + \epsilon_2 \bar{h}_{\text{het}}^2)^2 + \frac{5\eta_l}{3} \bar{\epsilon}$$

$$\overset{(b)}{\leq} -\frac{2\eta_l \sigma_{\min}(R) \mu^2}{\|\Sigma_{K_j^*}\|} (C^{(j)}(K_n) - C^{(j)}(K_j^*)) + \frac{5\eta_l \min\{n_x, n_u\}}{3} (\epsilon_1 \bar{h}_{\text{het}}^1 + \epsilon_2 \bar{h}_{\text{het}}^2)^2 + \frac{5\eta_l}{3} \bar{\epsilon},$$

where $(a)$ follows from the Young's inequality in Eq.(9); and $(b)$ follows from the gradient domination in Lemma 2.

Therefore, if the heterogeneity satisfies $(\epsilon_1 \bar{h}_{\text{het}}^1 + \epsilon_2 \bar{h}_{\text{het}}^2)^2 \leq \bar{h}_{\text{het}}^3$, then we have

$$(\epsilon_1 \bar{h}_{\text{het}}^1 + \epsilon_2 \bar{h}_{\text{het}}^2)^2 \leq \min_{j \in [M]} \left\{ \frac{3\mu^2 \sigma_{\min}(R) \left( C^{(j)}(K_0) - C^{(j)}(K_j^*) \right)}{5 \|\Sigma_{K_j^*}\| \min\{n_x, n_u\}} \right\}.$$

Since the error tolerance

$$\bar{\epsilon} = \min_{j \in [M]} \left\{ \frac{3\mu^2 \sigma_{\min}(R) \left( C^{(j)}(K_0) - C^{(j)}(K_j^*) \right)}{5 \|\Sigma_{K_j^*}\|} \right\},$$

we have

$$C^{(j)}(K_{n,1}^{(i)}) - C^{(j)}(K_j^*) \leq \left( 1 - \frac{2\eta\mu^2 \sigma_{\min}(R)}{\left\| \Sigma_{K_j^*} \right\|} \right) (C^{(j)}(K_n) - C^{(j)}(K_j^*))$$

$$+ \frac{\eta_l \mu^2 \sigma_{\min}(R) \left( C^{(j)}(K_0) - C^{(j)}(K_j^*) \right)}{\|\Sigma_{K_j^*}\|} + \frac{\eta_l \mu^2 \sigma_{\min}(R) \left( C^{(j)}(K_0) - C^{(j)}(K_j^*) \right)}{\|\Sigma_{K_j^*}\|}$$

$$\overset{(a)}{\leq} \left( 1 - \frac{2\eta\mu^2 \sigma_{\min}(R)}{\left\| \Sigma_{K_j^*} \right\|} \right) (C^{(j)}(K_0) - C^{(j)}(K_j^*)) + \frac{2\eta_l \mu^2 \sigma_{\min}(R) \left( C^{(j)}(K_0) - C^{(j)}(K_j^*) \right)}{\|\Sigma_{K_j^*}\|}$$

$$= C^{(j)}(K_0) - C^{(j)}(K_j^*), \forall j \in [M],$$

where we use the fact that $K_n \in \mathcal{G}^0$ in $(a)$. The above inequality implies $K_{n,1}^{(i)} \in \mathcal{G}^0$ with high probability $1 - \delta$ when $K_n \in \mathcal{G}^0$. Then we can use the induction method to obtain that $K_{n,2}^{(i)} \in \mathcal{G}^0$, since $K_{n,1}^{(i)} \in \mathcal{G}^0$. By repeating this step for $L$ times, we have that all the local polices $K_{n,l}^{(i)} \in \mathcal{G}^0$ holds for all $i \in [M]$ and $l = 0, 1, \cdots, L - 1$, when the global policy $K_n \in \mathcal{G}^0$. □

**Lemma 17.** *(Drift term analysis) Suppose $K_n \in \mathcal{G}^0$. If $\eta_l \leq \min \left\{ \frac{1}{4\bar{h}_{grad}}, \frac{1}{4}, \frac{\log 2}{L(3\bar{h}_{grad}+2)} \right\}$, the number of the sample size $n_s$ satisfies*

$$n_s \geq \frac{h_{sample,trunc} \left( \frac{\sqrt{\epsilon}}{4}, \frac{\delta}{L}, \frac{H^2}{\mu} \right)}{ML},$$

*the smoothing radius satisfies $r \leq h_r \left( \frac{\sqrt{\epsilon}}{4} \right)$ and the length of trajectories satisfies*

$$\tau \geq h_\tau \left( \frac{r\sqrt{\epsilon}}{4n_x n_u} \right),$$

*given any $\delta \in (0, 1)$, the difference between the local policy and global policy can be bounded by*

$$\left\| K_{n,l}^{(i)} - K_n \right\|_F^2 \leq 2\eta_l L \left[ \left\| \nabla C^{(i)}(K_n) \right\|_F^2 + ML\epsilon \right] = \frac{2\eta}{\eta_g} \left[ \left\| \nabla C^{(i)}(K_n) \right\|_F^2 + ML\epsilon \right]$$

*holds, with probability $1 - \delta$, for all $i \in [M]$ and $l = 0, 1, \cdots, L - 1$.*

*Proof.*

$$\left\| K_{n,l}^{(i)} - K_n \right\|_F^2 = \left\| K_{n,l-1}^{(i)} - K_n - \eta_l \tilde{\nabla} C^{(i)}(K_{n,l-1}^{(i)}) \right\|_F^2$$

$$= \left\| K_{n,l-1}^{(i)} - K_n \right\|_F^2 - 2\eta_l \left[ \left\langle \tilde{\nabla} C^{(i)}(K_{n,l-1}^{(i)}), K_{n,l-1}^{(i)} - K_n \right\rangle \right]$$

$$+ \left\| \eta_l \tilde{\nabla} C^{(i)}(K_{n,l-1}^{(i)}) \right\|_F^2$$

$$= \left\| K_{n,l-1}^{(i)} - K_n \right\|_F^2 - 2\eta_l \left[ \left\langle \tilde{\nabla} C^{(i)}(K_{n,l-1}^{(i)}) - \nabla C^{(i)}(K_{n,l-1}^{(i)}), K_{n,l-1}^{(i)} - K_n \right\rangle \right]$$

$$- 2\eta_l \left[ \left\langle \nabla C^{(i)}(K_{n,l-1}^{(i)}) - \nabla C^{(i)}(K_n), K_{n,l-1}^{(i)} - K_n \right\rangle \right] - 2\eta_l \left[ \left\langle \nabla C^{(i)}(K_n), K_{n,l-1}^{(i)} - K_n \right\rangle \right]$$

$$+ \left\| \eta_l \tilde{\nabla} C^{(i)}(K_{n,l-1}^{(i)}) \right\|_F^2$$

$$\overset{(a)}{\leq} \left\| K_{n,l-1}^{(i)} - K_n \right\|_F^2 - 2\eta_l \left[ \left\langle \tilde{\nabla} C^{(i)}(K_{n,l-1}^{(i)}) - \nabla C^{(i)}(K_{n,l-1}^{(i)}), K_{n,l-1}^{(i)} - K_n \right\rangle \right]$$

$$+ 2\eta_l \left\| \nabla C^{(i)}(K_{n,l-1}^{(i)}) - \nabla C^{(i)}(K_n) \right\|_F \left\| K_{n,l-1}^{(i)} - K_n \right\|_F + 2\eta_l \left\| \nabla C^{(i)}(K_n) \right\|_F \left\| K_{n,l-1}^{(i)} - K_n \right\|_F$$

$$+ \left\| \eta_l \tilde{\nabla} C^{(i)}(K_{n,l-1}^{(i)}) \right\|_F^2$$

$$\overset{(b)}{\leq} \left\| K_{n,l-1}^{(i)} - K_n \right\|_F^2 - 2\eta_l \left[ \left\langle \tilde{\nabla} C^{(i)}(K_{n,l-1}^{(i)}) - \nabla C^{(i)}(K_{n,l-1}^{(i)}), K_{n,l-1}^{(i)} - K_n \right\rangle \right]$$

$$+ 2\eta_l h_{\text{grad}}(K_n) \left\| K_{n,l-1}^{(i)} - K_n \right\|_F \left\| K_{n,l-1}^{(i)} - K_n \right\|_F + \eta_l \left\| \nabla C^{(i)}(K_n) \right\|_F^2 + \eta_l \left\| K_{n,l-1}^{(i)} - K_n \right\|_F^2$$

$$+ \left\| \eta_l \tilde{\nabla} C^{(i)}(K_{n,l-1}^{(i)}) \right\|_F^2 \tag{38}$$

where we use Cauchy–schwarz inequality for $(a)$; and for (b), we use Eq. (9).

Following the analysis in Eq.(38), we have

$$\left\| K_{n,l}^{(i)} - K_n \right\|_F^2 \leq (1 + 2\eta_l h_{\text{grad}}(K_n) + \eta_l) \left\| K_{n,l-1}^{(i)} - K_n \right\|_F^2 + \eta_l \left\| \nabla C^{(i)}(K_n) \right\|_F^2$$

$$- 2\eta_l \left[ \left\langle \tilde{\nabla} C^{(i)}(K_{n,l-1}^{(i)}) - \nabla C^{(i)}(K_{n,l-1}^{(i)}), K_{n,l-1}^{(i)} - K_n \right\rangle \right] + \left\| \eta_l \tilde{\nabla} C^{(i)}(K_{n,l-1}^{(i)}) \right\|_F^2$$

$$\overset{(a)}{\leq} (1 + 2\eta_l h_{\text{grad}}(K_n) + \eta_l) \left\| K_{n,l-1}^{(i)} - K_n \right\|_F^2 + \eta_l \left\| \nabla C^{(i)}(K_n) \right\|_F^2$$

$$+ 2\eta_l \left[ \left\| \tilde{\nabla} C^{(i)}(K_{n,l-1}^{(i)}) - \nabla C^{(i)}(K_{n,l-1}^{(i)}) \right\|_F \left\| K_{n,l-1}^{(i)} - K_n \right\|_F \right]$$

$$+ 2\eta_l^2 \left\| \tilde{\nabla} C^{(i)}(K_{n,l-1}^{(i)}) - \nabla C^{(i)}(K_{n,l-1}^{(i)}) \right\|_F^2 + 2\eta_l^2 \left\| \nabla C^{(i)}(K_{n,l-1}^{(i)}) \right\|_F^2$$

$$\overset{(b)}{\leq} (1 + 2\eta_l h_{\text{grad}}(K_n) + \eta_l) \left\| K_{n,l-1}^{(i)} - K_n \right\|_F^2 + \eta_l \left\| \nabla C^{(i)}(K_n) \right\|_F^2$$

$$+ \eta_l \left\| \tilde{\nabla} C^{(i)}(K_{n,l-1}^{(i)}) - \nabla C^{(i)}(K_{n,l-1}^{(i)}) \right\|_F^2 + \eta_l \left\| K_{n,l-1}^{(i)} - K_n \right\|_F^2$$

$$+ 2\eta_l^2 \left\| \tilde{\nabla} C^{(i)}(K_{n,l-1}^{(i)}) - \nabla C^{(i)}(K_{n,l-1}^{(i)}) \right\|_F^2$$

$$+ 4\eta_l^2 \left\| \nabla C^{(i)}(K_{n,l-1}^{(i)}) - \nabla C^{(i)}(K_n) \right\|_F^2 + 4\eta_l^2 \left\| \nabla C^{(i)}(K_n) \right\|_F^2$$

$$\overset{(c)}{\leq} (1 + 2\eta_l h_{\text{grad}}(K_n) + \eta_l) \left\| K_{n,l-1}^{(i)} - K_n \right\|_F^2 + (\eta_l + 4\eta_l^2) \left\| \nabla C^{(i)}(K_n) \right\|_F^2$$

$$+ \eta_l \left\| \tilde{\nabla} C^{(i)}(K_{n,l-1}^{(i)}) - \nabla C^{(i)}(K_{n,l-1}^{(i)}) \right\|_F^2 + \eta_l \left\| K_{n,l-1}^{(i)} - K_n \right\|_F^2$$

$$+ 2\eta_l^2 \left\| \tilde{\nabla} C^{(i)}(K_{n,l-1}^{(i)}) - \nabla C^{(i)}(K_{n,l-1}^{(i)}) \right\|_F^2 + 4\eta_l^2 h_{\text{grad}}(K_n)^2 \left\| K_{n,l-1}^{(i)} - K_n \right\|_F^2$$

$$\overset{(d)}{=} \left( 1 + 2\eta_l h_{\text{grad}}(K_n) + 2\eta_l + 4\eta_l^2 h_{\text{grad}}(K_n)^2 \right) \left\| K_{n,l-1}^{(i)} - K_n \right\|_F^2 + (\eta_l + 4\eta_l^2) \left\| \nabla C^{(i)}(K_n) \right\|_F^2$$

$$+ \left( \eta_l + 2\eta_l^2 \right) \left\| \tilde{\nabla} C^{(i)}(K_{n,l-1}^{(i)}) - \nabla C^{(i)}(K_{n,l-1}^{(i)}) \right\|_F^2$$

$$\leq \left( 1 + 2\eta_l \bar{h}_{\text{grad}} + 2\eta_l + 4\eta_l^2 \bar{h}_{\text{grad}}^2 \right) \left\| K_{n,l-1}^{(i)} - K_n \right\|_F^2 + (\eta_l + 4\eta_l^2) \left\| \nabla C^{(i)}(K_n) \right\|_F^2$$

$$+ \left( \eta_l + 2\eta_l^2 \right) \underbrace{\left\| \tilde{\nabla} C^{(i)}(K_{n,l-1}^{(i)}) - \nabla C^{(i)}(K_{n,l-1}^{(i)}) \right\|_F^2}_{T_1},$$

where we use Cauchy-Schwarz inequality and Eq.(6) for $(a)$; for $(b)$, we use Eq.(6) and (8); for $(c)$, we use the gradient smoothness lemma in Lemma 1; and for $(d)$, we use the fact that $K_n \in \mathcal{G}^0$.

From Lemma 15, we can bound $T_1$ term as follows:

$$T_1 = \left\| \tilde{\nabla} C^{(i)}(K_{n,l-1}^{(i)}) - \nabla C^{(i)}(K_{n,l-1}^{(i)}) \right\|_F^2 \leq ML\epsilon,$$

where it holds with probability $1 - \delta$, since $n_s \geq \frac{h_{\text{sample,trunc}}\left(\frac{\sqrt{\epsilon}}{4}, \delta, \frac{H^2}{\mu}\right)}{ML}$, the smoothing radius satisfies $r \leq h_r\left(\frac{\sqrt{\epsilon}}{4}\right)$ and the length of trajectories satisfies $\tau \geq h_\tau\left(\frac{r\sqrt{\epsilon}}{4n_x n_u}\right)$.

Hence, we obtain

$$
\begin{aligned}
\left\| K_{n,l}^{(i)} - K_n \right\|_F^2 &\leq \left(1 + 2\eta_l \bar{h}_{\text{grad}} + 2\eta_l + 4\eta_l^2 \bar{h}_{\text{grad}}^2\right) \left\| K_{n,l-1}^{(i)} - K_n \right\|_F^2 + \left(\eta_l + 4\eta_l^2\right) \left\| \nabla C^{(i)}(K_n) \right\|_F^2 \\
&\quad + \left(\eta_l + 2\eta_l^2\right) ML\epsilon \\
&\overset{(a)}{\leq} \left(1 + 3\eta_l \bar{h}_{\text{grad}} + 2\eta_l\right) \left\| K_{n,l-1}^{(i)} - K_n \right\|_F^2 + 2\eta_l \left\| \nabla C^{(i)}(K_n) \right\|_F^2 + 2\eta_l ML\epsilon \\
&\leq \left(1 + 3\eta_l \bar{h}_{\text{grad}} + 2\eta_l\right)^l \underbrace{\left\| K_{n,0}^{(i)} - K_n \right\|_F^2}_{=0} \\
&\quad + 2\eta_l \sum_{j=0}^{l-1} \left(1 + 3\eta_l \bar{h}_{\text{grad}} + 2\eta_l\right)^j \left[\left\| \nabla C^{(i)}(K_n) \right\|_F^2 + ML\epsilon\right] \\
&\leq 2\eta_l \times \frac{\left(1 + 3\eta_l \bar{h}_{\text{grad}} + 2\eta_l\right)^l - 1}{\left(1 + 3\eta_l \bar{h}_{\text{grad}} + 2\eta_l\right) - 1} \left[\left\| \nabla C^{(i)}(K_n) \right\|_F^2 + ML\epsilon\right] \\
&\leq 2 \times \frac{\left(1 + 3\eta_l \bar{h}_{\text{grad}} + 2\eta_l\right)^l - 1}{3\bar{h}_{\text{grad}} + 2} \left[\left\| \nabla C^{(i)}(K_n) \right\|_F^2 + ML\epsilon\right] \\
&\overset{(b)}{\leq} 2 \times \frac{1 + l(3\eta_l \bar{h}_{\text{grad}} + 2\eta_l) - 1}{3\bar{h}_{\text{grad}} + 2} \left[\left\| \nabla C^{(i)}(K_n) \right\|_F^2 + ML\epsilon\right] \\
&\leq 2\eta_l L \left[\left\| \nabla C^{(i)}(K_n) \right\|_F^2 + ML\epsilon\right],
\end{aligned}
$$

where $(a)$ is due to the choice of local step-size which satisfies $2\eta_l \bar{h}_{\text{grad}} + 2\eta_l + 4\eta_l^2 \bar{h}_{\text{grad}}^2 \leq 3\eta_l \bar{h}_{\text{grad}} + 2\eta_l$ and $\eta_l + 2\eta_l^2 \leq \eta_l + 4\eta_l^2 \leq 2\eta_l$, i.e., $\eta_l \leq \min\left\{\frac{1}{4\bar{h}_{\text{grad}}}, \frac{1}{4}\right\}$. For $(b)$, we used the fact that $(1+x)^{\tau+1} \leq 1 + 2x(\tau+1)$ holds for $x \leq \frac{\log 2}{\tau}$. In other words, $\left(1 + 3\eta_l \bar{h}_{\text{grad}} + 2\eta_l\right)^l \leq 1 + l(3\eta_l \bar{h}_{\text{grad}} + 2\eta_l)$ when $3\eta_l \bar{h}_{\text{grad}} + 2\eta_l \leq \frac{\log 2}{l}$, i.e., $\eta_l \leq \frac{\log 2}{L(3\bar{h}_{\text{grad}} + 2)}$. $\qquad\square$

**Lemma 18.** *(Per round progress) Suppose $K_n \in \mathcal{G}^0$. If we choose the local step-size as*

$$\eta_l = \frac{1}{2} \min\left\{ \frac{\underline{h}_\Delta \mu}{H^2(h_1 + \sqrt{\epsilon})}, \frac{1}{9\bar{h}_{grad}}, \frac{1}{4}, \frac{\log 2}{L(3\bar{h}_{grad} + 2)}, \frac{1}{256L\bar{h}_{grad}^2} \right\},$$

*with step-size $\eta := L\eta_l \eta_g = \frac{1}{2}\min\{\frac{\underline{h}_\Delta \mu}{H^2(h_1 + \sqrt{\epsilon})}, 1, \frac{1}{32\bar{h}_{grad}}\}$, and the smoothing radius[20]*

$$r \leq \min\left\{ \frac{\min_{i \in [M]} C^{(i)}(K_0)}{\bar{h}_{cost}}, \underline{h}_\Delta, h_r\left(\frac{\sqrt{\epsilon}}{4}\right) \right\},$$

---

[20] The exact requirement of $r$ is $r \leq \min\left\{ \frac{\min_{i \in [M]} C^{(i)}(K_0)}{\bar{h}_{\text{cost}}}, \underline{h}_\Delta, h_r\left(\frac{\sqrt{\epsilon}}{4}\right), h_r\left(\frac{\sqrt{\epsilon}}{4}\right) \right\}$. Here, without loss of generality, we drop the $h_r\left(\frac{\sqrt{\epsilon}}{4}\right)$ term from the min expression. This can be done because the error tolerance $\epsilon$ is usually small, and so $h_r\left(\frac{\sqrt{\epsilon}}{4}\right) \leq h_r\left(\frac{\sqrt{\epsilon}}{4}\right)$ holds. The assumptions on $\tau$ and $n_s$ follow similarly.

*where the trajectory length satisfies $\tau \geq h_\tau \left( \frac{r\sqrt{\epsilon}}{4n_x n_u} \right)$, and the number of the sample size satisfies*

$$n_s \geq \frac{h_{sample,trunc} \left( \frac{\sqrt{\epsilon}}{4}, \frac{\delta}{L}, \frac{H^2}{\mu} \right)}{ML},$$

*then with probability $1 - \delta$, for any small $\delta \in (0, 1)$, the `FedLQR` algorithm provides the following convergence guarantee:*

$$C^{(i)}(K_{n+1}) - C^{(i)}(K_i^*) \leq \left( 1 - \frac{\eta\mu^2\sigma_{\min}(R)}{\left\| \Sigma_{K_i^*} \right\|} \right) (C^{(i)}(K_n) - C^{(i)}(K_i^*)) + 2\eta\epsilon$$

$$+ 2\eta\min\{n_x, n_u\}(\epsilon_1\bar{h}_{het}^1 + \epsilon_1\bar{h}_{het}^2).^2 \tag{39}$$

*Proof.* For any $i \in [M]$, according to the local Lipschitz property in Lemma 1, we have that

$$C^{(i)}(K_{n+1}) - C^{(i)}(K_n) \leq \langle \nabla C^{(i)}(K_n), K_{n+1} - K_n \rangle + \frac{h_{\mathrm{grad}}(K_n)}{2} \|K_{n+1} - K_n\|_F^2$$

$$= -\left\langle \nabla C^{(i)}(K_n), \frac{\eta}{ML} \sum_{j=1}^{M} \sum_{l=0}^{L-1} \tilde{\nabla} C^{(j)}(K_{n,l}^{(j)}) \right\rangle + \frac{h_{\mathrm{grad}}(K_n)}{2} \left\| \frac{\eta}{ML} \sum_{j=1}^{M} \sum_{l=0}^{L-1} \tilde{\nabla} C^{(j)}(K_{n,l}^{(j)}) \right\|_F^2, \tag{40}$$

holds when $\left\| \frac{\eta}{ML} \sum_{j=1}^{M} \sum_{l=0}^{L-1} \tilde{\nabla} C^{(j)}(K_{n,l}^{(j)}) \right\|_F \leq \underline{h}_\Delta \leq h_\Delta(K_n)$. Following the same analysis as Eq.(37), this inequality holds when

$$\eta \leq \frac{\underline{h}_\Delta \mu}{H^2 \left( h_1 + \sqrt{\epsilon} \right)}, \quad r \leq \min \left\{ \frac{\min_{i \in [M]} C^{(i)}(K_0)}{\bar{h}_{\mathrm{cost}}}, \underline{h}_\Delta \right\}.$$

Following the analysis in Eq.(40), we have

$$C^{(i)}(K_{n+1}) - C^{(i)}(K_n)$$

$$\leq -\left\langle \nabla C^{(i)}(K_n), \frac{\eta}{ML} \sum_{j=1}^{M} \sum_{l=0}^{L-1} \tilde{\nabla} C^{(j)}(K_{n,l}^{(j)}) - \frac{\eta}{ML} \sum_{j=1}^{M} \sum_{l=0}^{L-1} \nabla C^{(j)}(K_{n,l}^{(j)}) \right\rangle$$

$$- \left\langle \nabla C^{(i)}(K_n), \frac{\eta}{ML} \sum_{j=1}^{M} \sum_{l=0}^{L-1} \nabla C^{(j)}(K_{n,l}^{(j)}) - \nabla C^{(j)}(K_n) \right\rangle$$

$$- \left\langle \nabla C^{(i)}(K_n), \frac{\eta}{M} \sum_{j=1}^{M} \nabla C^{(j)}(K_n) - \nabla C^{(i)}(K_n) \right\rangle - \eta \left\| C^{(i)}(K_n) \right\|^2$$

$$+ \frac{h_{\mathrm{grad}}(K_n)}{2} \left\| \frac{\eta}{ML} \sum_{j=1}^{M} \sum_{l=0}^{L-1} \tilde{\nabla} C^{(j)}(K_{n,l}^{(j)}) \right\|_F^2$$

$$\overset{(a)}{\leq} \eta \left\| \nabla C^{(i)}(K_n) \right\|_F \left\| \frac{1}{ML} \sum_{j=1}^{M} \sum_{l=0}^{L-1} \left[ \tilde{\nabla} C^{(j)}(K_{n,l}^{(j)}) - \nabla C^{(j)}(K_{n,l}^{(j)}) \right] \right\|_F$$

$$+ \eta \left\| \nabla C^{(i)}(K_n) \right\|_F \left\| \frac{1}{ML} \sum_{j=1}^{M} \sum_{l=0}^{L-1} \left[ \nabla C^{(j)}(K_{n,l}^{(j)}) - \nabla C^{(j)}(K_n) \right] \right\|_F$$

$$+ \eta \left\| \nabla C^{(i)}(K_n) \right\|_F \left\| \frac{1}{M} \sum_{j=1}^{M} \left[ \nabla C^{(j)}(K_n) - \nabla C^{(i)}(K_n) \right] \right\|_F - \eta \left\| \nabla C^{(i)}(K_n) \right\|_F^2$$

$$+ \frac{h_{\mathrm{grad}}(K_n)}{2} \left\| \frac{\eta}{ML} \sum_{j=1}^{M} \sum_{l=0}^{L-1} \tilde{\nabla} C^{(j)}(K_{n,l}^{(j)}) \right\|_F^2$$

$$\overset{(b)}{\leq} \frac{\eta}{4}\left\|\nabla C^{(i)}(K_n)\right\|_F^2 + \eta\left\|\frac{1}{ML}\sum_{j=1}^{M}\sum_{l=0}^{L-1}\left[\tilde{\nabla}C^{(j)}(K_{n,l}^{(j)}) - \nabla C^{(j)}(K_{n,l}^{(j)})\right]\right\|_F^2$$

$$+ \frac{\eta}{8}\left\|\nabla C^{(i)}(K_n)\right\|_F^2 + \frac{2\eta h_{\text{grad}}(K_n)^2}{ML}\sum_{j=1}^{M}\sum_{l=0}^{L-1}\left\|K_{n,l}^{(j)} - K_n\right\|_F^2$$

$$+ \frac{\eta}{4}\left\|\nabla C^{(i)}(K_n)\right\|_F^2 + \frac{\eta}{M}\sum_{j=1}^{M}\left\|\nabla C^{(j)}(K_n) - C^{(i)}(K_n)\right\|_F^2$$

$$- \eta\left\|\nabla C^{(i)}(K_n)\right\|_F^2 + \frac{h_{\text{grad}}(K_n)}{2}\left\|\frac{\eta}{ML}\sum_{j=1}^{M}\sum_{l=0}^{L-1}\tilde{\nabla}C^{(j)}(K_{n,l}^{(j)})\right\|_F^2,$$

where $(a)$ is due to Cauchy–Schwarz inequality; and $(b)$ is due to Cauchy–Schwarz inequality and Eq.(7). Moreover, we have

$$C^{(i)}(K_{n+1}) - C^{(i)}(K_n)$$

$$\overset{(b)}{\leq} \frac{\eta}{4}\left\|\nabla C^{(i)}(K_n)\right\|_F^2 + \eta\left\|\frac{1}{ML}\sum_{j=1}^{M}\sum_{l=0}^{L-1}\left[\tilde{\nabla}C^{(j)}(K_{n,l}^{(j)}) - \nabla C^{(j)}(K_{n,l}^{(j)})\right]\right\|_F^2$$

$$+ \frac{\eta}{8}\left\|\nabla C^{(i)}(K_n)\right\|_F^2 + \frac{2\eta h_{\text{grad}}(K_n)^2}{ML}\sum_{j=1}^{M}\sum_{l=0}^{L-1}\left\|K_{n,l}^{(j)} - K_n\right\|_F^2$$

$$+ \frac{\eta}{4}\left\|\nabla C^{(i)}(K_n)\right\|_F^2 + \frac{\eta}{M}\sum_{j=1}^{M}\left\|\nabla C^{(j)}(K_n) - C^{(i)}(K_n)\right\|_F^2$$

$$- \eta\left\|\nabla C^{(i)}(K_n)\right\|_F^2 + \frac{h_{\text{grad}}(K_n)}{2}\left\|\frac{\eta}{ML}\sum_{j=1}^{M}\sum_{l=0}^{L-1}\tilde{\nabla}C^{(j)}(K_{n,l}^{(j)})\right\|_F^2$$

$$\overset{(c)}{\leq} -\frac{3\eta}{8}\left\|\nabla C^{(i)}(K_n)\right\|_F^2 + \eta\epsilon + \frac{4\eta^2\bar{h}_{\text{grad}}^2}{\eta_g M}\sum_{j=1}^{M}\left[\left\|\nabla C^{(j)}(K_n)\right\|_F^2 + ML\epsilon\right]$$

$$+ \eta\min\{n_x, n_u\}(\epsilon_1\bar{h}_{\text{het}}^1 + \epsilon_2\bar{h}_{\text{het}}^2)^2 + \frac{h_{\text{grad}}(K_n)}{2}\left\|\frac{\eta}{ML}\sum_{j=1}^{M}\sum_{l=0}^{L-1}\tilde{\nabla}C^{(j)}(K_{n,l}^{(j)})\right\|_F^2, \tag{41}$$

where $(b)$ follows from the gradient Lipschitz property in Lemma 1; and $(c)$ follows from the policy gradient heterogeneity property in Lemma 3, Lemma 4 and Lemma 17.

Following the analysis in Eq.(41), we have

$$C^{(i)}(K_{n+1}) - C^{(i)}(K_n)$$

$$\overset{(d)}{\leq} -\frac{3\eta}{8}\left\|\nabla C^{(i)}(K_n)\right\|_F^2 + \eta\epsilon + \frac{4\eta^2\bar{h}_{\text{grad}}^2}{\eta_g M}\sum_{j=1}^{M}\left[\left\|\nabla C^{(j)}(K_n)\right\|_F^2 + ML\epsilon\right]$$

$$+ \eta\min\{n_x, n_u\}(\epsilon_1 h_{\text{het}}^1 + \epsilon_1 h_{\text{het}}^2)^2$$

$$+ \frac{4\eta^2\bar{h}_{\text{grad}}}{2}\left\|\frac{1}{ML}\sum_{j=1}^{M}\sum_{l=0}^{L-1}\tilde{\nabla}C^{(j)}(K_{n,l}^{(j)}) - \nabla C^{(j)}(K_{n,l}^{(j)})\right\|_F^2$$

$$+ \frac{4\eta^2\bar{h}_{\text{grad}}}{2}\left\|\frac{1}{ML}\sum_{j=1}^{M}\sum_{l=0}^{L-1}\nabla C^{(j)}(K_{n,l}^{(j)}) - \nabla C^{(j)}(K_n)\right\|_F^2$$

$$+ \frac{4\eta^2\bar{h}_{\text{grad}}}{2M}\sum_{j=1}^{M}\left\|\nabla C^{(j)}(K_n) - \nabla C^{(i)}(K_n)\right\|_F^2 + \frac{4\eta^2\bar{h}_{\text{grad}}}{2}\left\|\nabla C^{(i)}(K_n)\right\|_F^2$$

$$\overset{(e)}{\leq} -\frac{3\eta}{8}\left\|\nabla C^{(i)}(K_n)\right\|_F^2 + (\eta + 2\eta^2\bar{h}_{\text{grad}})\epsilon + \frac{4\eta^2\bar{h}_{\text{grad}}^2}{\eta_g M}\sum_{j=1}^{M}\left[\left\|\nabla C^{(j)}(K_n)\right\|_F^2 + ML\epsilon\right]$$

$$+ (\eta + 2\eta^2\bar{h}_{\text{grad}})\min\{n_x, n_u\}(\epsilon_1 h_{\text{het}}^1 + \epsilon_2 h_{\text{het}}^2)^2 + 2\eta^2\bar{h}_{\text{grad}}\left\|\nabla C^{(i)}(K_n)\right\|_F^2$$

$$+ \frac{4\eta^2\bar{h}_{\text{grad}}}{2}\left\|\frac{1}{ML}\sum_{j=1}^{M}\sum_{l=0}^{L-1}\nabla C^{(j)}(K_{n,l}^{(j)}) - \nabla C^{(j)}(K_n)\right\|_F^2$$

$$\overset{(f)}{\leq} -\left(\frac{3\eta}{8} + 2\eta^2\bar{h}_{\text{grad}}\right)\left\|\nabla C^{(i)}(K_n)\right\|_F^2 + (\eta + 2\eta^2\bar{h}_{\text{grad}})\epsilon$$

$$+ \frac{4\eta^2\bar{h}_{\text{grad}}^2}{\eta_g M}\sum_{j=1}^{M}\left[\left\|\nabla C^{(j)}(K_n)\right\|_F^2 + ML\epsilon\right] + (\eta + 2\eta^2\bar{h}_{\text{grad}})\min\{n_x, n_u\}(\epsilon_1 h_{\text{het}}^1 + \epsilon_2 h_{\text{het}}^2)^2$$

$$+ \frac{4\eta^2\bar{h}_{\text{grad}}^2}{2ML}\sum_{j=1}^{M}\sum_{l=0}^{L-1}\left\|K_{n,l}^{(j)} - K_n\right\|_F^2$$

$$\overset{(g)}{\leq} -\left(\frac{3\eta}{8} + 2\eta^2\bar{h}_{\text{grad}}\right)\left\|\nabla C^{(i)}(K_n)\right\|_F^2 + (\eta + 2\eta^2\bar{h}_{\text{grad}})\epsilon$$

$$+ \frac{4\eta^2\bar{h}_{\text{grad}}^2 + 4\eta^3\bar{h}_{\text{grad}}^2}{\eta_g M}\sum_{j=1}^{M}\left[\left\|\nabla C^{(j)}(K_n)\right\|_F^2 + ML\epsilon\right]$$

$$+ (\eta + 2\eta^2\bar{h}_{\text{grad}})\min\{n_x, n_u\}(\epsilon_1\bar{h}_{\text{het}}^1 + \epsilon_2\bar{h}_{\text{het}}^2)^2, \tag{42}$$

where $(d)$ is due to Eq.(8); $(e)$ is due to variance reduction property in Lemma 4 and policy gradient heterogeneity in Lemma 3; $(f)$ is due to gradient Lipschitz property in Lemma 1; $(g)$ is due to drift term analysis in Lemma 17.

Continuing the analysis in Eq.(42), we have that

$$C^{(i)}(K_{n+1}) - C^{(i)}(K_n) \leq -\left(\frac{3\eta}{8} + 2\eta^2\bar{h}_{\text{grad}}\right)\left\|\nabla C^{(i)}(K_n)\right\|_F^2 + (\eta + 2\eta^2\bar{h}_{\text{grad}})\epsilon$$

$$+ \frac{4\eta^2\bar{h}_{\text{grad}}^2 + 4\eta^3\bar{h}_{\text{grad}}^2}{\eta_g M}\sum_{j=1}^{M}\left[\left\|\nabla C^{(j)}(K_n)\right\|_F^2 + ML\epsilon\right]$$

$$+ (\eta + 2\eta^2\bar{h}_{\text{grad}})\min\{n_x, n_u\}(\epsilon_1\bar{h}_{\text{het}}^1 + \epsilon_1\bar{h}_{\text{het}}^2)^2$$

$$\overset{(a)}{\leq} -\left(\frac{3\eta}{8} + 2\eta^2\bar{h}_{\text{grad}}\right)\left\|\nabla C^{(i)}(K_n)\right\|_F^2 + (\eta + 2\eta^2\bar{h}_{\text{grad}})\epsilon$$

$$+ \frac{4\eta^2\bar{h}_{\text{grad}}^2 + 4\eta^3\bar{h}_{\text{grad}}^2}{\eta_g M}\sum_{j=1}^{M}\left[2\left\|\nabla C^{(j)}(K_n) - \nabla C^{(i)}(K_n)\right\|_F^2 + 2\left\|\nabla C^{(i)}(K_n)\right\|_F^2 + ML\epsilon\right]$$

$$+ (\eta + 2\eta^2\bar{h}_{\text{grad}})\min\{n_x, n_u\}(\epsilon_1\bar{h}_{\text{het}}^1 + \epsilon_1\bar{h}_{\text{het}}^2)^2$$

$$\overset{(b)}{\leq} -\left(\frac{3\eta}{8} + 2\eta^2\bar{h}_{\text{grad}} + \frac{8\eta^2\bar{h}_{\text{grad}}^2 + 8\eta^3\bar{h}_{\text{grad}}^2}{\eta_g}\right)\left\|\nabla C^{(i)}(K_n)\right\|_F^2$$

$$+ \left(\eta + 2\eta^2\bar{h}_{\text{grad}} + \frac{4\eta^2\bar{h}_{\text{grad}}^2 + 4\eta^3\bar{h}_{\text{grad}}^2}{\eta_g}ML\right)\epsilon$$

$$+ \left(\eta + 2\eta^2\bar{h}_{\text{grad}} + \frac{8\eta^2\bar{h}_{\text{grad}}^2 + 8\eta^3\bar{h}_{\text{grad}}^2}{\eta_g}\right)\min\{n_x, n_u\}(\epsilon_1\bar{h}_{\text{het}}^1 + \epsilon_2\bar{h}_{\text{het}}^2)^2$$

$$\overset{(c)}{\leq} -\frac{\eta}{4}\left\|\nabla C^{(i)}(K_n)\right\|_F^2 + 2\eta\epsilon + 2\eta\min\{n_x, n_u\}(\epsilon_1\bar{h}_{\text{het}}^1 + \epsilon_2\bar{h}_{\text{het}}^2)^2$$

$$\overset{(d)}{\leq} -\frac{\eta\mu^2\sigma_{\min}(R)}{\left\|\Sigma_{K_i^*}\right\|}(C^{(i)}(K_n) - C^{(i)}(K_i^*)) + 2\eta\epsilon + 2\eta\min\{n_x, n_u\}(\epsilon_1\bar{h}_{\text{het}}^1 + \epsilon_2\bar{h}_{\text{het}}^2)^2, \tag{43}$$

where $(a)$ is due to Eq.(8); $(b)$ is due to policy gradient heterogeneity in Lemma 3; and $(c)$ is due to the choice of step-size such that $\frac{3\eta}{8} + 2\eta^2\bar{h}_{\text{grad}} + \frac{8\eta^2\bar{h}_{\text{grad}}^2 + 8\eta^3\bar{h}_{\text{grad}}^2}{\eta_g} \leq \frac{\eta}{4}$ and

$$\eta + 2\eta^2\bar{h}_{\text{grad}} + \frac{8\eta^2\bar{h}_{\text{grad}}^2 + 8\eta^3\bar{h}_{\text{grad}}^2}{\eta_g} \leq 2\eta,$$

which holds when $\eta \leq \min\{\frac{1}{32\bar{h}_{\text{grad}}}, 1\}$ and $\eta_l \leq \frac{1}{256L\bar{h}_{\text{grad}}^2}$; for $(d)$ we use the gradient domination lemma in Lemma 2.

In conclusion, we have that

$$C^{(i)}(K_{n+1}) - C^{(i)}(K_i^*) \leq \left(1 - \frac{\eta\mu^2\sigma_{\min}(R)}{\left\|\Sigma_{K_i^*}\right\|}\right)(C^{(i)}(K_n) - C^{(i)}(K_i^*)) + 2\eta\epsilon$$
$$+ 2\eta\min\{n_x, n_u\}(\epsilon_1\bar{h}_{\text{het}}^1 + \epsilon_2\bar{h}_{\text{het}}^2)^2,$$

holds when the step-size, smoothing radius, trajectory length, and sample size satisfy the requirements mentioned above and those in Lemma 16 and Lemma 17. □

With this lemma, we are now ready to provide the convergence guarantees for the `FedLQR` under the model-free setting.

**Proof of the iterative stability guarantees of `FedLQR`:** Here, we leverage the method of induction to prove `FedLQR`'s iterative stability guarantees. First, we start from an initial policy $K_0 \in \mathcal{G}^0$. At round $n$, we assume $K_n \in \mathcal{G}^0$. According to Lemma 16, we have that all the local policies $K_{n,l}^{(i)} \in \mathcal{G}^0$. Furthermore, frame the hypotheses of in Lemma 18, we have that

$$C^{(i)}(K_{n+1}) - C^{(i)}(K_i^*) \leq \left(1 - \frac{\eta\mu^2\sigma_{\min}(R)}{\left\|\Sigma_{K_i^*}\right\|}\right)(C^{(i)}(K_n) - C^{(i)}(K_i^*)) + 2\eta\epsilon$$
$$+ 2\eta\min\{n_x, n_u\}(\epsilon_1\bar{h}_{\text{het}}^1 + \epsilon_2\bar{h}_{\text{het}}^2)^2.$$

Since $(\epsilon_1\bar{h}_{\text{het}}^1 + \epsilon_2\bar{h}_{\text{het}}^2)^2 \leq \bar{h}_{\text{het}}^3$, we have

$$C^{(i)}(K_{n+1}) - C^{(i)}(K_i^*) \leq \left(1 - \frac{\eta\mu^2\sigma_{\min}(R)}{\left\|\Sigma_{K_i^*}\right\|}\right)(C^{(i)}(K_0) - C^{(i)}(K_i^*)) + 2\eta\epsilon$$
$$+ \frac{\eta\mu^2\sigma_{\min}(R)}{2\left\|\Sigma_{K_i^*}\right\|}(C^{(i)}(K_0) - C^{(i)}(K_i^*))$$
$$\overset{(a)}{\leq} C^{(i)}(K_0) - C^{(i)}(K_i^*),$$

where $(a)$ follows from the fact that $\epsilon$ can be arbitrarily small by choosing a small smoothing radius, sufficient long trajectory length, and enough samples.

With this, we can easily have that the global policy $K_{n+1}$ at the next round $n + 1$ is also stabilizing, i.e., $K_{n+1} \in \mathcal{G}^0$. Therefore, we can finish proving `FedLQR`'s iterative stability property by inductively reasoning.

**Proof of `FedLQR`'s convergence:** From Eq.(39), we have

$$C^{(i)}(K_{n+1}) - C^{(i)}(K_i^*) \leq \left(1 - \frac{\eta\mu^2\sigma_{\min}(R)}{\left\|\Sigma_{K_i^*}\right\|}\right)(C^{(i)}(K_n) - C^{(i)}(K_i^*)) + 2\eta\epsilon + 2\eta\min\{n_x, n_u\}(\epsilon_1\bar{h}_{\text{het}}^1 + \epsilon_2\bar{h}_{\text{het}}^2)^2,$$

Using the above inequality recursively, `FedLQR` enjoys the following convergence guarantee after $N$ rounds:

$$C^{(i)}(K_N) - C^{(i)}(K_i^*) \leq \left(1 - \frac{\eta \mu^2 \sigma_{\min}(R)}{\left\|\Sigma_{K_i^*}\right\|}\right)^N (C^{(i)}(K_0) - C^{(i)}(K_i^*)) + \frac{2\left\|\Sigma_{K_i^*}\right\|}{\mu^2 \sigma_{\min}(R)} \epsilon$$
$$+ \frac{2\min\{n_x, n_u\}\left\|\Sigma_{K_i^*}\right\|}{\mu^2 \sigma_{\min}(R)} (\epsilon_1 \bar{h}_{\text{het}}^1 + \epsilon_2 \bar{h}_{\text{het}}^2)^2.$$

Suppose the trajectory length satisfies $\tau \geq h_\tau \left(\frac{r\epsilon'}{4n_x n_u}\right)$, the smoothing radius satisfies $r \leq h_r' \left(\frac{\epsilon'}{4}\right)$, where

$$h_r' \left(\frac{\epsilon'}{4}\right) := \min \left\{ \frac{\min_{i \in [M]} C^{(i)}(K_0)}{\bar{h}_{\text{cost}}}, \underline{h}_\Delta, h_r \left(\frac{\epsilon'}{4}\right) \right\},$$

and the number of the sample size of each agent $n_s$ satisfies

$$n_s \geq \frac{h_{\text{sample,trunc}} \left(\frac{\epsilon'}{4}, \frac{\delta}{ML}, \frac{H^2}{\mu}\right)}{ML},$$

with $\epsilon' = \frac{4\left\|\Sigma_{K_i^*}\right\|}{\mu^2 \sigma_{\min}(R)}\epsilon$.

When the number of rounds $N \geq \frac{c_{\text{uni},4}\left\|\Sigma_{K_i^*}\right\|}{\eta \mu^2 \sigma_{\min}(R)} \log \left(\frac{2(C^{(i)}(K_0) - C^{(i)}(K_i^*))}{\epsilon'}\right)$, our `FedLQR` algorithm enjoys the following convergence guarantee:

$$C^{(i)}(K_N) - C^{(i)}(K_i^*) \leq \left(1 - \frac{\eta \mu^2 \sigma_{\min}(R)}{\left\|\Sigma_{K_i^*}\right\|}\right)^N (C^{(i)}(K_0) - C^{(i)}(K_i^*)) + \frac{\epsilon'}{2}$$
$$+ \frac{2\min\{n_x, n_u\}\left\|\Sigma_{K_i^*}\right\|}{\mu^2 \sigma_{\min}(R)} (\epsilon_1 h_{\text{het}}^1 + \epsilon_2 h_{\text{het}}^2)^2$$
$$\leq \epsilon' + \frac{2\min\{n_x, n_u\}\left\|\Sigma_{K_i^*}\right\|}{\mu^2 \sigma_{\min}(R)} (\epsilon_1 h_{\text{het}}^1 + \epsilon_2 h_{\text{het}}^2)^2,$$

which completes the proof.

