# OpenReview forum: "Model-Free Learning with Heterogeneous Dynamical Systems: A Federated LQR Approach"
_TMLR — Accepted by TMLR_

### Review · Reviewer_i6JD · 2025-10-24

**Summary Of Contributions:**

The paper considers the Linear Quadratic Regulator (LQR) problem in a federated setting with M agents. Each of the agents have potentially different dynamics. The objective of the paper is to study the convergence under such heterogeneous conditions and to quantify the impact of heterogeneity on the final error and the rate of convergence. The paper studies this for the class policy gradient algorithms that aim to learn a common policy for all the agents.

The authors show that under both model-based and model-free approaches their proposed algorithm is stable and converges. The authors establish the rate of convergence and show that the final error has a bias term that depends on the heterogeneity among the models. They also demonstrate that under the model-free setting, their proposed approach offers a linear speedup in the error demonstrating the benefit of collaboration.

Lastly, the authors corroborate their theoretical results with empirical studies and show that their proposed algorithm performs well under synthetic settings.

Strengths:
- One of the key strengths of this paper is that it well-written. It was refreshing to see a nicely written paper. There are some minor typos here and there but nothing that a careful read can't fix.
- The paper clearly demonstrates how heterogeneity affects convergence and its role in the final error bounds. This provides a clear understanding of the impact of heterogeneity and hence helps understand when collaboration can be beneficial for learning.

Weakness:
- I think this paper falls a bit short on the novelty front. Typically in a theory paper like this, the main contribution comes from either the algorithm or the analysis. Clearly, the algorithm is a standard FedAvg template. In terms of analysis, it relies heavily on existing results from Fazel et al 2018 and Gravell et al 2020 and techniques in FL algorithms that are largely standard by now. I am not saying that there is no novelty at all. The paper seems more applying some well-known techniques to a different problem setting. Also, at times the authors seemed to have overclaimed their contributions (I have outlined it in the next section).

**Audience:**

Yes

**Audience Explanation:**

Yes, I think the paper is relevant to TMLR audience. It deals with Federated Learning in LQR systems and is potentially interesting to both the control and FL communities.

**Broader Impact Concerns:**

I think there is no particular need for a broader impact section for this work.

**Claims And Evidence:**

Yes

**Claims Explanation:**

Yes, I have gone through the proofs and they look correct to me (except for some minor comments outlined below).

**Requested Changes:**

I will add both suggestions and questions that I have to this section. I feel all of them are critical at this point.

1.  I am a bit confused about definition 2. From what I understand there has to be a $\beta$ dependence in the definition. Either $\mathcal{G}^{(i)}$ has to actually be defined as $\mathcal{G}^{(i)}(\beta)$, i.e., a function of a $\beta$. Or the definition has to be amended to $\exists$ a $\beta > 0$ such that the condition holds. I believe the authors are going for the first option, but it is not totally clear from the text following the paragraph. I would suggest the authors to be explicit about it. I would refer to it as $\mathcal{G}_0(\beta)$ going forward. I believe the second option is not the correct way to define it.
2.  The claim "...which only guarantee convergence to stationary points, our work investigates whether FedLQR can find a globally optimal policy." is a bit misleading in my opinion. Assumption 2, which states that you have a "good" $K_0$ to start with is akin to assuming that start "close enough" to the global optimizer. The closeness is determined by $\beta$. Furthermore, Lemma 2 is akin to the PL condition in optimization literature, under which convergence to global optimum has been discussed.
3. How is "sample randomness" in your setting different from the combination of "initial condition randomness" and "randomness from the smoothing matrices that show up in the gradient estimation process". From what I understand, you refer "sample randomness" to the uncertainty in gradient estimates. In your case, the uncertainty of gradient estimate is just the uncertainty its estimation which is a combination of initial condition and Alg 2.  In other words, if you had an oracle that gave you a "sample" of your gradient, then your problem reduces to the "sample randomness" scenario. The only difference that now that your oracle is not a blackbox and you outline how it works and hence also characterize its uncertainty. Yet again, this seems to be a bit misleading, or at least overclaiming.
4. It is important to specify that $\hat{\nabla}C(K)$ is a *biased* estimated of the true gradient and the bias comes from the smoothing steps. While the error bound that is adopted from (Gravell et al., 2020) takes care of the bias term, there is no mention of the same in this paper. This also lies at the heart of the choice of $r$, which has not been discussed much in the paper.
5. Can you elaborate how Eq 22 comes about? I know it is standard bound when then gradient is $L$-Lipschitz. However, in such cases the Lipschitz constant is independent of $x$, the decision variable or argument of the function. That is no longer true for this particular case. The correct way to derive that would be to go by a Talyor series expansion, which reads as $f(x) = f(x_0) + f'(x_0)(x-x_0) + f''(x_1)(x- x_0)^2/2$, which $x_0$ is the point around which we perform the expansion and $x_1$ is some point between $x$ and $x_0$. Typically, since we have $x$ independent bound, we have $|f''(x_1)| \leq L$ and we obtain the final result. However, in case how does this bound come about? It might possible to replace it with $\overline{h}_{grad}$ but for that we need to ensure that the point at which the residual is calculated is within $\mathcal{G}_0(\beta)$. This is a relation that has been used multiple times so this question applies everywhere.
6. The claim on page 32 that $K\_{n,1}^{(i)} \in \mathcal{G}\_0$ is again a bit ambiguous. In particular, the equation $C^{(j)}(K_{n,1}^{(i)}) - C^{(j)}(K_n) \leq C^{(j)}(K_0) - C^{(j)}(K_j^*)$ implies that if $K\_n \in \mathcal{G}\_0(\beta)$, then $K_{n,1}^{(i)} \in \mathcal{G}_0(\beta + 1)$, i.e., the $\beta$ changes. There are several points to be noted here. Firstly, this clearly shows why the definition in remark #1 is critical. Secondly, it is incorrect to keep allowing growing $\beta$'s across iterations because you want the iterates to remain bounded in a certain, *fixed* set to complete your convergence arguments. With that said, it should not be difficult to fix this, as long as you invoke the correct value of $\beta$ in Eqn 25. This will automatically force a bound on $\beta$. This will circle back to the point I raised in remark #2 on how $\beta$ affects the initial condition.
7. Eqn 33 as currently written is incorrect. The term $T_6$, with $ML\epsilon$ in the denominator, should have just $L\epsilon$ in the denominator. There are two ways to fix this. First, you restrict $\epsilon < 1/M$, which would yet again make $T_5$ the dominant term. Second, you carefully apply the Freedman to $ML$ terms instead of $L$, and you will have $M$ in the denominator in $T_6$ as well.
8. Can the authors elaborate on the choice of $L$? I could not find any discussion on how it chosen or how it affects convergence or communication costs? Typically in FL, there is a limit on how large $L$ can be because it affects the drift terms.
9. The final bound in Corollary 1 (and equivalently in the model-free case) is quite similar to that of strongly convex and smooth functions in optimization literature. Is it possible to obtain $\mathcal{O}(1/N)$ convergence bounds with no bias terms, similar to the optimization literature. Such a result would definitely add value to this paper.

---

> ### Author Response · Authors · 2025-11-23
> **Official Comments by the Authors to Reviewer i6JD (1/3)**
>
> **On the Weakness**: We thank the Reviewer for the opportunity to restate the core challenges in our work. While the FedAvg template is indeed standard, the core technical challenges in the *federated LQR* setting differ substantially from those in classical supervised FL and from the single-agent LQR policy gradient literature. Our contributions are not a direct transposition of existing techniques, but rather a rigorous treatment of some difficulties that arise only when federated optimization is combined with heterogeneous dynamical systems.
>
> First, even in the single-agent setting, policy gradient for LQR is a nonconvex problem requiring intricate analysis (e.g., Fazel et al. 2018; Gravell et al. 2020). Extending these guarantees to the multi-agent heterogeneous setting is not straightforward, as the average of multiple system-specific PL costs is not necessarily PL (i.e., gradient dominance no longer holds). Second, unlike standard FL, we must guarantee *stability* (i.e., the long-term behavior) of every closed-loop system at every aggregation step. This requirement is unique to control and becomes significantly more challenging under system heterogeneity, where the systems need to be sufficiently similar for a common stabilizing set of controllers to exist. To the best of our knowledge, ensuring that policies remain simultaneously stabilizing for multiple distinct systems has remained an open problem.
>
> Third, practical FL methods rely on multiple local gradient steps between communication rounds. In heterogeneous settings, such steps introduce a "client-drift" effect toward each agent's local minimizer. While client drift has been well studied in supervised FL, its interplay with *closed-loop stability* has not been analyzed before. In our work, we provide a detailed drift analysis and show how to control these effects so that the aggregated controller remains stabilizing for all systems throughout learning.
>
> Beyond addressing these challenges, our contributions include (i) per-iteration stability guarantees, (ii) a gradient heterogeneity bound derived directly from differences in system dynamics, and (iii) a rigorous sample-complexity reduction result that quantifies the benefits of collaboration under heterogeneity. These elements are not present in prior work.
>
> Finally, we note that our framework has already inspired subsequent works that extend policy gradient and federated learning to other settings [1]-[6].
>
> In passing, we do not see any immediate disadvantage to an algorithm being of the FedAvg template. Given that this is the simplest and most popular template for federated learning algorithms, it made sense for us to first fully characterize the behavior of such an algorithm in the dynamical systems context, before moving on to more advanced algorithms (for instance, ones that use control variates). As described above, the analysis of even such a "simple" algorithm poses numerous technical challenges that we systematically overcome in this paper. It should be noted that for the same reasons as above, algorithms that fall into the standard FedAvg template have been recently explored extensively for MDPs with finite state-action spaces; see, for instance, [7] and [8].  Notably, the issue of stability does not arise in these papers either, and is unique to the setting we study in our work.
>
> [1] Weber et al. (2024). Combining federated learning and control: A survey. IET Control Theory & Applications.
>
> [2] Ye et al. (2024). On the Convergence of Policy Gradient for Designing a Linear Quadratic Regulator by Leveraging a Proxy System. IEEE CDC.
>
> [3] Toso et al. (2024). Meta-learning linear quadratic regulators: a policy gradient MAML approach for model-free LQR. L4DC.
>
> [4] Zhao et al. (2025). Asynchronous Parallel Policy Gradient Methods for the Linear Quadratic Regulator. IEEE TAC.
>
> [5] Fujinami et al. (2025). Policy Gradient for LQR with Domain Randomization. arXiv.
>
> [6] Stamouli et al (2025). Policy Gradient Bounds in Multitask LQR. IEEE LCSS.
>
> [7] Khodadadian et al. (2022). Federated reinforcement learning: Linear speedup under markovian sampling. In International Conference on Machine Learning. PMLR.
>
> [8] Woo et al. (2023). The blessing of heterogeneity in federated Q-learning: Linear speedup and beyond. In International Conference on Machine Learning. PMLR.
>
> **On Definition 2**: We thank the Reviewer for pointing out this ambiguity. The intended notation is $\mathcal{G}^{(i)}(\beta)$ for some $\beta > 0$, we we choose $\beta$ sufficiently large depending on $K_0$ and it does not vary with iterations in our analysis.
>
> **On the initial stabilizing controller**:  We reemphasize that Assumption 2 is not that $K_0$ must be "close" to the global optimizer in terms of cost, but only that it is *stabilizing* for all systems. This requirement is standard in policy gradient for LQR and does not impose closeness to $K^\star$ (it can be suboptimal). Please refer to our general comment.

---

> > ### Comment · Reviewer_i6JD · 2025-12-05
> > **Response to authors**
> >
> > Thank you for your detailed response. It resolves most of my concerns.
> >
> > - **Regarding PL of average cost**: I see, I had not realized that distinction and sort of implicitly assumed that the condition on individual costs implies that it holds for the average cost as well. Is there a formal proof for that, or at least a high-level intuitive idea? In general, it does seem that it would be necessary but why is it not sufficient? I understand I am responding a bit late, so a formal proof might be too much of an ask, but an intuitive idea would definitely help me understand the challenges a bit better.
> >
> > - **Regarding $\beta$ and Lipschitz condition**: I agree with the authors, that $\beta$ is held fixed and appropriately chosen to ensure that it is not updated as iterations continue. This is not reflected in the analysis in latest version but I am assuming that the authors will do so in the next version. Moreover, I agree that Eqn. (1) (in the next rebuttal comment) is the correct version of the equation and it should replace Eqn. 22 (and similar ones) in the main paper.
> >
> > - **Regarding Eqn.(33)**: If the authors wish, they can remove the restriction $\varepsilon < 1/M$ using the careful application of Freedman I mentioned in my previous comment. $\varepsilon < 1/M$  seems a bit restrictive in my opinion which can be fixed relatively easily.

---

> ### Author Response · Authors · 2025-11-23
> **Official Comments by the Authors to Reviewer i6JD (2/2)**
>
> **On the sample randomness**: We thank the Reviewer for raising this point. In our setting, "sample randomness" refers to the uncertainty in the initial conditions and the gradient estimate arising from the trajectory rollouts used in Algorithm~2. We now make this explicitly in the revised manuscript.
>
> The reason we use the term "sample randomness" is that our analysis also extends naturally to settings with process noise, where the randomness in the gradient estimate comes from stochastic system evolution rather than only from initialization or perturbations. In such settings, even with a fixed initial condition and no smoothing perturbations, trajectory-based gradient estimates remain random.
>
> **On the biased estimate**: The one-point zeroth-order estimator $\widehat{\nabla C}(K)$ is indeed a biased estimator of the true gradient, and that the bias arises from the smoothing step. We have added a sentence on page 38 clarifying this and emphasizing that the choice of the smoothing radius $r$ controls the estimation error. In Appendix 8.2, we provide the precise conditions on $r$ and the number of samples needed to ensure that the estimation error remains within the tolerance required for our convergence analysis.
>
> **On the Lipschitz condition**: We appreciate this intricate question. To understand why Eq. 22 holds, we note that the LQR cost function satisfies a $(\phi, \rho)$ local-smoothness property (see [1] below). To be more precise, for a stable controller $K$, there exists a radius $\rho(K) > 0$ and a smoothness constant $\phi(K)$ such that for all $K'$ satisfying $\Vert K - K' \Vert_F \leq \rho(K)$, we have $\Vert \nabla C(K) - \nabla C(K') \Vert_F \leq \phi(K) \Vert K - K' \Vert_F$; here, we have dropped superscripts related to agent indices for simplicity of exposition. Now although the parameters $\rho(K)$ and $\phi(K)$ depend on the argument $K$ itself (as the Reviewer astutely notes), our analysis is always restricted to the common stabilizing sublevel set $\mathcal{G}_0(\beta)$, on which, $\rho(K)$ is bounded from below by some $\rho>0$, and $\phi(K)$ is bounded from above by some $\bar{h}_g < \infty$.   As such, fixing any $K$ in $\mathcal{G}_0(\beta)$, and $K' $ such that $\Vert K - K' \Vert_F \leq \rho$, one can then easily show the following property:
>
> $
> C(K') \leq C(K) + \langle \nabla C(K), K' - K \rangle  + \frac{\bar{h}_{g}}{2} ||K' - K||_F^2 \quad (1)
> $
>
> The proof of the above property is a direct application of Taylor's theorem and proceeds exactly in the same way as is done for optimization; in particular, it uses the fact that any $K''$ in the line segment connecting $K$ and $K'$ is also within a $\rho$-radius of $K$. Having explained how Eq. (1)  applies to our setting, now let us explain our high-level proof strategy for Lemma 10. Our analysis proceeds via induction where the initial controller $K_0$ is within $\mathcal{G}_0(\beta)$, and the step-size is chosen to ensure that the next iterate $K_1$ is within a $\rho$-radius of $K_0$. Thus, (1) becomes applicable, and we can establish a descent-type inequality. Under suitable assumptions on the heterogeneity level (which we interpret as a perturbation) and choice of the step-size, one can then argue that the cost at the new iterate $K_1$ is such that $K_1$ continues to belong to $\mathcal{G}_0(\beta)$. We then repeat this argument iteratively.
>
> [1] Malik et al. (2020). Derivative-free methods for policy optimization: Guarantees for linear quadratic systems. JMLR.
>
> **On the per-iteration stabilizing analysis**:   We emphasize that our intention is*not* to let $\beta$ grow across iterations, but rather to fix a stabilizing sublevel set $\mathcal{G}_0(\beta)$ and prove that all iterates remain in this set (and, in fact, move closer to the optimal controllers).  In what follows, we explain how our per-iteration descent analysis ensures that the next controller $K_1$ continues to remain in *precisely the same set*  $\mathcal{G}_0(\beta)$, i.e., with the same $\beta$. Given that $K_0 \in \mathcal{G}_0(\beta),$ and that our step-size is chosen to satisfy the local smoothness condition, we can obtain the following one-step progress recursion for each agent $j$:
>
> $
> C^{(j)}(K_{1}) - C^{(j)}(K_j^\star)
> \leq \rho \big( C^{(j)}(K_0) - C^{(j)}(K_j^\star) \big) + (I) + (II),
> $
>
> where  $(I)$ and  $(II)$ denotes heterogeneity and estimation terms. Also, $\rho \in (0,1)$. By making $(I)$ and  $(II)$ sufficiently small we ensure that
>
> $
> C^{(j)}(K_{1}) - C^{(j)}(K_j^\star) \leq 3\rho \left(C^{(j)}(K_0) - C^{(j)}(K_j^\star)\right) \leq  \beta \left(C^{(j)}(K_0) - C^{(j)}(K_j^\star)\right),
> $
>
> if choose $\beta \geq 3\rho$, $K_{1}$ remains in $\mathcal{G}_0(\beta)$. Thus, $\beta$ is fixed and does not need to increase with the iterations.
>
> We then emphasize that $\beta$ is chosen once (based on the initial controller), and note that the forward invariance of $\mathcal{G}_0(\beta)$ is guaranteed by $\rho$ together with the bounds on $(I)$ and $(II)$.

---

> > ### Author Response · Authors · 2025-11-23
> > **Official Comments by the Authors to Reviewer i6JD (3/3)**
> >
> > **On the Eq. (33)**: The Reviewer is correct,  as written, Eq. (33) is inconsistent with the preceding bound: the term $T_6$ should have $L \epsilon$ in the denominator (as we invert Eq.(32)), not $ML \epsilon$. In the revised manuscript, we now correct this typo and explicitly impose the restriction $\epsilon < 1/M$, under which $T_5$ remains the dominant term in the final bound.
> >
> > **On the choice of L**: The choice of $L$ follows the standard communication–drift trade-off in federated optimization. A larger number of local steps $L$ reduces communication but increases the drift between the local updates and the global gradient, while a smaller $L$ improves alignment with the global direction at the expense of more global communication rounds.
> >
> > In our analysis, $L$ appears in the drift term and must be chosen so that the combined effect of drift, heterogeneity, and gradient-estimation error remains within the tolerance that preserves stability and ensures per-round descent. We also emphasize that in Lemma 11 (local drift analysis) we need the local step-size $\eta_l$ to scale inversely with $L$, implying $L$ cannot be arbitrarily large. We now add a short discussion in the main text summarizing this trade-off and referencing the precise condition in the appendix.
> >
> > **On the bound in Corollary 1**:  Obtaining an $\mathcal{O}(1/N)$ rate would require a PL or strong-convexity-type property for the *average* cost. Unlike the typical optimization setting where averaging strongly convex and smooth functions preserves strong convexity, the average LQR cost across heterogeneous systems does *not* satisfy a PL condition in general. The reason for this stems from the fact that for the LQR problem, the PL condition is associated with a dynamical system, and the average cost cannot be directly related to any dynamical system (as far as we can tell). This issue is unique to our control setting, and does not arise in optimization. This is precisely why our analysis yields convergence ``up to" a heterogeneity-dependent bias rather than a global linear rate.

---

> ### Author Response · Authors · 2025-12-06
>
> **Regarding PL of average cost:** We thank the Reviewer for the follow-up comments. We give the following intuition regarding the lack of gradient dominance for the average cost when the system are heterogeneous. Each individual LQR cost $C^{(i)}(K)$ satisfies a PL (gradient-dominance) inequality
>
> $
> ||\nabla C^{(i)}(K)||^2  \leq 2\mu_i\big(C^{(i)}(K) - C^{(i)}(K_i^\star)\big),
> $
>
> where this inequality is derived from system-theoretic properties of the underlying dynamical system $(A^{(i)},B^{(i)})$ (see Lemma 11 in Fazel et al. ICML, 2018). However, the average cost
>
> $
> C_{\mathrm{avg}}(K) = \frac{1}{M}\sum_{i=1}^M C^{(i)}(K)
> $
>
> does not in general correspond to the LQR cost of any dynamical system unless all agents share identical system matrices. The PL property for LQR relies crucially on the Riccati structure of a single system, once the dynamics differ, the averaged cost loses this structure, and the PL inequality cannot be carried over. Thus, even though each $C^{(i)}$ is PL, their average need not be. This is because when you average functions with different minimizers, the gradients may cancel, while suboptimality remains positive, which would violate the PL condition.
>
> We also point the Reviewer to [1], where an "approximate" PL condition is obtained for the average cost under a very strict low-heterogeneity regime (almost zero heterogeneity in the system matrices in practice).
>
> Moreover, the authors in [2] have shown that in the setting where the system dynamics are identical, but the costs differ, the PL condition for the average cost is achieved. This is true because for the system homogeneous setting the average cost is associated to solving the LQR problem for the tuple $(A, B, \bar{Q}, \bar{R})$, where $\bar{Q}= (1/M) \sum_{i \in [M]} Q^{(i)}$ and $\bar{R}= (1/M) \sum_{i \in [M]} R^{(i)}$ are the averaged cost matrices.
>
> [1] Fujinami, T., Lee, B. D., Matni, N., Pappas, G. J. (2025). Policy Gradient for LQR with Domain Randomization. arXiv:2503.24371.
>
> [2] Zhu, F., Heath, R. W., Mitra, A. (2024). Towards fast rates for federated and multi-task
> reinforcement learning. Proceedings of the 63rd IEEE Conference on Decision and Control (CDC).
>
>
> **Regarding $\beta$ and Lipschitz condition:** Yes, in the revised version, we will include a sentences on $\beta$ being fixed and appropriately chosen to ensure that it is not updated throughout iterations and replace Eqn. 22 (and similar ones) by Eqn. (1) pointed out in our previous comments.
>
> **Regarding Eqn.(33):** We thank again the Reviewer for this suggestion and careful reading all of our proofs. We will remove this restriction on $\epsilon$ in the revised version.

---

> > ### Comment · Reviewer_i6JD · 2025-12-07
> > **Response to authors**
> >
> > Thank you for your response. This address all my comments.

---

### Review · Reviewer_cDQa · 2025-11-02

**Summary Of Contributions:**

The paper proposes FedLQR, a federated learning framework for linear quadratic regulation (LQR) across multiple linear time-invariant systems with distinct yet similar dynamics. Building on the policy gradient (PG) approach to model-free control, the authors develop a federated variant that allows agents to collaboratively learn a common feedback policy while keeping their data private. The goal is to exploit inter-agent similarity to accelerate convergence and improve sample efficiency compared to learning individually. Theoretical analysis establishes conditions under which the federated policy remains stabilizing for all agents, quantifies its deviation from each agent’s optimal controller (in terms of the cost incurred), and proves a reduction in sample complexity proportional to the number of participating agents.

The article is very well-written and sufficiently motivated; the literature review for the state of the art has also been performed thoroughly. All theoretical results are clearly stated, along with the underlying assumptions and supporting results.

**Audience:**

Yes

**Audience Explanation:**

The article will be of particular interest to researchers working at the intersection of optimization, machine learning, and control theory, as it connects federated learning methodologies with classical LQR control problems. Moreover, as highlighted in the motivation section, the work is also relevant to the robotics community, especially in scenarios involving multi-robot systems or distributed control, where agents share similar but not identical dynamics.

**Broader Impact Concerns:**

No concerns related to ethical implications were identified. The work is theoretical and methodological in nature, with experiments conducted on synthetic dynamical systems. There is no indication of ethical risks such as misuse of data, privacy violations, or societal harms.

**Claims And Evidence:**

Yes

**Claims Explanation:**

The claims made in the contribution have been established rigorously through theoretical analysis with clear and sufficient proofs. The same has also been demonstrated through numerical experiments.

**Requested Changes:**

1. The convergence guarantees in Theorem 1 and Corollary 1 are expressed in terms of the cost difference relative to the optimal value. The results do not provide explicit statements about the behaviour of the optimizers. The authors briefly acknowledge this limitation after Corollary 1, but the explanation is vague — the phrase “...M systems are close” lacks formal clarity. It would be helpful if the paper explicitly clarified whether this “closeness” refers to the bounded heterogeneity defined in Definition 1 or to a stronger notion that can be derived from system-theoretic properties. A more precise statement would strengthen the interpretation of the convergence results.

2. The authors establish via Propositions 1 and 2 that mere similarity in terms of small norm differences between system matrices does not necessarily ensure similar closed-loop behavior or the existence of a common stabilizing policy. However, this important insight appears relatively late in the paper. Since the question of whether “similar” systems can share a stabilizing policy naturally arises early, it would improve readability to mention this fact, perhaps in the Introduction or Motivation.

3. The paper emphasizes that FedLQR achieves global convergence, in contrast to the first-order stationary point guarantees typically seen in federated learning literature. While this statement is valid for the single-agent LQR problem—where the gradient-dominant property ensures global optimality of policy gradient methods—it is less clear that this property holds once multiple heterogeneous systems are considered jointly. When the objective becomes the average of the individual agents’ costs, the overall function may no longer satisfy the same gradient domination structure. Therefore, is it correct to state that a gradient-based algorithm can reach the global optimiser? It seems that the suboptimality introduced by nonconvexity and heterogeneity is effectively absorbed into the residual “ball” term (whose radius depends on the heterogeneity parameters).

4. The analysis relies on the existence of a common initial stabilizing controller that stabilizes all participating systems (Assumption 2). While this assumption is standard in LQR literature, it is quite strong in a federated setting involving multiple heterogeneous systems. It would be valuable if the authors could discuss practical approaches to obtain such a controller, both in model-based and model-free contexts.

---

> ### Author Response · Authors · 2025-11-23
> **Official Comments by the Authors to Reviewer cDQa**
>
> **On the closeness between $M$ systems**: The term "closeness" indeed refers to the bounded system heterogeneity introduced in Definition 1. All convergence statements in Theorem 1 and Corollary 1 are expressed with respect to this heterogeneity measure, which quantifies how similar the underlying systems are in terms of their system matrices. We have clarified this in the text of the revised version.
>
> **On the open-loop heterogeneity measure**:  We agree that this point is important and arises naturally early in the discussion. In our work, systems are deemed "similar" according to the parameter deviation notion introduced in Definition 1, which measures heterogeneity through norm differences between system matrices. As Propositions 1 and 2 illustrate, such similarity does not imply similar closed-loop behavior nor guarantee the existence of a common stabilizing controller. We now highlight this upfront after Definition 1.
>
> We also note that more recent work refines this notion of similarity by introducing bisimulation-based metrics that capture *closed-loop* heterogeneity [1]. While our paper intentionally focuses on the parameter-space notion for clarity, we have added a reference to this distinction and briefly mention that closed-loop metrics can also be derived in the multitask LQR setting. However we emphasize that the core of our results, i.e., the FedLQR's stability and convergence analysis under system heterogeneity, and the rigorous derivation of the sample-complexity reduction under collaboration, is agnostic to the specific heterogeneity metric employed.
>
> [1] Stamouli, C., Toso, L. F., Tsiamis, A., Pappas, G. J., Anderson, J. (2025). Policy Gradient Bounds in Multitask LQR. IEEE Control Systems Letters.
>
> **On the convergence guarantees**:  We thank the Reviewer for raising this point. You are correct that, once the objective becomes the average cost across heterogeneous systems, the global PL (gradient-dominance) condition no longer holds in general. For this reason, our guarantees are stated *"up to" a heterogeneity-induced bias* (please refer to pages 3, 12, 14 and 16), which is unavoidable whenever the local system dynamics differ. The residual ball in our bounds captures precisely this bias. Thus, we claim convergence to a neighborhood whose radius depends on the system heterogeneity.
>
> Our analysis of Theorem 1 and 2 proceeds by characterizing convergence with respect to each system-specific cost $C^{(i)}(K)$, for which the PL property does hold. Recent work has shown that an approximate PL condition for the *average* cost can be recovered under *very* strong low-heterogeneity assumptions [1], but such assumptions are considerably more restrictive than those required in our setting. Instead, our bounds explicitly retain a heterogeneity bias term, which is consistent with convergence guarantees in heterogeneous federated reinforcement learning [2]-[4].
>
> [1] Fujinami, T., Lee, B. D., Matni, N., Pappas, G. J. (2025). Policy Gradient for LQR with Domain Randomization. arXiv.
>
> [2] Wang, H., Mitra, A., Hassani, H., Pappas, G. J., Anderson, J. (2023). Federated temporal difference learning with linear function approximation under environmental heterogeneity. TMLR.
>
> [3] Zhang, C., Wang, H., Mitra, A., Anderson, J. (2024). Finite-time analysis of on-policy heterogeneous federated reinforcement learning. ICLR 2024.
>
> [4] Jin H, Peng Y, Yang W, Wang S, Zhang Z. (2022) Federated reinforcement learning with environment heterogeneity. InInternational Conference on Artificial Intelligence and Statistics. PMLR.
>
> **On the initial stabilizing controller**: Regarding the assumption on the initial stabilizing controller please refer to our general comment.

---

> > ### Comment · Reviewer_cDQa · 2025-12-08
> >
> > I thank the authors for providing detailed responses to my queries; I am satisfied with them.

---

### Review · Reviewer_gAUU · 2025-11-08

**Summary Of Contributions:**

This paper studies a federated reinforcement-learning setting where $M$ agents, each possessing a distinct but similar linear dynamical system, collaborate to learn a common linear-quadratic regulator (LQR) controller without revealing their raw data. The authors propose a new federated policy-gradient algorithm named $\textbf{FedLQR}$. Each agent performs multiple local updates (using zeroth-order gradient estimates) and then transmits their local model to a central server, which averages them and re-broadcasts a global policy.  The proposed theoretical analysis focuses on three central questions:  (i) Does the common policy remain stabilizing for all systems at every iteration?  (ii) How close is the learned global policy to each agent’s own optimal policy? (iii) Can agents reduce sample complexity by collaborating?. The authors develop a complete finite-time analysis to address these questions. They show that under a low-heterogeneity regime,  $\textbf{FedLQR}$ maintains stability across all systems, converges linearly to a neighborhood of each agent’s optimal controller, and achieves an $\mathcal{O}(1/M)$ per-agent improvement in sample complexity in the model-free setting.

**Strengths**
-  The paper unifies federated learning and control theory in a principled manner. The heterogeneity issue in federated LQR has not been rigorously analyzed before; this is the first rigorous theoretical treatment of federated, model-free control under heterogeneity.

- The theoretical results are fully justified. The proofs are structured into auxiliary lemmas that make them accessible despite their complexity.

- The impossibility results in Section~5 convincingly establishes that the “low heterogeneity” condition is necessary and not an artifact of the analysis. The simple counterexamples illustrating the failure of stabilization under high heterogeneity are well-founded and perfectly illustrate the requirement of low heterogeneity.

 - The analysis links physical system heterogeneity to policy-gradient heterogeneity bounds. This bridge allows reusing tools from single-agent LQR analysis in the federated setting.

-  Combining the variance-reduction lemma with the stability argument, the paper establishes the first provable linear-speedup in sample complexity ($\mathcal{O}(1/M)$) for model-free control with heterogeneous systems. This is an impactful theoretical advance in federated reinforcement learning.

-  The related work section does an excellent job of situating the paper within three relevant research areas: (i) Federated Learning, (ii) Policy-Gradient LQR, and (iii) Federated Reinforcement Learning. This clear positioning highlights how the paper bridges these distinct literatures and underscores its novelty

**Weaknessess**

-  Assumption~2 requires access to an initial controller $K_0$ that simultaneously stabilizes all systems. While this assumption is standard in theoretical LQR analyses, it might pose a practical obstacle in the heterogeneous, model-free setting. The paper only briefly refers to single-agent works capable of identifying such stabilizing policies, without discussing how such initialization might be achieved or approximated in the federated heterogeneous case.

- The experimental section is minimal and restricted to synthetic, low-dimensional (3-state, 3-input) systems. Although the setting is clean and controlled, its small scale limits the ability to assess the robustness and effectiveness of FedLQR under realistic conditions. A more comprehensive evaluation would substantially strengthen the empirical evidence.

- There appear to be inaccuracies in the mathematical statements. In particular, the definition of $\epsilon'$ in Theorem 2 seems inconsistent: based on the first inequality on page 53, $\epsilon'$ should likely be defined as

  $$ \epsilon' = \sqrt{\frac{16 \| \Sigma_{K_i^{\star}} \|}{\mu^2 \sigma_{\min}(R)}\,\epsilon}.
    $$

  Furthermore, in Lemma 18, the bound involving the heterogeneity parameters appears to contain a typographical error. The term
    $2\eta \min \{ n_x, n_u \} (\epsilon_1 h_{het}^1 + \epsilon_1 h_{het}^2)^2$
    should instead read $2\eta \min \{ n_x, n_u \} (\epsilon_1 h_{\text{het}}^1 + \epsilon_2 h_{\text{het}}^2)^2.$
    These corrections are minor but should be addressed.

-  While the analysis clearly characterizes the heterogeneity-induced bias term, the practical impact of this bias on learning performance is not well explored. In particular, when heterogeneity increases, this term may significantly degrade convergence and stability. The authors do not discuss why variance-corrected or control-variate methods such as $\texttt{Scaffold}$[1] are not employed to mitigate this effect, leaving open whether such approaches could reduce the bias without compromising stability guarantees.

- The introduction of the universal constant $c_{\text{uni},2}$ in Theorem~2 seems unnecessary, as its value is explicitly given as $2$. If this constant is fixed, it could be substituted directly to simplify the statement and improve readability.

[1] Karimireddy et al, SCAFFOLD: Stochastic Controlled Averaging for Federated Learning, ICML

**Additional Comments:**

Overall, I find the work technically sound, well written, and of clear interest to the TMLR audience; I therefore support its acceptance.

**Audience:**

Yes

**Audience Explanation:**

Yes. The paper lies at the intersection of federated learning, reinforcement learning, and control theory, three areas of strong current interest to the TMLR community. The results advance our understanding of model-free control in the heterogeneous federated setting, providing novel mathematical tools that could inspire further work in both theory and applications.

**Broader Impact Concerns:**

There are no evident ethical or societal risks associated with this work, as it is primarily theoretical in nature.

**Claims And Evidence:**

Yes

**Claims Explanation:**

Yes. The theoretical claims are carefully proven, with all assumptions stated explicitly and supported by intuition and examples. The numerical results qualitatively match the theoretical trends (improvement with larger $M$, degradation with increased heterogeneity). While empirical validation is limited, the theoretical development is sufficiently rigorous and complete to justify the main claims.

**Requested Changes:**

-  The authors should justify how Assumption~2 can be satisfied in practice, for instance by extending or adapting the initialization techniques proposed in [1]. Alternatively, they could explicitly acknowledge that finding such a common stabilizing controller is non-trivial in heterogeneous, model-free settings and leave this as an open direction for future work.

-  Address minor inconsistencies noted in Theorem 2 and Lemma 18 regarding the definition of $\epsilon'$ and the heterogeneity term $(\epsilon_1 h_{\text{het}}^1 + \epsilon_2 h_{\text{het}}^2)^2$.

- The bias term can greatly degrade learning. A well-known technique to learn a common optimal policy without suffering from bias is the SCAFFOLD-type method. Can the authors add a paragraph on why it is more advantageous to use their methods rather than SCAFFOLD-type methods?

- Can the authors add the following missing references in federated reinforcement learning:
(i)Zhang et al, 2025, Gap-Dependent Bounds for Federated Q-learning, ICML
(ii) Labbi et a,l 2025, Federated UCBVI: Communication-efficient federated regret minimization with heterogeneous agents, AISTATS


[1] Zhao et al, 2022,  Convergence and Sample Complexity of Policy Gradient Methods for Stabilizing Linear Systems , IEEE

---

> ### Author Response · Authors · 2025-11-23
> **Official Comments by the Authors to Reviewer gAUU**
>
> **On the initial stabilizing controller**: Regarding Assumption 2 (initial stabilizing controller) please refer to our general comment.
>
> **On the experimental section**: While we agree that larger-scale empirical studies can be informative, we emphasize that the main contribution of our work is theoretical. The primary focus of the paper is to provide a rigorous finite-time convergence and stability analysis for federated policy gradient methods under heterogeneous dynamical systems, an aspect that is technically challenging and constitutes the central novelty of our contribution. The numerical example we included was designed to showcase the theory at work and contrast with the single system $M=1$ and identical agent setting. Detailed case studies and comparisons with more complicated systems are beyond the scope of this work.
>
> We note that our work is also consistent with the broader literature on policy gradient methods for control, numerical experiments are typically conducted on low-dimensional systems. For instance, several works on model-free policy gradient methods for LQR also use synthetic low-dimensional benchmarks [1]-[4]. Our experimental setting follows this standard practice.
>
> Moreover, increasing the dimensionality of the systems would not substantially change the qualitative behaviors already demonstrated, namely, the benefits of collaboration under low heterogeneity and the interplay between heterogeneity-induced bias and variance reduction.
>
> [1] Malik et al. (2020). Derivative-free methods for policy optimization: Guarantees for linear quadratic systems. JMLR.
>
> [2] Gravell et al. (2020). Learning optimal controllers for linear systems with multiplicative noise via policy gradient. IEEE TAC.
>
> [3] Fujinami et al. (2025). Policy Gradient for LQR with Domain Randomization. arXiv.
>
> [4] Zhao et al. (2025). Asynchronous Parallel Policy Gradient Methods for the Linear Quadratic Regulator. IEEE TAC.
>
> **On the typos on page 53 and Lemma 18**:  We thank the Reviewer for carefully checking the details of the proof and for pointing out these typos. We have corrected them in the revised manuscript.
>
> **On the application of control-variate methods**: We thank the Reviewer for raising this important point. Let us now explain that there is a fundamental difference between applying control-variate techniques in optimization (as is done in SCAFFOLD) versus applying them in our dynamical systems context. To see this difference, recall that in standard FL, the goal is to minimize $f(x)= (1/M) \sum_{i\in [M]} f_i(x)$. In algorithms like SCAFFOLD, a bias-correction term is added to the local update direction of each agent to ensure that the global iterates maintained by the server follow the "ideal" gradient-descent update scheme: $x_{n+1}= x_{n}- \alpha \nabla f(x_n)$. Now, if the function $f(x)$ is $\mu$-strongly convex, we can use the *gradient-domination* property $|| \nabla f(x) ||^2 \geq 2 \mu (f(x) - f(x^\star))$ to ensure exponentially fast convergence to the minimizer. Applying essentially the same bias-correction idea to our setting, one can cause the server-maintained controller sequence ${K_n}$ to (approximately) evolve as $K_{n+1} \approx K_{n}- \alpha \nabla C_{\text{avg}}(K_n)$. *However, this is the extent to which bias-correction/control-variates can help us.* To understand why, we note that the gradient-domination property for the LQR problem is associated with a dynamical system. When we average LQR costs from $M$ agents, *it is completely unclear whether the average cost is the LQR cost of some "averaged" dynamical system.* In the absence of this equivalence between the average LQR cost and a corresponding dynamical system, we cannot just write down a gradient domination property for the average cost.
>
> **When can control-variates help?** A notable exception where control variates can in fact continue to help is the scenario where all agents share the same dynamics, but differ just in their cost functions (capturing diverse tasks), i.e., a multi-task scenario.  In this case, minimizing the average LQR cost boils down to solving an LQR problem for the tuple $(A, B, \bar{Q}, \bar{R})$, where $\bar{Q}= (1/M) \sum_{i \in [M]} Q_i$ and $\bar{R}= (1/M) \sum_{i \in [M]} R_i$ are the averaged cost matrices. The recent paper [1] below has explored this idea in the context of finite state-action MDPs, and the extension to the LQR setting remains open (especially under noisy, imperfect gradients). While this is an interesting direction to pursue, it is not within the scope of our paper where we focus primarily on heterogeneity in system dynamics (and not tasks).
>
> We have now added a brief discussion summarizing the above points.
>
> **On the constant $c_{\text{uni},2}$**: We have now substituted by its value in the revised version.
>
> **On the additional references in federated RL**: We thank the Reviewer for pointing out these relevant recent works. We now include them in the revised manuscript.

---

> > ### Comment · Reviewer_gAUU · 2025-12-07
> >
> > I thank the authors for their detailed response, which successfully addressed all of my concerns.

---

### Author Response · Authors · 2025-11-23
**General Comments by Authors**

First of all, we thank the Reviewers for their valuable and constructive feedback. We appreciate the time and effort invested in evaluating our work, and the insightful comments that have helped us improve the clarity and presentation of the manuscript. Below, we address the Reviewers' comments in detail, clarifying all raised questions and highlighting (in purple) the changes made to the paper in response to their suggestions.

$\bullet$ **Initial Stabilizing Controllers (Assumption 2):**  All Reviewers raised questions regarding the existence of an initial controller $K_0$ that simultaneously stabilizes all systems. We emphasize that this assumption is standard and unavoidable in model-free LQR, as without a stabilizing controller, neither the cost function nor the policy gradient is well defined. Our analysis therefore requires the existence of *some* stabilizing $K_0$, but places no restriction on its suboptimality, it may be arbitrarily far from any $K_i^\star$ in terms of cost.

Importantly, such a common stabilizing controller can be *learned directly from data*. Recent work has shown that solving a sequence of discounted multitask LQR problems using the *average* $C_{\mathrm{avg}}(K) = \frac{1}{M} \sum_{i = 1}^M C^{(i)}(K)$ cost yields a stabilizing controller for all systems under low heterogeneity conditions [1]. This provides a practical, data-driven approach for obtaining an initial stabilizing policy even in heterogeneous multi-agent settings. We now include a brief discussion of this paper and clarify that Assumption 2 can be satisfied via these existing initialization procedures.

$\bullet$ **Reemphasizing the Main Challenges of Federated Learning for Control:** We take this opportunity to clearly restate the core challenges that distinguish FedLQR from classical federated optimization or single-agent LQR analyses.

First, when system dynamics differ across agents, each individual cost $C^{(i)}(K)$ satisfies a PL (gradient-dominance) condition, but their average $C_{\mathrm{avg}}(K)$ *does not*. Unlike the strongly convex setting, where averaging preserves strong convexity, the PL property is not preserved under averaging (unless systems share the same dynamics). Consequently, global convergence guarantees from the single-agent case do not carry over, and any gradient-based method must converge only *up to* an unavoidable heterogeneity-induced bias.

Second, although methods such as SCAFFOLD can eliminate local drift in standard federated learning, they rely on the assumption that all agents share the *same* underlying system dynamics. Recent work in federated RL confirms this, as control-variate corrections remove drift only when the system dynamics are identical across clients [2]. In the heterogeneous control setting we consider, such techniques cannot eliminate the fundamental mismatch between $\nabla C_i(K)$ and $\nabla C_j(K)$, and thus cannot remove the heterogeneity bias.

Finally, in contrast to stochastic optimization, ensuring that all controllers remain *stabilizing* at every iteration introduces additional constraints on step sizes, drift bounds, and gradient-estimation error. This requirement is unique to control and substantially complicates the analysis of FedAvg.

Our work directly addresses these challenges by (i) characterizing the heterogeneity bias inherent to the problem, (ii) proving per-iteration stability under system heterogeneity, and (iii) providing convergence and sample-complexity reduction guarantees despite the absence of a PL condition for the averaged objective.

[1] Fujinami, T., Lee, B. D., Matni, N., Pappas, G. J. (2025). Policy Gradient for LQR with Domain Randomization. arXiv:2503.24371.

[2] Zhu, F., Heath, R. W., Mitra, A. (2024). Towards fast rates for federated and multi-task reinforcement learning. Proceedings of the 63rd IEEE Conference on Decision and Control (CDC).

---

### Decision · Action_Editor_TXGh · 2026-01-07

**Recommendation:** Accept as is

**Additional Comments:**

While the paper meets TMLR’s standards for acceptance, the reviewers identified several areas where clarity and impact could be further improved. In particular, reviewers had comments about initialization assumptions, presentation and exposition, and positioning of novelty. The authors have given a satisfactory response to these comments. Please make sure that these changes are reflected in the final version of the manuscript.

**Audience:**

Yes

**Audience Explanation:**

Yes. All reviewers agreed that the paper is well aligned with the interests of the TMLR audience. The work sits at the intersection of federated learning, reinforcement learning, and control theory, and provides rigorous finite-time analysis of federated, model-free LQR under system heterogeneity. Reviewers noted that the paper offers conceptual value by bridging tools from policy-gradient analysis, federated optimization, and dynamical systems theory, and that it addresses challenges, such as per-iteration stability under heterogeneity, that are unique to control settings and largely absent from standard federated learning literature. As such, the paper is expected to be of sustained interest to researchers in learning theory, control, and federated reinforcement learning.

**Claims And Evidence:**

Yes

**Claims Explanation:**

Yes. Across all three expert reviews, there is a strong consensus that the paper’s claims are supported by rigorous and careful theoretical analysis. Reviewers independently verified the correctness of the main proofs and confirmed that the assumptions are clearly stated and appropriately justified within the context of policy-gradient methods for LQR. While reviewers initially raised concerns regarding technical clarity (e.g., minor typographical errors, ambiguity in definitions, interpretation of convergence guarantees, and overstatement of global optimality), these issues were comprehensively addressed in the authors’ rebuttal. In particular, the authors clarified the role of heterogeneity-induced bias, corrected identified inaccuracies, refined misleading statements, and strengthened the exposition around stability, convergence, and gradient bias. Following the rebuttal, all reviewers explicitly agreed that the claims are accurate, well-supported, and convincingly demonstrated, and each updated their assessment to affirm the validity of the evidence.